# Measuring and Enhancing Trustworthiness of LLMs in RAG through Grounded Attributions and Learning to Refuse

**Maojia Song**[*]**, Shang Hong Sim**[*]**, Rishabh Bhardwaj**
Singapore University of Technology and Design
{maojia_song, shanghong_sim, rishabh_bhardwaj}@mymail.sutd.edu.sg

**Hai Leong Chieu**
DSO National Laboratories
chaileon@dso.org.sg

**Navonil Majumder, Soujanya Poria**
Singapore University of Technology and Design
{navonil_majumder, sporia}@sutd.edu.sg

## Abstract

LLMs are an integral component of retrieval-augmented generation (RAG) systems. While many studies focus on evaluating the overall quality of end-to-end RAG systems, there is a gap in understanding the appropriateness of LLMs for the RAG task. To address this, we introduce Trust-Score, a holistic metric that evaluates the trustworthiness of LLMs within the RAG framework. Our results show that various prompting methods, such as in-context learning, fail to effectively adapt LLMs to the RAG task as measured by Trust-Score. Consequently, we propose Trust-Align, a method to align LLMs for improved Trust-Score performance. 26 out of 27 models aligned using Trust-Align substantially outperform competitive baselines on ASQA, QAMPARI, and ELI5. Specifically, in LLaMA-3-8b, Trust-Align outperforms FRONT on ASQA (↑12.56), QAMPARI (↑36.04), and ELI5 (↑17.69). Trust-Align also significantly enhances models' ability to correctly refuse and provide quality citations. We also demonstrate the effectiveness of Trust-Align across different open-weight models, including the LLaMA series (1b to 8b), Qwen-2.5 series (0.5b to 7b), and Phi3.5 (3.8b). We release our code at https://github.com/declare-lab/trust-align.

## 1 Introduction

LLMs are widely used for information retrieval but often produce hallucinations—factually incorrect yet convincing responses (Ji et al., 2023), undermining their reliability. A common mitigation is Retrieval-Augmented Generation (RAG), which integrates external knowledge to improve correct token generation, reducing perplexity (Khandelwal et al., 2019) and enhancing downstream tasks like machine translation (Zheng et al., 2021) and classification (Bhardwaj et al., 2023). Connecting LLMs to external documents via retrieval also improves response quality (Shuster et al., 2021; Béchard & Ayala, 2024), further enhanced by attribution mechanisms (Gao et al., 2023b; Hsu et al., 2024).

In this paper, we investigate LLMs' ability to ground responses in provided documents instead of relying on their *parametric* knowledge from training. A response is considered grounded if it correctly answers using only the attached documents, with in-text citations supporting its claims. Key aspects include LLMs' *refusal* capability—whether they abstain from answering when documents lack sufficient information. Additionally, we analyze their overall tendency to answer, the fraction of claims grounded in documents, and whether cited sources substantiate generated statements.

To comprehensively understand LLMs' groundedness, we propose a new metric **Trust-Score**. It assesses an LLM across multiple dimensions: 1) The ability to discern which questions can be

---

[*]Equal contribution.

answered or refused based on the provided documents (Grounded Refusals); 2) The correctness of LLM response for the answerable questions; 3) The extent to which generated statements are supported by the corresponding citations; and 4) The relevance of the citations to the statements. Unlike existing metrics that primarily assess the overall performance of RAG systems (Gao et al., 2023b)—where a weak retriever can significantly decrease the scores—TRUST-SCORE is designed to specifically measure the LLM's performance within a RAG setup, isolating it from the influence of retrieval quality.

Our investigation in Section 6.1 shows that many state-of-the-art systems, including GPT-4 and Claude-3.5-Sonnet, heavily rely on their parametric knowledge to answer questions (OpenAI, 2023; Anthropic, 2024). This reliance limits their suitability for RAG tasks, where models should base responses solely on the provided documents, resulting in a low TRUST-SCORE. Additionally, prompting approaches intended to enhance model groundability have proven ineffective, as models become overly sensitive to the prompt, leading to exaggerated refusals or excessive responsiveness shown in Appendix F.4. To enhance the groundedness of LLMs, i.e., achieve a higher TRUST-SCORE, we propose an alignment method, **TRUST-ALIGN**. This approach first constructs an alignment dataset consisting of 19K questions, documents, positive (preferred) responses, and negative (unpreferred) responses. The dataset covers a range of LLM errors—Inaccurate Answers, Over-Responsiveness, Excessive Refusal, Over-Citation, and Improper Citation. We regard these errors as LLM hallucinations within an RAG framework.

Evaluations on the benchmark datasets ASQA, QAMPARI, and ELI5 show that models trained with TRUST-ALIGN outperform the competitive baselines on TRUST-SCORE in 26 out of 27 model family and dataset configurations. Notably, in LLaMA-3-8b, TRUST-ALIGN achieves substantial improvements over Huang et al. (2024b) FRONT, a leading baseline, with respective gains of 12.56% (ASQA), 36.04% (QAMPARI), and 17.69% (ELI5). Additionally, TRUST-ALIGN substantially enhances the ability of models to correctly refuse or provide grounded answers in all 27 model family and dataset configurations, with LLaMA-3-8b showing increases of 23.87%, 47.95%, and 45.77% correct refusals compared to FRONT. Citation groundedness scores also improved in 24 out of 27 model family and dataset configurations, with notable increases of 22.12%, 38.35%, and 5.55% in LLaMA-3-8b compared to FRONT. Due to the gamification of the metric, where parametric knowledge can artificially inflate the scores, we notice mixed results on answer correctness scores. Specifically, we observe a notable increase in answer correctness scores for all models in QAMPARI, 5/9 models in ELI5, and 2/9 models for ASQA.

Our key contributions to this work are as follows:

- We study LLM groundedness problem, where model model responses should be derived from retrieved documents (external memory) rather than the parametric knowledge (knowledge stored in model parameters).

- To measure LLM's groundedness under RAG, we introduce **TRUST-SCORE**, a holistic metric for quantifying LLM's grounding errors.

- We propose **TRUST-ALIGN**, an alignment approach designed to improve the trustworthiness of LLMs in RAG (Figure 2). It first creates an alignment dataset of 19K samples with paired positive and negative responses, followed by aligning the model using direct preference optimization (DPO) (Rafailov et al., 2024b).

**Comparison with existing approaches.** Current evaluations of RAG focus on the overall system performance (Gao et al., 2023b; Xu et al., 2024), conflating the effects of retriever quality and LLM performance in the metric scores (Fan et al., 2024). This highlights the need for new ways to measure LLM effectiveness in RAG systems without the influence of the retriever. The work by Thakur et al. (2024) is closest to ours, as it analyzes the refusal capabilities of LLMs in a RAG context but lacks holistic evaluation, as it does not account for both response and citation groundedness. On the other hand, Ye et al. (2024); Hsu et al. (2024); Huang et al. (2024b) propose frameworks to improve LLM response groundedness but overlook refusal behaviors in their metrics. Ignoring refusal behaviors, retriever influence, citation and answer groundedness weakens the ability of current metrics to effectively measure LLM performance in RAG. TRUST-SCORE comprehensively evaluates LLM performance, including refusal, citation, and answer groundedness, while TRUST-ALIGN creates a corresponding alignment dataset, making the metric and approach more unique and holistic for LLM evaluations and alignment in RAG. A more detailed comparison can be found in Appendix C.

## 2 PROBLEM DESCRIPTION

### 2.1 TASK SETUP

Given a question $q$ and a set of retrieved documents $D$ as input, the LLM is instructed to generate a response $S$ which consists of a set of citation-grounded statements $\{s_1, \ldots, s_n\}$; each statement $s_i$ follows a set of inline citations $\mathcal{C}_i = \{c_{i,1}, c_{i,2}, \ldots\}$ referring to the documents in $D$. If $D$ is not sufficient to answer $q$, the gold response would be a refusal statement[1], such as, *"I apologize, but I couldn't find an answer to your question in the search results"*. Otherwise, the response would follow the pattern: "statement1 [1][2] statement2 [3]" where [1][2] and [3] denote the enumeration of documents that supports each statement respectively.

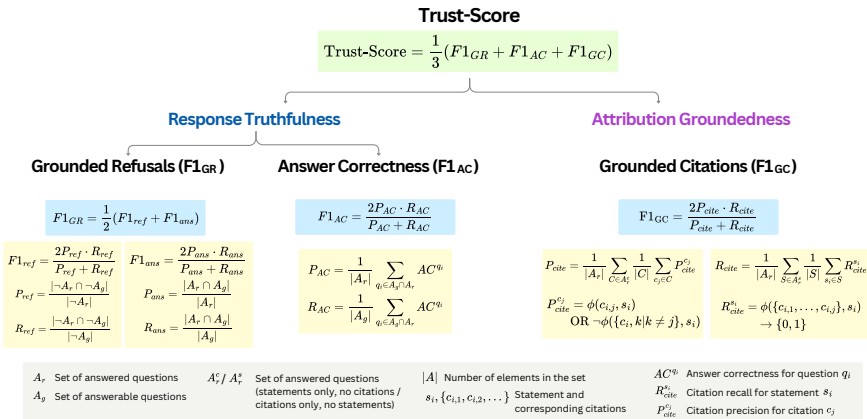

Figure 1: TRUST-SCORE calculation shown as a computational graph.

### 2.2 ON ANSWERABILITY OF A QUESTION

To label if a response should be a refusal or consist of claims, we define the notion of **answerability**. A question $q$ is considered answerable if $D$ contains sufficient information to answer $q$. Formally, we label a question as answerable if a subset of the retrieved documents entails at least one of the gold claims; otherwise, $q$ is unanswerable and thus should result in a ground truth **refusal**. A refusal response contains no claims or citations but provides a generic message conveying the LLM's inability to respond to $q$.

### 2.3 HALLUCINATION IN LLM IN RAG

We define an LLM's response as grounded when it correctly answers a question using only the information in the documents, and the response can be inferred from the inline citations to those documents. When a response is not grounded, it is considered a case of hallucination. We define **hallucination** as an erroneous LLM response, categorized into five types: (1) *Inaccurate Answer* – The generated statements $S$ fail to cover the claims in the gold response, (2) *Over-Responsiveness* – The model answers a question that should result in a refusal, (3) *Excessive Refusal* – The model refuses to answer a question that is answerable, (4) *Overcitation* – The model generates redundant citations, and (5) *Improper Citation* – The citations provided do not support the statement. Next, we introduce a comprehensive metric to concretely measure hallucinations in LLMs, i.e., to assess an LLM's groundedness or trustworthiness[2].

---

[1]There are many applications where LLM parametric knowledge use is expected and retrieved documents serve to improve the LLM's response. However, in this paper we study the problem of complete groundedness—i.e., all claims should be documents derivable, making this an IR task.

[2]In this paper, we use LLM groundedness and trustworthiness interchangeably in the context of RAG.

## 3 METRICS FOR LLM-IN-RAG

Given a question $q$ and the corresponding ground truth response $A_G = \{a_{g1}, \ldots, a_{gn}\}$ consisting of gold claims, we define the claims obtainable from the provided documents as $A_D = \{a_{d1}, \ldots, a_{dn}\}$ and the claims generated in the response as $A_R = \{a_{r1}, \ldots, a_{rn}\}$. We aim to measure two aspects of an LLM in RAG: 1) the correctness of the generated claims (Response Truthfulness); and 2) the correctness of citations generated (Attribution Groundedness).

**Insufficiency of the existing metrics.** Gao et al. (2023b) measure Response Truthfulness by first computing the per-sample Answer Correctness recall ($AC_{reg}^q$) score for gold claims $A_G$, disregarding how many of these claims are obtainable from $D$. This is followed by averaging the recall scores across samples to obtain a single score for the dataset. This method introduces inconsistencies: models that rely on parametric knowledge ($\mathcal{M}_p$) may generate gold claims not found in $D$, leading to an artificially inflated recall value. In contrast, an ideal LLM ($\mathcal{M}_i$) would rely solely on $D$ to generate responses (a desired trait) and would be constrained by an upper recall limit of $\frac{|A_G \cap A_D|}{|A_G|}$, which varies depending on the question. This approach presents two key problems: (1) *Recall Consolidation*: Since the measurement range depends on the claims present in $D$, it is infeasible to provide a consistent, consolidated $AC_{reg}$ score across the dataset, (2) *Recall Gamification*: $\mathcal{M}_p$ may have a higher upper limit on $AC_{reg}$ (up to 1) because they can generate gold claims not present in $D$ (an undesirable trait), unlike $\mathcal{M}_i$ that depend entirely on $D$.

**Answer Calibration.** To address the challenges of recall consolidation and gamification in existing evaluation metrics, we propose new metrics that measure sample-wise recall score based on the fraction of gold claims ontainable from $D$. Specifically, this involves computing $|A_G \cap A_D|$, which measures the Answer Correctness (AC) recall after calibrating the gold claims. This approach sets a maximum recall limit of 1 for all models. For dataset-wide scoring, we consolidate per-sample AC scores using two methods: 1) $\mathbf{P_{AC}}$: The average AC score across samples *answered* by the LLM, i.e., samples where $A_R \neq \emptyset$, reflecting a precision oriented perspective; 2) $\mathbf{R_{AC}}$: The average AC score across samples that are *answerable*, i.e., samples where $A_G \cap A_D \neq \emptyset$, reflecting a recall oriented perspective[3]. These metrics, illustrated in Fig. 1, are then combined into a single score, $\mathbf{F1_{AC}}$, which serves as a comprehensive measure of how well the LLM grounds its claims on the document $D$. This combined metric not only facilitates the consolidation of recall but also addresses issues related to recall gamification.

**Scoring refusals.** An important capability of an LLM in RAG is its ability to identify when a response is unanswerable based on the provided documents $D$. To measure this, we introduce a metric called Grounded Refusals. This metric evaluates the model's refusal performance by calculating dataset-wide precision and recall for both ground-truth answerable cases and refusals. These values are then combined into their respective F1 scores, $\mathbf{F1_{ref}}$ for refusals and $\mathbf{F1_{ans}}$ for answerable cases. The final score, $\mathbf{F1_{GR}}$, is the average of these two F1 scores, as shown in Figure 1.

**Measuring attribution groundedness.** While Response Truthfulness metrics like $\mathbf{F1_{AC}}$ and $\mathbf{F1_{GR}}$ evaluate the quality of generated claims, it is equally important to measure how well these statements are supported by relevant citations—what we call Attribution Groundedness. To this end, we adopt two sub-metrics from (Gao et al., 2023b): Citation Recall ($\mathbf{R_{cite}}$) and Citation Precision ($\mathbf{P_{cite}}$). To compute $\mathbf{R_{cite}}$, we first determine if a generated statement $s_i$ is supported by its cited documents using an NLI model[4], thus obtaining sample-wise recall scores $\mathbf{R_{cite}}^{s_i}$. Then we take the mean across all samples to obtain the final $\mathbf{R_{cite}}$ score (Figure 1). To compute $\mathbf{P_{cite}}$, we first score each citation $c_{i,j}$ of a statement $s_i$, followed by computing the average across citations in a response $S$ (sample-wise score). The dataset-wide citation score is computed by averaging the citation scores across all the samples. To quantify the Groundedness of Citations, we compute $\mathbf{F1_{GC}}$, the harmonic mean of $\mathbf{P_{cite}}$ and $\mathbf{R_{cite}}$. A detailed breakdown of this metric is provided in Appendix D and Figure 1.

Thus, we define a new metric $\textbf{TRUST-SCORE} = \frac{1}{3}(\mathbf{F1_{GR}} + \mathbf{F1_{AC}} + \mathbf{F1_{GC}})$.

---

[3] Notably, both $\mathbf{P_{AC}}$ and $\mathbf{R_{AC}}$ sum over samples that are both answered and answerable, differing primarily in their normalization values.

[4] An NLI model checks if the cited document entails the statement.

**Responsiveness.** To measure the answering tendency of an LLM, we define **Responsiveness**. It is the fraction of answered questions, denoted by the Answered Ratio (AR %), which is calculated as $AR \% = \frac{|A_r|}{|A_g| + |\neg A_g|}$. $|A_r|$, $|A_g|$, and $|\neg A_g|$ are the number of answered, answerable, and unanswerable questions respectively. A model is expected to show a high AR% for answerable questions and a low AR% for unanswerable ones, with the scores expected to align with the dataset distribution.

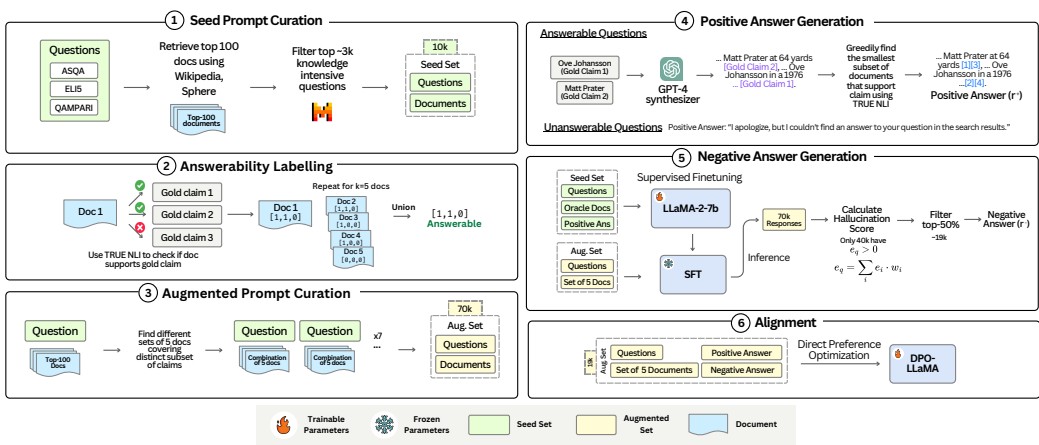

Figure 2: Overview of the TRUST-ALIGN. **Left**: The curation of both seed and augmented prompts (Q-D pairs) and an example of the answerability labeling process during the retrieval stage. **Right**: The response paired data generation process. First, we obtain positive answers and then select hard negative answers. Finally, we align our model via DPO.

## 4 THE TRUST-ALIGN DATASET

To align LLMs towards trustworthiness, we propose a new approach, **TRUST-ALIGN**. The approach constructs an LLM trustworthiness alignment dataset, where each sample in the dataset consists of a question $q$, a set of retrieved documents $D$, and a pair of positive (preferred) and negative (unpreferred) responses ($r^+$, $r^-$). The positive response corresponds to an answer that encompasses expected gold claims for $q$ and corresponding citations referring to the documents. If $D$ is not sufficient to answer $q$, $r^+$ is assigned a refusal response, while $r^-$ is its non-refusal counterpart. We build the dataset in multiple steps: 1) Obtain a set of high-quality and diverse questions, 2) Obtain documents for each question, 3) Augmenting $(q, D)$ pairs that cover diverse hallucination types, 4) Construct positive responses entailing gold claims, and 5) Construct negative (unpreferred) responses by prompting a fine-tuned model and observing its hallucinations. We relegate fine-grained details about the dataset to Figure 2 and Appendix E.

**Collecting quality questions.** The dataset construction begins by collecting a set of high-quality and diverse questions from the training splits of ASQA, QAMPARI, and ELI5, referred to as **seed samples**. We first divide the questions into $k$ clusters and use Mixtral-8x7B to assign each a quality score from 1 to 7, based on how difficult they are to answer without additional information. Clusters with scores of 4 or higher are selected. Next, we sample questions from the clusters of each dataset to construct approximately 10K questions in the seed set.

**Collecting $D$'s.** Next, we collect relevant documents for each question in the seed set by querying Wikipedia and Common Crawl, retrieving the top 100 documents. We filter out seed questions where relevant documents are not retrieved. We then identify 5 documents that perform as well as the full

100 in terms of EM recall, referring to these as ***oracle*** documents for question $q$.[5] Gold claims for each $q$ are sourced from the respective datasets.

**Augmenting *(q,D)* set.** Using the questions and oracle documents, we create diverse samples (i.e., varying combinations of relevant and irrelevant documents) to trigger multiple hallucinations from LLMs (Section 2.3). The document order is shuffled to avoid citation bias. To construct unanswerable questions, we select documents similar to those entailing gold claims but still irrelevant to $q$. This process results in approximately 70K question-document pairs.

**Obtaining $\mathbf{r}^+$ and $\mathbf{r}^-$.** To generate preferred responses, by prompting GPT-4, we stitch together the gold claims and citations[6]. For unanswerable questions, we assign a ground truth refusal response. To obtain quality negative (unpreferred) responses, we fine-tune LLaMA-2-7b on the source datasets, creating $\mathcal{M}_{sft}$. Testing $\mathcal{M}_{sft}$ on the 70K dataset identified 40K responses with hallucinations. Table 1 shows hallucination severity ($e_i$) and frequency ($w_i$). To obtain good negative samples, we first rank each of the 40K responses according to their severity score $e_q$, where $e_q = \sum_i e_i \cdot w_i$. We then select the top 50%[7] of the corresponding samples for both answerable and unanswerable responses. We perform DPO using this set of 19k samples to obtain the final aligned model.

## 5 EXPERIMENTAL SETUP

**Models studied.** To comprehensively measure performance of open-source models, we perform TRUST-SCORE computations on vanilla and TRUST-ALIGNed version of a range of open-weight models such as LLaMA series (LLaMA-2-7b, LLaMA-2-13b, LLaMA-2-13b, etc.), Qwen series (Qwen-2.5-0.5b, Qwen-2.5-7b, etc.) and Phi3.5-mini. See Appendix H.1 for more details.

Table 1: Fraction of each hallucination amongst all the observed hallucinations in $\mathcal{M}_{sft}$ (40,985), with possible overlap. $w_i$ shows the severity computation of each hallucination. $I_{\text{condition}} = 1$ if condition is True otherwise it is 0. See Fig. 5 for the detailed breakdown of the last three errors.

| Hallucination Type (HT) | Frequency ($w_i$) | Severity ($e_i$) |
|---|---|---|
| Unwarranted Refusal | 8,786 | 0.50 | $I_{(A_g \neq \emptyset, A_r = \emptyset)}$ |
| Over Responsiveness | 13,067 | 0.50 | $I_{(A_g = \emptyset, A_r \neq \emptyset)}$ |
| Overcitation | 12,656 | 0.34 | 1 - CP |
| Improper Citation | 9,592 | 0.26 | 1 - CR |
| Inaccurate Claims | 14,783 | 0.40 | 1 - F1$_{AC}$ |

**Evaluation datasets.** We evaluate on the test-set of attributable factoid and long-form question-answering tasks from ASQA (Stelmakh et al., 2023), QAMPARI (Amouyal et al., 2023), and ELI5 (Fan et al., 2019). Additionally, we include ExpertQA (Malaviya et al., 2024) for OOD evaluations. For each question, we append the top 5 retrieved documents. For ELI5 and ExpertQA, the ground truth answers are decomposed into three claims. The dataset statistics are detailed in Appendix H.2.

**Baselines.** Models[8] trained with TRUST-ALIGN are compared against the following baselines:

- ICL (Gao et al., 2023b): Prepends two demonstrations to each query, consisting of an example query, top-5 retrieved documents, and an inline cited answer
- PostCite (Gao et al., 2023b): Generates an uncited answer in a closed-book setting, then retrieves most similar documents from top-5 documents using GTR for citations.
- PostAttr (Gao et al., 2023b): Similar to POSTCITE, produces an uncited response in a closed-book setting, but uses the TRUE-NLI model to find the best matching citation among top-5 documents.
- Self-RAG (Asai et al., 2024): Trains the LLM to retrieve relevant documents on demand using reflection tokens, enhancing generation quality. We evaluated the provided 7b and 13b model checkpoints from HF using the default settings.
- FRONT (Huang et al., 2024b): Uses a fine-grained attribution framework to improve grounding and citation quality. We followed the provided instructions to train a 7b model for comparison.

---

[5]Clustering and document retrieval details are in Appendix E.

[6]Prompt template can be found at Table 23.

[7]See Appendix F.8 for more details on this hyperparameter.

[8]All models used are instruct tuned or chat versions.

# 6 RESULTS AND ANALYSIS

Table 2: LLaMA family evaluated on the ASQA, QAMPARI, and ELI5 datasets. Best values within each family are highlighted ). **AR%** := Answered Ratio in %; **F1$_{AC}$** := Answer Correctness F1; **F1$_{GR}$** := Grounded Refusals F1; **F1$_{GC}$** := Grounded Citations F1; **TRUST** := TRUST-SCORE; **Resp.** := Responsiveness; **Att-Grd.** := Attribution Groundedness.

| Model | Type | ASQA (610 answerable, 338 unanswerable) | | | | | QAMPARI (295 answerable, 705 unanswerable) | | | | | ELI5 (207 answerable, 793 unanswerable) | | | | |
|---|---|---|---|---|---|---|---|---|---|---|---|---|---|---|---|---|
| | | Resp. | Trustworthiness | | | | Resp. | Trustworthiness | | | | Resp. | Trustworthiness | | | |
| | | AR (%) | Truthfullness | | Att-Grd. | TRUST | AR (%) | Truthfullness | | Att-Grd. | TRUST | AR (%) | Truthfullness | | Att-Grd. | TRUST |
| | | | F1$_{AC}$ | F1$_{GR}$ | F1$_{GC}$ | | | F1$_{AC}$ | F1$_{GR}$ | F1$_{GC}$ | | | F1$_{AC}$ | F1$_{GR}$ | F1$_{GC}$ | |
| LLaMA-2 -7b | ICL | 0.00 | 0.00 | 26.28 | 0.00 | 8.76 | 0.00 | 0.00 | 41.35 | 0.00 | 13.78 | 0.50 | 0.00 | 46.71 | 0.00 | 15.57 |
| | PostCite | 10.44 | 0.07 | 35.23 | 0.00 | 11.77 | 34.40 | 0.00 | 57.34 | 9.50 | 22.28 | 0.90 | 1.86 | 44.98 | 5.04 | 17.29 |
| | PostAttr | 10.44 | 0.07 | 35.23 | 0.00 | 11.77 | 34.40 | 0.00 | 57.34 | 3.78 | 20.37 | 0.90 | 1.86 | 44.98 | 0.00 | 15.61 |
| | Self-RAG | 100.00 | 45.19 | 39.15 | 63.49 | 49.28 | 96.00 | 6.81 | 28.23 | 19.95 | 18.33 | 73.50 | 14.94 | 40.20 | 13.80 | 22.98 |
| | FRONT | 100.00 | 60.47 | 39.15 | 68.86 | 56.16 | 100.00 | 17.27 | 22.78 | 24.26 | 21.44 | 100.00 | 21.66 | 17.15 | 52.72 | 30.51 |
| | TRUST-ALIGN (DPO) | 65.30 | 52.48 | 66.12 | 83.94 | 67.51 | 32.30 | 32.03 | 71.67 | 49.42 | 51.04 | 21.60 | 22.54 | 63.27 | 47.35 | 44.39 |
| LLaMA-2 -13b | ICL | 17.41 | 21.52 | 41.40 | 13.83 | 25.58 | 26.50 | 0.44 | 59.57 | 0.00 | 20.00 | 46.40 | 19.97 | 54.81 | 4.73 | 26.50 |
| | PostCite | 90.51 | 2.21 | 49.91 | 1.53 | 17.88 | 100.00 | 0.00 | 22.78 | 8.05 | 10.28 | 76.60 | 2.27 | 38.05 | 0.72 | 13.68 |
| | PostAttr | 90.51 | 2.21 | 49.91 | 0.17 | 17.43 | 100.00 | 0.00 | 22.78 | 2.95 | 8.58 | 76.60 | 2.27 | 38.05 | 0.09 | 13.47 |
| | Self-RAG | 100.00 | 48.52 | 39.15 | 69.79 | 52.49 | 72.70 | 2.71 | 48.58 | 26.91 | 26.07 | 22.10 | 12.77 | 58.68 | 24.54 | 32.00 |
| LLaMA-3.2 -1b | ICL | 60.23 | 35.95 | 50.94 | 9.96 | 32.28 | 19.20 | 6.32 | 52.64 | 0.38 | 19.78 | 88.40 | 12.87 | 27.10 | 5.23 | 15.07 |
| | PostCite | 43.57 | 0.59 | 50.22 | 0.24 | 17.02 | 41.20 | 0.32 | 49.79 | 1.61 | 17.24 | 18.40 | 2.04 | 50.88 | 1.02 | 17.98 |
| | PostAttr | 45.78 | 0.48 | 48.42 | 0.00 | 16.30 | 34.00 | 0.63 | 48.43 | 0.21 | 16.42 | 18.40 | 2.04 | 50.88 | 0.07 | 17.66 |
| | FRONT | 79.11 | 48.22 | 54.48 | 48.29 | 50.33 | 98.60 | 7.57 | 24.54 | 15.32 | 15.81 | 97.20 | 16.11 | 20.76 | 30.19 | 22.35 |
| | TRUST-ALIGN (DPO) | 41.67 | 38.64 | 58.61 | 79.35 | 58.87 | 20.00 | 27.22 | 67.92 | 49.42 | 48.19 | 9.60 | 13.20 | 59.35 | 48.21 | 40.25 |
| LLaMA-3.2 -3b | ICL | 1.27 | 2.04 | 27.98 | 53.95 | 27.99 | 34.10 | 16.06 | 59.65 | 12.87 | 29.53 | 21.90 | 18.55 | 55.56 | 30.70 | 34.94 |
| | PostCite | 47.26 | 31.03 | 56.59 | 22.99 | 36.87 | 39.60 | 6.34 | 55.22 | 6.83 | 22.80 | 92.80 | 18.12 | 25.14 | 4.44 | 15.90 |
| | PostAttr | 47.15 | 29.76 | 56.71 | 4.69 | 30.39 | 42.00 | 5.10 | 53.74 | 0.27 | 19.70 | 92.80 | 18.48 | 25.14 | 0.53 | 14.72 |
| | FRONT | 95.25 | 63.19 | 49.45 | 57.46 | 56.70 | 92.70 | 12.99 | 32.89 | 19.19 | 21.69 | 86.90 | 19.95 | 32.21 | 41.97 | 31.38 |
| | TRUST-ALIGN (DPO) | 77.85 | 59.82 | 66.38 | 84.21 | 70.14 | 48.20 | 29.13 | 70.85 | 45.65 | 48.54 | 17.50 | 18.33 | 62.79 | 55.87 | 45.66 |
| LLaMA-3 -8b | ICL | 1.48 | 3.01 | 28.58 | 86.50 | 39.36 | 3.90 | 5.92 | 48.60 | 20.24 | 24.92 | 0.00 | 0.00 | 44.23 | 0.00 | 14.74 |
| | PostCite | 77.53 | 32.98 | 53.31 | 28.01 | 38.10 | 87.00 | 6.10 | 34.52 | 8.42 | 16.35 | 62.00 | 20.80 | 45.88 | 8.06 | 24.91 |
| | PostAttr | 77.53 | 32.98 | 53.31 | 5.95 | 30.75 | 87.00 | 6.10 | 34.52 | 1.46 | 14.09 | 62.00 | 20.80 | 45.88 | 1.25 | 22.64 |
| | FRONT | 99.05 | 62.25 | 41.62 | 66.14 | 56.67 | 100.00 | 13.53 | 22.78 | 20.42 | 18.91 | 99.50 | 18.99 | 17.85 | 44.69 | 27.18 |
| | TRUST-ALIGN (DPO) | 56.43 | 53.94 | 65.49 | 88.26 | 69.23 | 22.40 | 35.35 | 70.73 | 58.77 | 54.95 | 15.50 | 20.81 | 63.57 | 50.24 | 44.87 |

**TRUST-ALIGN boosts trustworthiness over baseline methods.** As shown in Table 2 and Table 3, TRUST-ALIGNed models demonstrate substantial improvements on TRUST-SCORE over the baselines in 26 out of 27 model family and dataset configurations. Specifically, with LLaMA-3-8b, TRUST-ALIGN outperforms FRONT by 12.56% (ASQA), 36.04% (QAMPARI), and 17.69% (ELI5) on TRUST-SCORE. This suggests that TRUST-ALIGNed models are more capable of generating responses grounded in the documents.

**TRUST-ALIGN improves models' refusal capability.** Across all 27 configurations, TRUST-ALIGN yields substantial improvements in F1$_{GR}$. In LLaMA-3-8b, TRUST-ALIGN outperforms FRONT by 23.87% (ASQA), 47.95% (QAMPARI), and 45.72% (ELI5). This indicates that TRUST-ALIGN substantially enhances models' ability to correctly refuse or provide answers.

**TRUST-ALIGN enhances models' citation quality.** F1$_{GC}$ is substantially improved over baselines in 24 out of 27 model family and dataset configurations after the application of TRUST-ALIGN. Specifically, with LLaMA-3-8b, TRUST-ALIGN outperforms FRONT on F1$_{GC}$ by 22.12% (ASQA), 38.35% (QAMPARI), and 5.55% (ELI5). This demonstrates that aligning with TRUST-ALIGN improves the model's ability to provide citations that sufficiently and precisely support claims.

**TRUST-ALIGN has mixed effects on F1$_{AC}$.** We observe that applying TRUST-ALIGN yields a notable increase in F1$_{AC}$ for QAMPARI (9/9) but mixed performance on ELI5 (5/9) and ASQA (2/9). The mixed performance in ASQA and ELI5 can be explained by the composition of F1$_{AC}$, which is derived from P$_{AC}$ and R$_{AC}$ (Eq. (9)).

Taking LLaMA-3.2-3b on ASQA as an example (Appendix I), TRUST-ALIGN models generally achieve higher P$_{AC}$ compared to baselines (54.63% for TRUST-ALIGN vs. 52.94% for FRONT) despite having a lower AR% (77.85% for TRUST-ALIGN vs. 95.25% for FRONT). This suggests that our models have a higher expected value for AC$^q$ (per-sample AC recall), as the denominator depends on the number of answered questions. This trend is observed across models and datasets.

However, in ASQA and ELI5, our models underperform in $F1_{AC}$ due to the overwhelmingly adverse impact of $R_{AC}$. The recall of answerable questions ($R_{ans}$) is lower for our model compared to baselines (89.02% for TRUST-ALIGN vs. 98.69% for FRONT), which rarely refuse questions. As a result, fewer terms are summed in the numerator of $R_{AC}$, while the denominator remains constant (the number of answerable questions). This leads to a lower overall $F1_{AC}$ score. To further analyze the baseline models' performance, we investigated how much of their answering ability relies on parametric knowledge versus document-based information (Section 6.1 and Appendix F.3).

Table 3: Qwen2.5 and Phi3.5 families evaluated on the three datasets.

| Model | Type | ASQA (610 answerable, 338 unanswerable) | | | | | QAMPARI (295 answerable, 705 unanswerable) | | | | | ELI5 (207 answerable, 793 unanswerable) | | | | |
| | | Resp. | Trustworthiness | | | | Resp. | Trustworthiness | | | | Resp. | Trustworthiness | | | |
| | | | Truthfullness | | Att-Grd. | | | Truthfullness | | Att-Grd. | | | Truthfullness | | Att-Grd. | |
| | | AR (%) | $F1_{AC}$ | $F1_{GR}$ | $F1_{GC}$ | TRUST | AR (%) | $F1_{AC}$ | $F1_{GR}$ | $F1_{GC}$ | TRUST | AR (%) | $F1_{AC}$ | $F1_{GR}$ | $F1_{GC}$ | TRUST |
| Qwen-2.5-0.5b | ICL | 29.85 | 20.96 | 47.19 | 0.35 | 22.83 | 11.40 | 2.45 | 50.67 | 0.00 | 17.71 | 82.30 | 13.73 | 33.14 | 0.37 | 15.75 |
| | PostCite | 46.10 | 8.55 | 50.84 | 8.23 | 22.54 | 17.00 | 0.67 | 52.51 | 5.72 | 19.63 | 89.80 | 9.87 | 27.10 | 4.10 | 13.69 |
| | PostAttr | 46.10 | 8.55 | 50.84 | 2.23 | 20.54 | 17.00 | 0.67 | 52.51 | 0.90 | 18.03 | 89.80 | 9.87 | 27.10 | 0.68 | 12.55 |
| | FRONT | 100.00 | 42.83 | 39.15 | 45.87 | 42.62 | 99.30 | 11.52 | 23.23 | 15.90 | 16.88 | 99.90 | 13.74 | 17.29 | 27.95 | 19.66 |
| | TRUST-ALIGN (DPO) | 71.84 | 50.59 | 61.28 | 52.40 | 54.76 | 17.90 | 15.76 | 61.84 | 29.73 | 35.78 | 21.70 | 13.68 | 60.79 | 22.72 | 32.40 |
| Qwen-2.5-1.5b | ICL | 98.52 | 50.55 | 41.74 | 6.69 | 32.99 | 85.00 | 15.60 | 41.27 | 8.61 | 21.83 | 99.40 | 20.56 | 17.78 | 4.99 | 14.44 |
| | PostCite | 71.73 | 16.36 | 52.46 | 15.40 | 28.07 | 11.20 | 3.44 | 51.11 | 13.95 | 22.83 | 91.50 | 15.63 | 26.71 | 5.17 | 15.84 |
| | PostAttr | 71.73 | 16.36 | 52.46 | 4.45 | 24.42 | 11.20 | 3.44 | 51.11 | 1.07 | 18.54 | 91.50 | 15.63 | 26.71 | 0.62 | 14.32 |
| | FRONT | 99.26 | 57.74 | 41.36 | 55.70 | 51.60 | 98.80 | 16.05 | 24.45 | 11.60 | 17.37 | 99.90 | 19.57 | 17.29 | 37.70 | 24.85 |
| | TRUST-ALIGN (DPO) | 72.57 | 52.68 | 62.38 | 66.81 | 60.62 | 20.00 | 23.80 | 68.46 | 50.98 | 47.75 | 33.60 | 19.03 | 57.91 | 31.63 | 36.19 |
| Qwen-2.5-3b | ICL | 27.43 | 37.72 | 51.36 | 51.72 | 46.93 | 22.30 | 23.17 | 63.27 | 41.20 | 42.55 | 68.80 | 29.12 | 46.31 | 34.34 | 36.59 |
| | PostCite | 8.76 | 9.58 | 35.30 | 10.94 | 18.61 | 0.10 | 0.00 | 41.31 | 0.00 | 13.77 | 49.70 | 21.73 | 48.49 | 7.56 | 25.93 |
| | PostAttr | 8.76 | 9.58 | 35.30 | 36.29 | 27.06 | 0.10 | 0.00 | 41.31 | 25.00 | 22.10 | 49.70 | 21.73 | 48.49 | 1.31 | 23.84 |
| | FRONT | 97.47 | 55.15 | 44.01 | 62.72 | 53.96 | 79.10 | 20.69 | 48.62 | 25.67 | 31.66 | 93.60 | 18.69 | 25.37 | 37.40 | 27.15 |
| | TRUST-ALIGN (DPO) | 49.47 | 55.19 | 63.76 | 78.64 | 65.86 | 48.10 | 35.69 | 70.31 | 45.64 | 50.55 | 13.50 | 22.52 | 64.38 | 42.01 | 42.97 |
| Qwen-2.5-7b | ICL | 92.09 | 58.94 | 54.34 | 75.46 | 62.91 | 56.30 | 28.92 | 63.67 | 39.28 | 43.96 | 82.70 | 28.27 | 37.13 | 44.13 | 36.51 |
| | PostCite | 91.46 | 27.52 | 45.93 | 4.19 | 25.88 | 26.70 | 8.59 | 60.16 | 1.05 | 23.27 | 95.60 | 21.82 | 22.23 | 7.03 | 17.03 |
| | PostAttr | 91.46 | 27.52 | 45.93 | 17.92 | 30.46 | 26.70 | 8.59 | 60.16 | 13.55 | 27.43 | 95.60 | 21.82 | 22.23 | 0.96 | 15.00 |
| | FRONT | 86.39 | 64.58 | 60.08 | 58.27 | 60.98 | 84.70 | 17.02 | 42.85 | 24.48 | 28.12 | 57.60 | 28.27 | 54.14 | 56.61 | 46.34 |
| | TRUST-ALIGN (DPO) | 59.49 | 55.04 | 66.22 | 83.57 | 68.28 | 32.10 | 30.11 | 70.68 | 53.48 | 51.42 | 21.00 | 24.30 | 63.79 | 47.02 | 45.04 |
| Phi3.5-mini | ICL | 63.19 | 50.24 | 51.95 | 42.64 | 48.28 | 70.20 | 11.91 | 43.90 | 12.26 | 22.69 | 81.50 | 27.59 | 37.17 | 30.14 | 31.63 |
| | PostCite | 23.10 | 14.98 | 41.38 | 9.40 | 21.92 | 76.90 | 3.57 | 42.36 | 4.49 | 16.81 | 84.50 | 20.50 | 30.81 | 4.67 | 18.66 |
| | PostAttr | 23.10 | 14.98 | 41.38 | 1.24 | 19.20 | 76.90 | 3.57 | 42.36 | 0.46 | 15.46 | 84.50 | 21.26 | 30.81 | 0.68 | 17.58 |
| | FRONT | 99.79 | 63.30 | 39.79 | 71.63 | 58.24 | 100.00 | 11.97 | 22.78 | 21.50 | 18.75 | 96.60 | 21.46 | 21.35 | 61.41 | 34.74 |
| | TRUST-ALIGN (DPO) | 66.56 | 52.23 | 64.20 | 85.36 | 67.26 | 30.10 | 36.42 | 73.95 | 53.40 | 54.59 | 24.90 | 23.39 | 67.62 | 47.42 | 46.14 |

**TRUST-ALIGN generalizes across model families and sizes.** Table 3 demonstrates that TRUST-ALIGN improves the models' TRUST-SCORE across various sizes and architectures. In small models like Qwen-2.5-0.5b, TRUST-ALIGN significantly outperforms ICL baselines, achieving notable gains in ASQA (22.83% → 54.76%). Similarly, for larger models such as Qwen-2.5-7b, TRUST-ALIGN delivers substantial improvements, as seen in ASQA (62.91% → 68.28%), highlighting its scalability. The largest gains are observed in smaller models; for example, Phi3.5-mini shows remarkable improvements over ICL: 18.98% (ASQA), 31.90% (QAMPARI), and 14.51% (ELI5).

**Models aligned with DPO generally outperform those trained with SFT.** Table 4 shows that DPO models outperform SFT models on TRUST-SCORE in 26 out of 27 model family and dataset configurations. In LLaMA-3.2-3b, DPO yields substantial improvements on ASQA (6.70%), QAMPARI (3.09%), and ELI5 (1.71%). Additionally, DPO models also attain substantially better $F1_{GC}$ compared to SFT on 25 out of 27 configurations, with substantial improvements on ASQA (8.58%), QAMPARI (7.62%), and ELI5 (2.54%) for LLaMA-3.2-3b. This highlights DPO's effectiveness in enhancing citation quality. While results on $F1_{AC}$ and $F1_{GR}$ are mixed, DPO yields better overall TRUST-SCORE scores.

## 6.1 ANALYSIS

**Data ablation.** Table 5 shows that adding samples targeting each of the five hallucination types improves TRUST-SCORE by 1.50% (ASQA), 1.78% (QAMPARI), and 2.23% (ELI5). We observe that removing data corresponding to each hallucination type causes a notable decrease in TRUST-SCORE, suggesting the importance of each subtype. In particular, removing refusal-related hallucinations adversely affects $F1_{GR}$: ↓2.79% (ASQA), ↓0.48% (QAMPARI), underscoring the importance of incorporating refusal-related data to improve a model's ability to discern when to provide an answer.

Table 4: Performance of models with only SFT applied as compared to TRUST-ALIGN models. Best values within each family are **bolded**).

| Model | Type | ASQA (610 answerable, 338 unanswerable) | | | | | QAMPARI (295 answerable, 705 unanswerable) | | | | | ELI5 (207 answerable, 793 unanswerable) | | | | |
|---|---|---|---|---|---|---|---|---|---|---|---|---|---|---|---|---|
| | | Resp. AR (%) | F1_AC | F1_GR | F1_GC | TRUST | Resp. AR (%) | F1_AC | F1_GR | F1_GC | TRUST | Resp. AR (%) | F1_AC | F1_GR | F1_GC | TRUST |
| LLaMA-2-7b | SFT | 80.17 | **53.21** | 63.43 | 79.61 | 65.42 | 31.60 | **33.76** | 71.13 | 46.37 | 50.42 | 29.50 | 21.58 | **63.30** | 39.59 | 41.49 |
| | TRUST-ALIGN (DPO) | 65.30 | 52.48 | **66.12** | **83.94** | **67.51** | 32.30 | 32.03 | **71.67** | **49.42** | **51.04** | 21.60 | **22.54** | 63.27 | **47.35** | **44.39** |
| LLaMA-3.2-1b | SFT | 63.82 | **45.61** | **63.91** | 73.10 | **60.87** | 26.00 | **27.98** | **68.20** | 37.96 | 44.71 | 20.50 | **14.56** | **63.93** | 37.28 | 38.59 |
| | TRUST-ALIGN (DPO) | 41.67 | 38.64 | 58.61 | **79.35** | 58.87 | 20.00 | 27.22 | 67.92 | **49.42** | **48.19** | 9.60 | 13.20 | 59.35 | **48.21** | **40.25** |
| LLaMA-3.2-3b | SFT | 68.04 | 49.23 | 65.47 | 75.63 | 63.44 | 27.60 | 28.09 | 70.22 | 38.03 | 45.45 | 14.70 | 15.92 | 62.59 | 53.33 | 43.95 |
| | TRUST-ALIGN (DPO) | 77.85 | **59.82** | **66.38** | **84.21** | **70.14** | 48.20 | **29.13** | **70.85** | **45.65** | **48.54** | 17.50 | **18.33** | **62.79** | **55.87** | **45.66** |
| LLaMA-3-8b | SFT | 68.99 | 52.35 | **66.06** | 80.95 | 66.45 | 24.20 | **33.85** | **71.11** | 48.01 | 50.99 | 23.60 | **22.57** | **65.06** | 46.85 | 44.83 |
| | TRUST-ALIGN (DPO) | 56.43 | **53.94** | 65.49 | **88.26** | **69.23** | 22.40 | 35.35 | 70.73 | **58.77** | **54.95** | 15.50 | 20.81 | 63.57 | **50.24** | **44.87** |
| Qwen-2.5-0.5b | SFT | 83.44 | 38.71 | 58.03 | **57.47** | 51.40 | 18.50 | **16.02** | 61.35 | 27.82 | 35.06 | 35.50 | 10.50 | 57.19 | 19.57 | 29.09 |
| | TRUST-ALIGN (DPO) | 71.84 | **50.59** | **61.28** | 52.40 | **54.76** | 17.90 | 15.76 | **61.84** | **29.73** | **35.78** | 21.70 | **13.68** | **60.79** | **22.72** | **32.40** |
| Qwen-2.5-1.5b | SFT | 78.27 | 44.23 | 58.75 | **71.08** | 58.02 | 25.50 | **23.89** | **69.66** | 37.68 | 43.74 | 41.30 | 14.14 | 55.35 | 27.69 | 32.39 |
| | TRUST-ALIGN (DPO) | 72.57 | **52.68** | **62.38** | 66.81 | **60.62** | 20.00 | 23.80 | 68.46 | **50.98** | **47.75** | 33.60 | **19.03** | **57.91** | **31.63** | **36.19** |
| Qwen-2.5-3b | SFT | 75.21 | 47.26 | 60.61 | 73.09 | 60.32 | 27.20 | 28.80 | 68.12 | 37.34 | 44.75 | 34.50 | 14.85 | 61.47 | 35.87 | 37.40 |
| | TRUST-ALIGN (DPO) | 49.47 | **55.19** | **63.76** | **78.64** | **65.86** | 48.10 | **35.69** | **70.31** | **45.64** | **50.55** | 13.50 | **22.52** | **64.38** | **42.01** | **42.97** |
| Qwen-2.5-7b | SFT | 65.30 | 50.73 | 64.50 | 82.07 | 65.77 | 31.70 | **33.58** | 70.10 | 49.08 | 50.92 | 25.50 | 20.78 | **64.25** | 46.89 | 43.97 |
| | TRUST-ALIGN (DPO) | 59.49 | **55.04** | **66.22** | **83.57** | **68.28** | 32.10 | 30.11 | **70.68** | **53.48** | **51.42** | 21.00 | **24.30** | 63.79 | **47.02** | **45.04** |
| Phi3.5-mini | SFT | 66.46 | 51.92 | **64.34** | 82.77 | 66.34 | 29.10 | 35.04 | 73.93 | 49.38 | 52.78 | 24.50 | 22.50 | 65.70 | 46.79 | 45.00 |
| | TRUST-ALIGN (DPO) | 66.56 | **52.23** | 64.20 | **85.36** | **67.26** | 30.10 | **36.42** | **73.95** | **53.40** | **54.59** | 24.90 | **23.39** | **67.62** | **47.42** | **46.14** |

Table 5: Ablations of data synthesis techniques for LLaMA-2-7b on three evaluation datasets using refusal prompting; The original error types in Section 2.3 were summarized into three main classes: answer-related (Inaccurate Answer), citation-related (Overcitation, Improper Citation), refusal-related (Over Responsiveness, Excessive Refusal).

| | ASQA | | | | | QAMPARI | | | | | ELI5 | | | | |
|---|---|---|---|---|---|---|---|---|---|---|---|---|---|---|---|
| | Resp. AR (%) | F1_AC | F1_GR | F1_GC | TRUST | Resp. AR (%) | F1_AC | F1_GR | F1_GC | TRUST | Resp. AR (%) | F1_AC | F1_GR | F1_GC | TRUST |
| **DPO-LLaMA-2-7b** | 65.30 | 52.48 | 66.12 | 83.94 | **67.51** | 31.10 | 32.09 | 71.83 | 51.33 | **51.75** | 21.60 | 22.54 | 63.27 | 48.43 | **44.75** |
| TRUST-ALIGN w/o. augmented instructions | 79.43 | 53.54 | 63.33 | 81.15 | 66.01 | 32.20 | 33.14 | 70.82 | 45.94 | 49.97 | 29.50 | 23.98 | 63.30 | 40.28 | 42.52 |
| TRUST-ALIGN w/o. answer HT | 77.74 | 53.29 | 63.7 | 81.2 | 66.06 | 33.40 | 33.56 | 71.36 | 46.17 | 50.36 | 27.60 | 23.47 | 63.56 | 38.28 | 41.77 |
| TRUST-ALIGN w/o. citation HT | 77.32 | 52.55 | 63.88 | 81.51 | 65.98 | 33.10 | 34.13 | 71.40 | 46.91 | 50.81 | 26.70 | 22.65 | 64.33 | 42.81 | 43.26 |
| TRUST-ALIGN w/o. refusal HT | 79.11 | 53.55 | 63.33 | 81.85 | 66.24 | 31.10 | 34.40 | 71.35 | 48.12 | 51.29 | 28.30 | 22.93 | 64.05 | 41.18 | 42.72 |
| GPT-4 as critic | 70.36 | 54.91 | 65.29 | 78.47 | 66.22 | 25.90 | 30.77 | 70.29 | 48.87 | 49.98 | 23.50 | 17.27 | 62.24 | 42.38 | 40.63 |

We validated our data construction approach against the GPT-4-as-critic pipeline (Li et al., 2024a; Huang et al., 2024b), where GPT-4 iteratively identifies and corrects errors to generate positive and negative responses (details in Appendix G). In LLaMA-2-7b, TRUST-ALIGN outperforms GPT-4 critic on TRUST-SCORE, with gains of 1.29% (ASQA), 1.77% (QAMPARI), and 4.12% (ELI5).

**Importance of refusal samples in TRUST-ALIGN.** To verify the importance of refusal samples in our pipeline, we removed all unanswerable questions from the training set, creating a dataset without refusals. Table 6 shows a significant drop in TRUST-SCORE scores without refusals, including declines of 10.2% (LLaMA-3-8b) and

Table 6: Effect of adding refusal samples on the ASQA.

| TRUST-ALIGN Models | | AR% | F1_AC | F1_GR | F1_GC | TRUST |
|---|---|---|---|---|---|---|
| Only Answerable | DPO-LLaMA-2-7b | 100 | 51.79 | 39.15 | 77.37 | 56.10 |
| | DPO-LLaMA-3-8b | 100 | 56.54 | 39.15 | 81.39 | 59.03 |
| With Refusal | DPO-LLaMA-2-7b | 65.30 | 52.48 | 66.12 | 83.94 | 67.51 |
| | DPO-LLaMA-3-8b | 56.43 | 53.94 | 65.49 | 88.26 | 69.23 |

11.41% (LLaMA-2-7b). Notably, $F1_{GR}$ decreases by 26.34% (LLaMA-3-8b) and 26.97% (LLaMA-2-7b), and $F1_{GC}$ by 6.87% (LLaMA-3-8b) and 6.57% (LLaMA-2-7b).

We also observe that in LLaMA-3-8b, $F1_{AC}$ is higher in the answerable-only setting compared to with refusals setting. This occurs because $R_{AC}$ favors over-responsive models, which artificially inflates $F1_{AC}$, as discussed in main results. The resulting models answer all questions (AR% of 100%), even without supporting documents, suggesting an increased reliance on ungrounded parametric knowledge, as discussed in Section 6.1.

**Out-of-domain analysis.** Following Huang et al. (2024a), we use ExpertQA (Malaviya et al., 2024) to assess our model's generalizability. As shown in Table 7, TRUST-ALIGN model outperforms FRONT on TRUST-SCORE across all 27 open-source model family and dataset configurations. We also observe that the open-source ICL models perform significantly worse on TRUST-SCORE as compared to the closed-source ICL models, with a 9.79% gap between LLaMA-3-8b and GPT-4. TRUST-ALIGN not only closes this gap but establishes a lead: TRUST-ALIGNed LLaMA-3-8b achieves the highest TRUST score of 54.85%, surpassing 54.69% of GPT-4.

Table 7: Generalization test results on ExpertQA using refusal prompting.

| Model | Type | AR (%) | F1$_{AC}$ | F1$_{GR}$ | F1$_{GC}$ | TRUST |
|---|---|---|---|---|---|---|
| LLaMA-2-7b | ICL | 0.51 | 0.00 | 41.01 | 9.52 | 16.84 |
| | PostCite | 5.62 | 4.85 | 44.27 | 5.23 | 18.12 |
| | PostAttr | 5.62 | 4.85 | 44.27 | 2.26 | 17.13 |
| | FRONT | 100 | 9.33 | 23.92 | 74.75 | 36.00 |
| | TRUST-ALIGN (DPO) | 20.01 | 25.03 | 67.91 | 62.46 | **51.8** |
| LLaMA-3.2-1b | ICL | 90 | 21.55 | 32.83 | 9.04 | 21.14 |
| | PostCite | 30.84 | 5.48 | 49.1 | 2.67 | 19.08 |
| | PostAttr | 48.41 | 8.24 | 47.72 | 1.5 | 19.15 |
| | FRONT | 95.62 | 20.83 | 29.26 | 37.45 | 29.18 |
| | TRUST-ALIGN (DPO) | 15.44 | 20.32 | 64.87 | 62.1 | **49.1** |
| LLaMA-3.2-3b | ICL | 58.74 | 33.5 | 51.21 | 38.37 | 41.03 |
| | PostCite | 82.85 | 25.68 | 38.11 | 5.29 | 23.03 |
| | PostAttr | 82.85 | 25.45 | 38.58 | 3.4 | 22.48 |
| | FRONT | 83.36 | 27.24 | 43.34 | 50.91 | 40.5 |
| | TRUST-ALIGN (DPO) | 7.24 | 11.72 | 56.93 | 78.35 | **49.0** |
| LLaMA-3-8b | ICL | 0.65 | 2.82 | 42.5 | 69.46 | 38.26 |
| | PostCite | 15.68 | 14.06 | 50.08 | 7.09 | 23.74 |
| | PostAttr | 15.68 | 14.06 | 50.08 | 6.29 | 23.47 |
| | FRONT | 99.26 | 30.34 | 24.92 | 56.7 | 37.32 |
| | TRUST-ALIGN (DPO) | 16.41 | 27.36 | 67.07 | 70.11 | **54.85** |
| GPT-3.5 | ICL | 59.47 | 36.65 | 56.39 | 63.93 | 52.32 |
| GPT-4 | ICL | 72.20 | 41.32 | 52.91 | 69.83 | **54.69** |
| GPT-4o | ICL | 66.07 | 42.62 | 64.4 | 54.61 | 51.24 |
| | TRUST-ALIGN (SFT) | 36.84 | 28.85 | 71.68 | 61.98 | **53.82** |
| Claude-3.5 | ICL | 73.95 | 11.68 | 51.91 | 10.7 | 24.76 |

| Model | Type | AR (%) | F1$_{AC}$ | F1$_{GR}$ | F1$_{GC}$ | TRUST |
|---|---|---|---|---|---|---|
| Qwen-2.5-0.5b | ICL | 78.24 | 21.42 | 38.71 | 0.44 | 20.19 |
| | PostCite | 51.41 | 13.32 | 48.08 | 5.6 | 22.33 |
| | PostAttr | 51.41 | 13.32 | 48.08 | 1.49 | 20.96 |
| | FRONT | 99.86 | 18.27 | 24.05 | 34.62 | 25.65 |
| | TRUST-ALIGN (DPO) | 32.96 | 18.16 | 63.31 | 35.07 | **38.85** |
| Qwen-2.5-1.5b | ICL | 98.34 | 30.67 | 26.09 | 6.89 | 21.22 |
| | PostCite | 62.19 | 22.22 | 48.66 | 16.92 | 29.27 |
| | PostAttr | 62.19 | 22.22 | 48.66 | 13.15 | 28.01 |
| | FRONT | 99.59 | 29.15 | 24.6 | 50.22 | 34.66 |
| | TRUST-ALIGN (DPO) | 30.2 | 25.06 | 68.38 | 51.44 | **48.29** |
| Qwen-2.5-3b | ICL | 68.88 | 35.14 | 49.65 | 42.67 | 42.49 |
| | PostCite | 0.05 | 0 | 40.66 | 0 | 13.55 |
| | PostAttr | 0.05 | 0 | 40.66 | 0 | 13.55 |
| | FRONT | 95.48 | 25.67 | 29.86 | 44.48 | 33.34 |
| | TRUST-ALIGN (DPO) | 17.15 | 20.97 | 65.79 | 60.25 | **49.0** |
| Qwen-2.5-7b | ICL | 84.56 | 36.33 | 42.28 | 56.09 | 44.9 |
| | PostCite | 42.14 | 25.58 | 54.9 | 13.77 | 31.42 |
| | PostAttr | 42.14 | 25.58 | 54.9 | 12.46 | 30.98 |
| | FRONT | 65.51 | 32.41 | 55.56 | 67.35 | 51.77 |
| | TRUST-ALIGN (DPO) | 24.99 | 25.57 | 69.16 | 62.7 | **52.48** |
| Phi3.5-mini | ICL | 85.15 | 37.49 | 40.22 | 36.14 | 37.95 |
| | PostCite | 52.01 | 27.96 | 53.64 | 7.39 | 29.66 |
| | PostAttr | 52.01 | 27.96 | 53.64 | 5.7 | 29.1 |
| | FRONT | 97.37 | 28.19 | 27.5 | 65.82 | 40.5 |
| | TRUST-ALIGN (DPO) | 26.05 | 27.69 | 69.56 | 61.6 | **52.95** |

In LLaMA-3-8B, TRUST-ALIGN outperforms ICL on F1$_{GR}$ by 16.59% and substantially outperforms GPT-3.5 and Claude 3.5 in both F1$_{GC}$ and F1$_{GR}$. Although GPT-3.5 and GPT-4 achieve higher F1$_{AC}$ scores, indicating better answer coverage, they rely heavily on parametric knowledge (Section 6.1 and Appendix F.3). This leads to less grounded and less trustworthy responses, as reflected in lower TRUST-SCORE scores compared to TRUST-ALIGN. Similar trends are observed in other model families.

**Studying parametric knowledge access.** For an LLM-in-RAG task, it is important to study the tendency of LLM towards grounding its knowledge on the provided documents. To partially quantify this, we compute the answer correctness score for questions that are unanswerable by the provided documents (defined as S$_{param}$); thus a fraction of cases where $A_G \cap A_D = \emptyset$ but $A_G \neq \emptyset$ (more details on the metric in Appendix F.2). In Table 10, our analysis reveals that responsive models (high AR%) tend to rely on parametric knowledge more frequently (high S$_{param}$). Notably, closed-source models like GPT-4 exhibit higher parametric knowledge usage compared to open-source and TRUST-ALIGN models. However, S$_{param}$ only partially captures the models' utilization of parametric knowledge. For instance, it does not account for cases where the document contains the answer, and the model still relies on parametric knowledge to generate the correct answer (also present in the document). This phenomenon is evident in Table 12, where on ASQA, GPT-4 achieves a significantly higher F1$_{AC}$ than our models, yet its attribution groundedness score F1$_{GC}$ is five points lower.

## 7 CONCLUSION

In this study, we introduced a new holistic metric to evaluate the suitability of LLMs for RAG applications, where they are expected to ground their responses in the provided documents. We proposed TRUST-SCORE, which comprehensively measures the quality of answers, citations, and refusal performance of an LLM. Additionally, we presented TRUST-ALIGN, a method that uses a constructed dataset to align models for improved TRUST-SCORE performance. By applying Direct Preference Optimization (DPO) techniques, we trained LLaMA-2-7b and LLaMA-3-8b on this dataset, significantly reducing hallucinations in an RAG environment. Our approach, TRUST-ALIGN, demonstrates performance comparable to major closed-source models like GPT-4.

## ACKNOWLEDGEMENT

This research/project is supported by the National Research Foundation, Singapore under its AI Singapore Programme (AISG Award No: AISG3-GV-2023-010). This work is also supported by the Microsoft Research Accelerate Foundation Models Academic Research program.

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

# Table of Contents

## A    NUANCES OF ANSWERABILITY

Determining answerability can be challenging. To determine answerability, we use a system that evaluates the entailment of gold claims against provided documents, referred to as the Natural Language Inference (NLI) system. An NLI system can range from a simple exact match (EM) identifier to an LLM or even a human evaluator, with answerability determined based on $q, D$ and biases of the NLI[9]. These biases can be useful in specific RAG applications, such as solving mathematical problems where the documents provide a formula and the question assigns values to variables. The choice of NLI depends on whether the RAG system requires the LLM to have mathematical understanding. **Ideally, to prevent improper evaluations, the NLI model used to construct the gold claims should also be used to evaluate the LLM responses.**

In this paper, our focus is on evaluating the generic comprehension capabilities of LLMs without specialized knowledge. Thus, we use two NLI mechanisms: 1) identifying whether an exact match of claims is present in the gold claims, and 2) using a Machine Learning (ML) model to determine if the documents can entail the gold claims. The ML-based NLI model is used for multiple purposes, such as alignment dataset construction (data/training) and evaluating generated responses (metric/testing). For this, we adopt the NLI model from Rashkin et al. (2022). $\phi(c_{ij}, s_i) = 1$ if $c_{ij}$ (premise) entails $s_i$ (hypothesis); otherwise, 0. To determine answerability, we employ the TRUE-based method (Honovich et al., 2022) to assess whether a gold claim can be entailed by a given document.

**The knowledge grounding problem.**    Typically, LLMs are designed to perform question-answering tasks, where response generation heavily relies on the parametric (internal) knowledge acquired during their pre-training, tuning, and alignment phases (OpenAI, 2023; Anthropic, 2024). Thus, most of their knowledge is grounded in parametric memory. This makes them inherently less suitable for RAG applications, where the knowledge generated by the LLM is expected to be grounded in input documents. RAG is analogous to a reading comprehension task, where the answers must come from the provided passage (documents in RAG) rather than the prior knowledge of the person taking the test. Thus, any reliance on parametric knowledge can result in statements that are not fully grounded in the documents, including providing answers to unanswerable questions. *Our investigation shows that state-of-the-art models, such as GPT-4 and Claude-3.5-Sonnet, overtly rely on parametric knowledge even when used in a RAG setting.*[10]

## B    ANSWERABILITY: A CASE STUDY

Prior works (Liu et al., 2023; Gao et al., 2023b; Ye et al., 2024; Huang et al., 2024a; Li et al., 2024a) have employed substring matching to indicate entailment. While this syntactic approach is fast, it often proves inadequate in complex, long contexts. A case study is presented in Table 8. To address the limitations of this superficial entailment, we adopt a TRUE-based method (Honovich et al., 2022), which combines the strengths of both syntactic and semantic approaches. Specifically, we enhance the process by using the TRUE model, a T5-11B model (Raffel et al., 2020) fine-tuned for the NLI task, to verify, from a semantic perspective, whether a substring match corresponds to meaningful entailment within document passages. The input to the TRUE model is the concatenation of a premise and a hypothesis, and the output is an entailment score between 0 and 1, indicating the degree to which the premise entails the hypothesis. We treat the corresponding documents as the premise, and to minimize ambiguity, the associated question is concatenated with each gold answer as the hypothesis. In cases where the TRUE model does not yield a positive entailment score despite a substring match, we rely on the TRUE judgment as the final label. However, if the substring match fails, we bypass TRUE calculation, thus reducing the computational cost of relying solely on TRUE for semantic entailment.

---

[9]For EM, the bias is that a $q$ is answerable if an exact match for claims is present in $D$.

[10]We show a detailed analysis in Appendices F.2 and F.3.

Table 8: Case study showcasing the limitations of substring matching and necessity of TRUE judgement.

| Question | How many state parks are there in Virginia? |
|---|---|
| Gold Answer | 38 |
| Retrieved document | Virginia has 30 National Park Service units, such as Great Falls Park and the Appalachian Trail, and one national park, the Shenandoah National Park. With over 500 miles of trails, including **38** miles of the iconic Appalachian Trail, it's a paradise for hikers, nature lovers, and those seeking serene mountain landscapes. |
| Substring match | Substring is matched and as such the question is answerable. |
| TRUE Judgement | Not entailed as such the question is unanswerable given the document. |

## C  RELATED WORKS

### C.1  ATTRIBUTABLE RETRIEVAL AUGMENTED GENERATION

Retrieval Augmented Generation (RAG) has been widely studied for reducing the knowledge gap and providing more referenced information to enhance answer generation (Karpukhin et al., 2020; Lewis et al., 2021; Gao et al., 2023c). However, LLMs are prone to being misled by irrelevant information, leading to hallucinations and less factual outputs (Shi et al., 2023; Yoran et al., 2024; Xu et al., 2023). This challenge has spurred research into attributable RAG, which aims to verify model outputs by identifying supporting sources. Rashkin et al. (2022) first introduced the concept of Attributable to Identified Sources (AIS) to evaluate attribution abilities. Subsequently, Gao et al. (2023b) adapted this approach to verify generated content with citations, improving the reliability of RAG systems. Simultaneously, Press et al. (2024) and Song et al. (2024) explored related aspects: citation attribution for paper identification and the verifiability of long-form generated text, respectively. Further fine-grained evaluations have been examined, such as assessing the degree of support (Zhang et al., 2024b) and the granularity of claims (Xu et al., 2024). Recent studies (Buchmann et al., 2024; Hsu et al., 2024) have also investigated attribution ability by disentangling the confounding effects of retrievers and LLMs. Unlike existing works, we design TRUST-SCORE to prioritize trustworthiness in LLMs by ensuring that generated responses are strictly grounded in the provided documents, thereby minimizing the generation of unverifiable content. This focus on verifiable accuracy strengthens the reliability of LLM outputs and enhances user trust.

### C.2  ENHANCE GROUNDED TEXT GENERATION IN ATTRIBUTED LARGE LANGUAGE MODELS

To enhance grounded text generation, various attributed LLMs have been proposed, falling into two main paradigms: training-free and training-based. For training-free methods: 1) In-context learning (Gao et al., 2023b) is used to generate in-line citations with few-shot demonstrations. 2) Post-hoc attribution (Gao et al., 2023a; Li et al., 2024b) first generates an initial response and then retrieves evidence as attribution. 3) Ji et al. (2024) demonstrate that using chain-of-thought reasoning improves the quality of text generated with citations. For training-based methods: 1) Asai et al. (2024); Slobodkin et al. (2024); Xia et al. (2024); Ye et al. (2024) apply supervised fine-tuning (SFT) to LLMs, training them to identify useful information from documents and guide cited text generation with them. 2) Beyond simple SFT, recent studies model the task as preference learning, employing Reinforcement Learning with Human Feedback (RLHF) (Ouyang et al., 2022) and Direct Preference Optimization (DPO) (Rafailov et al., 2024a). Huang et al. (2024a) proposed a method to improve attribution generation using fine-grained rewards and Proximal Policy Optimization (PPO) (Schulman et al., 2017), while Li et al. (2024a); Huang et al. (2024b) introduced the modified DPO framework to enhance fine-grained attribution abilities. 3) While many approaches rely on external documents provided by the user or retrieved during generation, Khalifa et al. (2024); Zhang et al. (2024a) focus on tuning LLMs to cite sources from pre-training data using learned parametric knowledge. In contrast to previous approaches, we introduce TRUST-ALIGN, which advances alignment data generation through a multi-step process that disentangles answer generation from citation quality. This separation enables TRUST-ALIGN to simultaneously improve the quality of answer generation, citation accuracy, and refusal precision. Additionally, TRUST-ALIGN addresses a broader range of

hallucination errors, including inappropriate refusals, thereby enhancing the overall trustworthiness and reliability of the model's outputs.

# D  METRICS

In this section, we elaborate on how we compute metrics that are components of TRUST-SCORE.

## D.1  RESPONSE TRUTHFULNESS

Truthfulness captures the model's ability to answer or refuse a question correctly by computing the grounded refusal ($F1_{GR}$) and the factual accuracy by computing the answer-calibrated answer correctness score ($F1_{AC}$).

**Grounded Refusal [$F1_{GR}$]:**   A macro-averaged F1 score that measures the LLM's ability in correctly refusing to answer a question ($F1_{ref}$) and correctly providing an answer when required ($F1_{ans}$).

- **$F1_{ref}$**: This metric evaluates a model's ability to correctly refuse unanswerable questions. We calculate it based on how accurately the model identifies and refuses these questions. Let $A_g$ and $\neg A_g$ represent the sets of ground truth answerable and unanswerable questions, respectively, and $A_r$ and $\neg A_r$ denote the sets of questions where the model provided an answer and refused to answer, respectively. $F1_{ref}$ is computed from precision $P_{ref}$ and recall $R_{ref}$:

$$P_{ref} = \frac{|\neg A_r \cap \neg A_g|}{|\neg A_r|} \tag{1}$$

$$R_{ref} = \frac{|\neg A_r \cap \neg A_g|}{|\neg A_g|} \tag{2}$$

$$F1_{ref} = \frac{2P_{ref} \cdot R_{ref}}{P_{ref} + R_{ref}}, \tag{3}$$

  where $P_{ref}$ measures the proportion of correctly refused unanswerable questions among all refused questions, and $R_{ref}$ measures the proportion of correctly refused unanswerable questions out of all unanswerable questions. Here, $|\cdot|$ denote the cardinality of the set, thus $P_{ref}$, $R_{ref}$, and $F1_{ref}$ are scalar values.

- **$F1_{ans}$**: This metric evaluates a model's ability to correctly answer answerable questions. It is computed based on the precision $P_{ans}$ and recall $R_{ans}$ for non-refusal responses to answerable questions:

$$P_{ans} = \frac{|A_r \cap A_g|}{|A_r|} \tag{4}$$

$$R_{ans} = \frac{|A_r \cap A_g|}{|A_g|} \tag{5}$$

$$F1_{ans} = \frac{2P_{ans} \cdot R_{ans}}{P_{ans} + R_{ans}} \tag{6}$$

**$F1_{GR}$ (Grounded Refusals)** provides an overall assessment of the model's refusal capabilities by computing the macro-average of $F1_{ref}$ and $F1_{ans}$:

$$F1_{GR} = \frac{1}{2}(F1_{ref} + F1_{ans}) \tag{7}$$

$F1_{ref}$ evaluates the model's ability to correctly refuse unanswerable questions, while $F1_{ans}$ assesses its ability to correctly answer answerable ones. By penalizing both incorrect refusals and incorrect non-refusals, $F1_{GR}$ offers a balanced evaluation of the model's over-responsiveness and under-responsiveness

**Answer Correctness (Answer Calibrated) [F1$_{AC}$]:** Given a question $q$ and the corresponding gold claims $A_G = \{a_{g1}, \ldots, a_{gn}\}$, we define the claims obtainable from the provided documents as $A_D = \{a_{d1}, \ldots, a_{dn}\}$ and the claims generated in the response $r$ as $A_R = \{a_{r1}, \ldots, a_{rn}\}$. $AC^q$ disregards the claims that cannot be inferred from $D$ (answer calibration), and the exact match recall scores is computed on the remaining claims, i.e., $A_G \cap A_D$:

$$\text{AC}^{qi} = \frac{|A_G \cap A_D \cap A_R|}{|A_G \cap A_D|} \tag{8}$$

For the whole dataset with multiple questions $\{q_1 \ldots q_k\}$, one can compute the average:

$$\text{AC} = \frac{1}{k} \sum_{q_i \in A_g \cap A_r} \text{AC}^{q_i} \tag{9}$$

Where $A_g$ denote the set of questions that are answerable using the provided documents, fully or partially; $A_r$ denote the set of questions that are answered by the model (non-refusal). There are two variants of AC we study— precision-oriented **P$_{AC}$** with denominator $k = |A_r|$ (number of answered questions). Second variant, recall-oriented **R$_{AC}$** with denominator $k = |A_g|$ (number of answerable questions). Here $|\cdot|$ denotes the cardinality of the set. We denote the aggregated score by

$$\text{F1}_{AC} = \frac{2\,\text{P}_{AC}\,\cdot\,\text{R}_{AC}}{\text{P}_{AC} + \text{R}_{AC}}. \tag{10}$$

The primary reason for adjusting the conventional Answer Correctness (AC) metric to account for the presence of answers in retrieved documents is to avoid rewarding models for generating correct answers without locating them in the provided documents. This approach discourages models from relying solely on their pre-trained knowledge to answer questions, instead encouraging them to find and ground their answers within the provided documents.

### D.2 ATTRIBUTION GROUNDEDNESS

Attribution or citation groundedness measures the relevance of generated citations to their corresponding statements, both individually and collectively. A citation $c_{i,j}$ is deemed "*relevant*" when the statement it cites can be inferred from the cited document. The collective importance of citations is assessed using a statement-wise recall metric, while the individual importance of each citation is evaluated using a precision metric. Given that a generated response $r$ consists of multiple statements $\mathcal{S}$ and their corresponding citations $\mathcal{C}$, we first compute statement-wise citation recall and per-citation precision. These scores are then averaged to obtain sample-wise scores, which are finally averaged to produce dataset-wide scores.

**Grounded Citation F1 [F1$_{GC}$]:** For a given statement $s_i$, statement-wise citation recall is computed by:

$$\text{R}_{\text{cite}}^{s_i} = \phi(\{c_{i,1}, \ldots, c_{i,j}\}, s_i) \tag{11}$$

where $\phi(\{c_{i,1}, \ldots, c_{i,j}\}, s_i) \rightarrow \{0, 1\}$ is a function that determines whether the concatenation of all cited documents fully supports the statement $s_i$. Next, we compute precision for a generated citation $c_{i,j}$ for statement $s_i$ as:

$$\text{P}_{\text{cite}}^{c_j} = \phi(c_{i,j}, s_i) \tag{12}$$
$$\text{OR} \quad \neg\phi(\{c_{i,k} \mid k \neq j\}, s_i)$$

Thus, citation precision is 0 if and only if the cited document $c_{i,j}$ does not entail the statement $s_i$, while all other citations collectively entail $s_i$ without $c_{i,j}$.

As an aggregate measure, we report **F1$_{GC}$**, which computes the F1 score using cumulative precision and recall over the answered questions only (non-refusals):

$$R_{cite} = \frac{1}{|A_r|} \sum_{S \in A_r^s} \frac{1}{|S|} \sum_{s_i \in S} R_{cite}^{s_i} \tag{13}$$

$$P_{cite} = \frac{1}{|A_r|} \sum_{C \in A_r^c} \frac{1}{|C|} \sum_{c_j \in C} P_{cite}^{c_j} \tag{14}$$

$$F1_{GC} = \frac{2 \cdot P_{cite} \cdot R_{cite}}{P_{cite} + R_{cite}} \tag{15}$$

Where $A_r$ denotes the number of samples answered by the model, $S$ denotes the set of statements in a generated response, and $A_r^s$ denotes the set of responses (including only statements, ignoring citations) in the dataset. Similarly, $C$ denotes the set of citations in a generated response, and $A_r^c$ denotes the set of responses (including only citations, ignoring statements) in the dataset.

**TRUST-SCORE:** Finally, we combine the metrics to produce a single trustworthiness score, which allows us to rank models based on their trustworthiness. This score is calculated as the average of each component metric.

$$\text{TRUST-SCORE} = \frac{1}{3}(\textbf{F1}_{GR} + \textbf{F1}_{AC} + \textbf{F1}_{GC}) \tag{16}$$

## E  THE TRUST-ALIGN DATASET

To align LLMs towards trustworthiness, we propose a new approach, **TRUST-ALIGN**. The approach constructs an LLM trustworthiness alignment dataset, where each sample in the dataset consists of a question $q$, a set of retrieved documents $D$, and a pair of positive (preferred) and negative (unpreferred) responses $(r^+, r^-)$. The positive response corresponds to an answer that encompasses expected gold claims for $q$ and corresponding citations referring to the documents. If $D$ is not sufficient to answer $q$, $r^+$ is assigned a refusal response, while $r^-$ is its non-refusal counterpart. We build the dataset in multiple steps: 1) Obtain a set of high-quality and diverse questions, 2) Obtain documents for each question, 3) Augmenting $(q, D)$ pairs that cover diverse hallucination types, 4) Construct positive responses entailing gold claims, and 5) Construct negative (unpreferred) responses by prompting a fine-tuned model and observing its hallucinations.

### E.1  COLLECTING QUALITY QUESTIONS

The dataset construction process begins with gathering a diverse set of high-quality, challenging questions from the training splits of source datasets, including ASQA, QAMPARI, and ELI5. To collect **seed samples**, we first divide the questions in a dataset into $k$ clusters using a Hugginface pipeline[11]. After identifying the diverse clusters, we use Mixtral-8x7B with the prompt described in Table 24 to assign each a quality score ranging from 1 to 7. The quality of a cluster is determined by how difficult it is to answer the questions without requiring additional information i.e. a higher score corresponds to a high difficulty. We then select clusters with a quality score of 4 or higher and sample the desired number of questions from these top clusters. Suppose we have three clusters, $C_1, C_2, C_3$, with respective sizes $N_1, N_2, N_3$, where $N_c = N_1 + N_2 + N_3$. To sample $N_s$ questions from the clusters, we sample $N_s \times \frac{C_i}{N_c}$ questions from cluster $C_i$. If this number exceeds the available questions in the cluster, we randomly sample the remaining questions from the filtered-out clusters (those with a quality score below 4). This process ensures that the seed set prioritizes both high quality and diversity. For this paper, we set $N_s$ to 3K, 3K, and 4K for ASQA, QAMPARI, and ELI5, respectively, resulting in approximately 10K questions in the seed set.

### E.2  COLLECTING $D$'S

For each seed question $q$ that is obtained from ASQA and QAMPARI, we used `gtr-t5-xxl` (Ni et al., 2022) to retrieve the top 100 relevant documents $D$ from the 2018-12-20 Wikipedia snapshot.

---

[11]`https://github.com/huggingface/text-clustering/`

For the ELI5 dataset, we employed BM25 in conjunction with Sphere (Piktus et al., 2021), a filtered version of Common Crawl, as it better encompasses the wide range of topics present in ELI5. We filter seed questions for which the retriever fails to retrieve relevant documents.

We utilize TRUE-NLI to derive the entailment pattern for each document. This pattern represents the set of gold claims that the document supports. The TRUE model takes as input a concatenation of a premise and a hypothesis, producing an entailment score (0 or 1) that indicates whether the premise entails the hypothesis. In our approach, the documents serve as the premise, while the hypothesis is formed by combining the relevant question with each corresponding gold claim to reduce ambiguity. We take the union of the entailment patterns across documents to assess the answerability of each question—if the pattern contains at least one supporting claim, the question is considered answerable.

Following Gao et al. (2023b), we identify 5 documents that are equally effective for the model as the 100 documents in terms of achieving the Exact Match (EM) recall value; we refer to such documents as *oracle* documents for question $q$. Notably, to compute EM, gold claims are obtained from respective source datasets.

### E.3    AUGMENTING *(q,D)* SET

Now that we have the questions and the most relevant (oracle) documents, our goal is to create samples of diverse types (i.e., different proportions of relevant documents for the same question) that can trigger multiple hallucinations from LLMs (Section 2.3). As illustrated in Fig. 3, for answerable questions, we first utilize the identified entailment patterns to generate all possible combinations of documents, then select $k$ combinations that cover diverse patterns. To create samples with unanswerable questions, we select documents that are similar to gold-claim-entailing documents but do not entail any gold claims. To minimize the risk of introducing bias in citation indices, we shuffle the order of documents in each sample. As a result, we generate approximately 70K question-document pairs.

After obtaining $(q, D)$ pairs for the alignment dataset, we obtain positive and negative responses $(r^+, r^-)$ for each pair—an essential component of the dataset signaling the model's preferred and unpreferred responses. To achieve this, we introduce a response generation pipeline.

### E.4    OBTAINING $\mathbf{r}^+$

We develop an automated data labeling pipeline that synthesizes natural responses from gold claims and maps each statement to the corresponding documents for embedded in-line citations. The gold claims are obtained from the source datasets (ASQA, QAMPARI, ELI5) and calibrated to the provided documents, i.e., filtering out claims that cannot be derived from $D$. We first split the questions into answerable and unanswerable samples based on whether the provided documents entail the gold claims. For an answerable sample, consisting of a question $q$, a set of documents $D$, and a list of (calibrated) gold claims, we prompt GPT-4 to generate a natural response by stitching together the gold claims using a template (Table 23). Please refer to the subsection below for more details

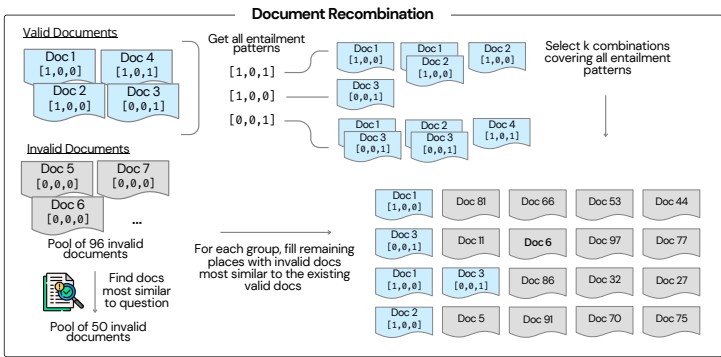

Figure 3: Document recombination process in augmented prompt curation.

on how the prompt is structured for each dataset. The prompt template asks GPT-4 to label each gold claim used with its index from the provided list (e.g., "[Gold Claim X]"), allowing for later matching of claims to documents. For unanswerable questions, a refusal response is assigned. To generate citations corresponding to each statement generated, we map the "[Gold Claim X]" labels to the appropriate documents. First, we extract all such labels from a sentence (which may contain multiple claims and labels). Then, we greedily identify the smallest combination of documents that covers these claims, minimizing over-citation. Details of this process is illustrated in Fig. 4.

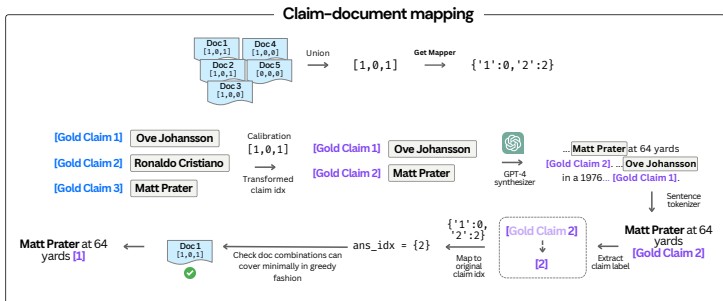

Figure 4: Claim-document-mapping process.

**Details on prompt structure for each dataset.** For ASQA, we include the question $q$, a list of (calibrated) gold claims, and their corresponding supporting documents $D$ as additional context. For ELI5, we follow Gao et al. (2023b) by decomposing each labeled response into three claims, which serve as a set of ground truth answers. Since the claim labels already provide sufficient context, we only fit the question and calibrated claims into the template. For QAMPARI, since its response format aligns with its labeled ground truth format (a list of entities), no additional action is required.

### E.5 OBTAINING $\mathbf{r}^-$

To create high-quality preference data, we aim to obtain quality negative (unpreferred) responses. We first fine-tune LLaMA-2-7b on the training set of the source datasets[12], creating $\mathcal{M}_{sft}$. We then test $\mathcal{M}_{sft}$ on the above-obtained dataset with approximately 70K questions and identify that 40K responses exhibit hallucinations. Table 9 shows the severity computation ($e_i$) and the frequency of each hallucination type ($w_i$). Thus, we can compute hallucination severity for each sample as $e_q = \sum_i e_i \cdot w_i$.

Table 9: Fraction of each hallucination amongst all the observed hallucinations in $\mathcal{M}_{sft}$ (40,985), with possible overlap. $w_i$ shows the severity computation of each hallucination. $I_{condition} = 1$ if condition is True otherwise it is 0. See Fig. 5 for the detailed breakdown of the last three errors.

| Hallucination type | Frequency | ($w_i$) | Severity ($e_i$) |
|---|---|---|---|
| Unwarranted Refusal | 8,786 | 0.50 | $I_{(A_g \neq \emptyset, A_r = \emptyset)}$ |
| Over Responsiveness | 13,067 | 0.50 | $I_{(A_g = \emptyset, A_r \neq \emptyset)}$ |
| Overcitation | 12,656 | 0.34 | 1 - CP |
| Improper Citation | 9,592 | 0.26 | 1 - CR |
| Inaccurate Claims | 14,783 | 0.40 | 1 - F1$_{AC}$ |

To obtain good negative samples, we first rank each of the 40K responses according to their severity score $e_q$. We then select the top 50% of the corresponding samples for both answerable and unanswerable responses. **Thus, we demonstrate the alignment data construction phase of TRUST-ALIGN, i.e., obtaining 19K samples with all the desired attributes $(\mathbf{q}, \mathbf{D}, \mathbf{r}^+, \mathbf{r}^-)$.** We perform DPO using this set of 19k samples to obtain the final aligned model.

---

[12]Seed questions, corresponding oracle documents, and the gold answers ($r^+$) are concatenated together using the refusal prompt in Table 25.

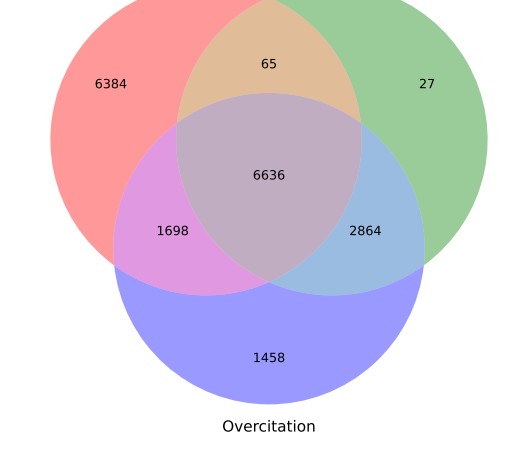

Figure 5: Statistics of hallucinations from the output of LLaMA-2-7b SFT model prompted using 70K $(q, D)$ samples obtained in Step-2 of TRUST-ALIGN.

# F ADDITIONAL ANALYSIS

## F.1 REVISED METRICS ARE LESS BIASED

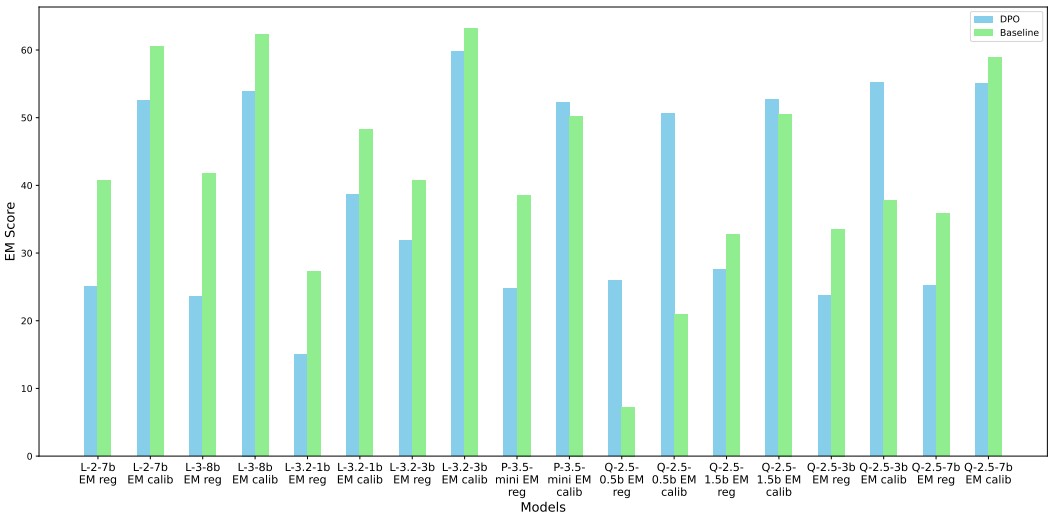

Figure 6: Comparison of AC regular and AC calibrated ($F1_{AC}$) across models on ASQA. EM regular/calibrated is synonymous with AC regular/calibrated.

Fig. 6 shows the performance of our models on EM regular and EM calibrated. In ASQA, our measure of Answer Correctness (AC) was Exact Match (EM) and thus EM regular/calibrated is synonymous with AC regular/calibrated. As seen across the models, AC regular tends to unduly penalize our models for refusals, resulting in baselines performing disproportionately better than our models. By measuring AC on both the answerable and answered set for $F1_{AC}$, we arrive at an AC metric that is more fair in the presence of refusals. By correcting for the bias toward answering at all costs, we are able to reveal more balanced perspective on model performance as demonstrated by a reduction in the performance gap (e.g. in LLaMA-2-7b, LLaMA-3-8b, LLaMA-3.2-1b, LLaMA-3.2-3b) or even revealing our model's stronger performance as compared to baseline (e.g. Phi-3.5

mini, Qwen-2.5-1.5b, Qwen2.5-3b). The ability to grade the model more fairly underscores the need for our calibrated metrics.

## F.2 UTILIZATION OF PARAMETRIC KNOWLEDGE

For an LLM used for an RAG task, it is important to study the tendency of LLM towards grounding its knowledge on the provided documents. To partially quantify this, we compute an uncalibrated answer correctness (AC) score for questions that are unanswerable by the provided documents; thus $A_G \cap A_D = \emptyset$ but $A_G \neq \emptyset$,

$$S_{\text{param}} = \frac{1}{|\mathcal{N}_r|} \sum_{q_i \in \mathcal{N}_r} \frac{|(A_R - (A_R \cap A_D)) \cap A_G|}{|A_R|} \tag{17}$$

Where, $A_G$, $A_D$, and $A_R$ are claims in the ground truth answer, claims present in the documents, and the claims generated in the response, respectively. $\mathcal{N}_r$ is the number of answered questions.

In Table 10, our analysis reveals that responsive models tend to rely on parametric knowledge more frequently. Notably, closed-source models like GPT-4 exhibit higher parametric knowledge usage compared to our models. However, this metric only partially captures the models' utilization of parametric knowledge. For instance, cases where models correctly generate gold claims without proper grounding may also indicate reliance on parametric knowledge. This phenomenon is evident in Table 12, where on ASQA, GPT-4 achieves a significantly higher $\textbf{F1}_{\textbf{AC}}$ than our models, yet its attribution groundedness score $\textbf{F1}_{\textbf{GC}}$ is five points lower.

Table 10: Detection of parametric knowledge usage under refusal prompting.

| Model | ASQA | | QAMPRARI | | ELI5 | |
|---|---|---|---|---|---|---|
| | AR (%) | $S_{\text{param}}$ | AR (%) | $S_{\text{param}}$ | AR (%) | $S_{\text{param}}$ |
| ICL-LLaMA-2 7B | 0.00 | 0.00 | 0.00 | 0.00 | 0.50 | 0.00 |
| ICL-LLaMA-3 8B | 1.48 | 1.79 | 3.90 | 16.92 | 0.00 | 0.00 |
| ICL-GPT-3.5 | 71.20 | 9.74 | 65.30 | 11.45 | 49.00 | 7.89 |
| ICL-GPT-4 | 86.81 | 12.71 | 73.40 | 13.05 | 61.50 | 9.05 |
| ICL-Claude-3.5 | 84.60 | 12.99 | 69.80 | 12.55 | 59.00 | 1.76 |
| TRUST-ALIGN (DPO-LLaMA-2-7B) | 65.30 | 8.15 | 31.10 | 8.45 | 21.60 | 5.56 |
| TRUST-ALIGN (DPO-LLaMA-3-8B) | 56.42 | 8.65 | 23.10 | 8.97 | 15.50 | 7.26 |

## F.3 THE SOURCE OF LLM HALLUCINATIONS

Model errors can be categorized into two primary sources:

1. **Parametric knowledge-based hallucination:** Errors arising from the model's internal knowledge representation.
2. **Information extraction failures:** Inability to accurately extract relevant information from provided documents.

To quantify these error types, we employ the following methodology:

- For the non-refused questions with errors, calculate the proportion of the incorrect answers that are:
  - Present in the provided documents
  - Absent from the provided documents

For answers absent from the documents, we can attribute the error to parametric knowledge-based hallucination. For answers present in the documents, the specific source of the error remains indeterminate as it can be attributed to both.

The substring matching (Gao et al., 2023b) is used here for searching for the existence of incorrect answers in the documents. As the model's response only on QAMPARI can be decomposed into atomic facts, we chose to perform this analysis on it. Specifically, for every answered question,

we calculate the proportion of incorrect answers present in or absent from the documents using the equations below:

$$\text{Presence} = \frac{1}{|\mathcal{N}_e|} \sum_{q_i \in \mathcal{A}_e} \frac{|A_R^e \cap A_D|}{|A_R^e|} \tag{18}$$

$$\text{Absence} = \frac{1}{|\mathcal{N}_e|} \sum_{q_i \in \mathcal{A}_e} \frac{|A_R^e - (A_R^e \cap A_D)|}{|A_R^e|} \tag{19}$$

Where $\mathcal{A}_e$ denotes the set of answerable questions that answered by the model with one or more incorrect answers; $A_D$, $A_R^e$ are facts present in the documents and erroneous facts generated in the response, respectively.

The findings are presented in Table 11. Our analysis reveals that, with the exception of LLaMA-2 7B which provides no responses, all other ICL-based models exhibit a higher tendency to produce erroneous answers based on their parametric knowledge compared to our models. Notably, Claude-3.5 demonstrates a more frequent reliance on its parametric knowledge, which elucidates its significantly lower TRUST-SCORE score in Table 12.

In summary, our investigation indicates that baseline models, including GPT-4 and GPT-3.5, are more susceptible to hallucinations stemming from their parametric knowledge.

Table 11: The proportions of erroneous answers present in or absent from the documents.

| Model | QAMPARI | |
|---|---|---|
| | Presence (%) | Absence (%) |
| ICL-LLaMA-2 7B | 0.00 | 0.00 |
| ICL-LLaMA-3 8B | 84.41 | 15.59 |
| ICL-GPT-3.5 | 85.04 | 14.96 |
| ICL-GPT-4 | 89.3 | 10.7 |
| ICL-Claude-3.5 | 72.18 | 27.82 |
| TRUST-ALIGN (DPO-LLaMA-2-7B) | 93.26 | 6.74 |
| TRUST-ALIGN (DPO-LLaMA-3-8B) | 95.63 | 4.37 |

### F.4 TRUST-ALIGN ENHANCES TRUSTWORTHINESS MORE ROBUSTLY THAN PROMPTING

Aligning with TRUST-ALIGN leads to more significant improvements in TRUST-SCORE compared to using prompting alone. While adding a refusal prompt has inconsistent effects on TRUST-SCORE and its subcomponents, it tends to be more beneficial in more capable models, such as LLaMA-2-13b and LLaMA-3-8b.

Relying solely on prompting to teach refusal is ineffective, as models' responsiveness becomes overly sensitive to the prompt. Under the default prompt, models rarely refuse (AR% close to 100), while adding a refusal prompt in ICL drastically reduces AR%, often to near zero, indicating indiscriminate refusal. This lack of nuanced refusal ability is also seen in post hoc methods. At both extremes, TRUST-SCORE scores suffer due to errors in correctly refusing questions and lower citation groundedness scores. In contrast, TRUST-ALIGN enables models to identify and correctly answer appropriate questions, resulting in AR% closer to the maximum answerable percentage and improvements in $F1_{GR}$.

It's important to note that responsiveness should not be the primary metric for comparing RAG systems when the retrieved documents are the same. The TRUST score rewards accurate answers, appropriate refusals, and correct citations while penalizing failures. Systems with low responsiveness will score poorly on TRUST, regardless of their overall response rate.

As shown in Table 19, Table 20, and Table 21, for PostCite, PostAttr, and Self-RAG, adding a refusal prompt results in minimal changes in TRUST-SCORE (e.g., ASQA Self-RAG with LLaMA-2-13b: 51.69% vs. 52.49%). Subcomponent analysis shows little difference in $F1_{GR}$ (42.74% vs. 39.15%), indicating that the refusal prompt does not effectively help models distinguish between answerable and unanswerable questions. These findings highlight the instability of relying on prompting to enhance trustworthiness and underscore the robustness of our system in achieving this goal.

## F.5 COMPARISON WITH CLOSED-SOURCE MODELS

We continue our comparison of trustworthiness against competitive closed-source models utilizing in-context learning techniques. As shown in Table 12, our aligned models outperform GPT-3.5 (69.23 vs. 67.64) and Claude-3.5 (69.23 vs. 64.36) on the ASQA dataset, and substantially outperform GPT-3.5 (55.31 vs. 38.95), GPT-4 (55.31 vs. 40.35), and Claude-3.5 (55.31 vs. 39.78) on QAMPARI. However, the responsiveness of current closed-source models remains much higher than that of our models: even with a refusal prompt, ICL-GPT-4 still answers a significant fraction of questions (86.81% on ASQA, 73.40% on QAMPARI). As discussed in Section 6, this tendency allows GPT-4 to achieve higher $F1_{AC}$ scores on ASQA, but it negatively impacts its attribution groundedness: its $F1_{GC}$ scores on both datasets are lower than those of our models. Similarly, GPT-4's $F1_{GR}$ scores on both datasets are also lower. On QAMPARI, the $F1_{AC}$ scores of all closed-source models are lower than those of our models.

Moreover, there still remains a gap between our models and the closed-source models on the ELI5 dataset. Our models' TRUST-SCORE is 2.45 points lower than that of the advanced ICL-GPT-4, and specifically, the $F1_{AC}$ and $F1_{GC}$ scores are lower. For higher $F1_{AC}$, as discussed in Section 6, it is due to a higher number of its answered answerable questions with comparable $EM_{AC}^{\alpha}$. As for higher $F1_{GC}$, We hypothesize that this gap could be attributed to the information density of the extracted claims utilized in constructing the alignment data (Section 4). Specifically, the three claims derived from the decomposition process may either be redundant or inadequate to fully encapsulate the information inherent in the original labelled response. In some cases, the decomposed claims may even fail to align with the original facts. First, insufficient information can lead the model to learn to extract fewer facts from the document, thereby reducing the answerability by covering fewer correct answers after training. Second, redundant information can impair grounded citation learning, as it repeats the same information across different claims, making the model less capable of performing precise citations from the corresponding documents. This issue is illustrated in the case study presented in Table 13.

> This experiment reveals that proprietary models demonstrate greater responsiveness compared to our models. While GPT-4 achieves superior $F1_{AC}$ scores, it underperforms in terms of $F1_{GC}$ and $F1_{GR}$, suggesting limitations in its ability to ground responses and refuse unanswerable questions. Overall, GPT-3.5 and GPT-4 outperform our models in utilizing retrieved documents for long-form question answering, primarily due to the limited capacity of our base model.

Table 12: Our models vs closed source: $AR\%$ := Answered Ratio in %; $F1_{AC}$ := Answer Correctness F1 (Calibrated); $F1_{GR}$ := Grounded refusals F1; $F1_{GC}$ := Grounded Citation F1; $TRUST$ := TRUST score. $R$ := Refusal prompt is used. $D$ := Default prompt is used.

| | | ASQA | | | | | QAMPARI | | | | | ELI5 | | | | |
| | | Responsiveness | Trustworthiness | | | | Responsiveness | Trustworthiness | | | | Responsiveness | Trustworthiness | | | |
| | | AR (%) | Truthfullness | | Attr. Grdness | TRUST | AR (%) | Truthfullness | | Attr. Grdness | TRUST | AR (%) | Truthfullness | | Attr. Grdness | TRUST |
| | Prompt | | $F1_{AC}$ | $F1_{GR}$ | $F1_{GC}$ | | | $F1_{AC}$ | $F1_{GR}$ | $F1_{GC}$ | | | $F1_{AC}$ | $F1_{GR}$ | $F1_{GC}$ | |
| --- | --- | --- | --- | --- | --- | --- | --- | --- | --- | --- | --- | --- | --- | --- | --- | --- |
| | | | | | | | | | Closed-source Models | | | | | | | |
| ICL-GPT-3.5 | R | 71.20 | 52.91 | 66.07 | 83.94 | 67.64 | 65.30 | 26.57 | 58.49 | 31.80 | 38.95 | 49.00 | 32.38 | 58.27 | 57.29 | 49.31 |
| ICL-GPT-4 | R | 86.81 | **62.96** | 61.85 | 84.35 | **69.72** | 73.40 | 30.13 | 55.46 | 35.45 | 40.35 | 61.50 | **33.05** | 53.11 | **61.84** | **49.33** |
| ICL-Claude-3.5 | R | 84.60 | 59.97 | 64.77 | 68.35 | 64.36 | 69.80 | 28.40 | 58.10 | 32.83 | 39.78 | 59.00 | 11.34 | 54.00 | 12.43 | 25.92 |
| ICL-GPT-3.5 | D | 94.41 | 55.03 | 52.48 | 78.04 | 61.85 | 94.50 | 20.30 | 29.54 | 21.22 | 23.69 | 93.50 | 23.88 | 24.68 | 46.28 | 31.61 |
| ICL-GPT-4 | D | 92.72 | 62.37 | 54.17 | 79.70 | 65.41 | 87.70 | 26.19 | 40.03 | 30.02 | 32.08 | 82.80 | 29.09 | 37.02 | 48.33 | 38.15 |
| ICL-Claude-3.5 | D | 82.49 | 54.20 | **66.49** | 58.88 | 59.86 | 69.90 | 0.00 | 57.40 | 0.00 | 19.13 | 56.60 | 11.56 | 56.03 | 11.22 | 26.27 |
| | | | | | | | | | TRUST-ALIGN Models | | | | | | | |
| DPO-LLaMA-2-7b | R | 65.30 | 52.48 | 66.12 | 83.94 | 67.51 | 31.10 | 32.09 | **71.83** | 51.33 | 51.75 | 21.60 | 22.54 | 63.27 | 48.43 | 44.75 |
| DPO-LLaMA-3-8b | R | 56.43 | 53.94 | 65.49 | **88.26** | 69.23 | 23.10 | **35.94** | 71.11 | **58.87** | **55.31** | 15.50 | 22.81 | **64.00** | 53.84 | 46.88 |

Table 13: A case study of the failure of decomposition.

---

**Insufficient case**

Question:  Why do burns blister and why do burn wounds remain warm long after the injury occurred?
Label:  Burn blisters occur when the second layer of the skin is damaged, they occur to protect the underlying skin layers from more damage and infection.  You could see it as the bodys/skins natural bandage, so never pop them.  The skin remain warm because of the increased blood in the area to repair and replace the damaged skin.

Decomposed claims:
1.  Burn blisters occur when the second layer of skin is damaged.
2.  Burn wounds remain warm due to increased blood flow to the area to repair and replace damaged skin.

Missing points:
1.  Protection and Infection:  The first claim does not mention that the blisters protect the underlying skin from more damage and infection, which is a significant part of the explanation in the answer.
2.  Never Pop Them:  The answer advises against popping blisters, which is a preventive measure not mentioned in the claims.

---

**Redundant case**

Question:  How do fitness trackers know that you actually sleeping but not just laying there resting, being awake?
Label:  Your heart beats slows down when you sleep, they will use a mixture of heart rate and how long you haven't moved to determine how you've slept

Decomposed claims:  1.  The combined factors of heart rate and inactivity determine sleep assessment.
2.  Fitness trackers consider the duration of inactivity to assess sleep.
3.  A slowed heart rate is an indicator of sleep that fitness trackers monitor.

Redundant point:  The first claim has already summarised the core statement, and the last two claims just expand it and give more details

---

## F.6 ADAPTABILITY WITH DIFFERENT ALIGNMENT TECHNIQUES

To demonstrate the robustness of our synthesized alignment data across different training methods, Table 14 also includes the performance of SFT and SIMPO (Meng et al., 2024) methods. Compared to the SFT baseline, which only utilizes the positive data points in the alignment pairs to fine-tune the base model, preference optimization methods, such as DPO and SIMPO, consistently show performance improvements, highlighting the versatility of our data pipeline. Unlike the SFT approach, DPO and SIMPO demonstrate improved TRUST scores, albeit with a reduction in responsiveness. This decrease in responsiveness is actually a favorable outcome, as it indicates that the models are less likely to attempt to answer questions for which they lack sufficient information.

Table 14: Results using different alignment methods on the ASQA dataset.

| Alignment | TRUST-ALIGN Model | Responsiveness (AR%) | $F1_{AC}$ | $F1_{GR}$ | $F1_{GC}$ | TRUST |
|---|---|---|---|---|---|---|
| DPO | LLaMA-2-7b | 65.30 | 52.48 | 66.12 | 83.94 | 67.51 |
| | LLaMA-3-8b | 56.43 | **53.94** | 65.49 | **88.26** | **69.23** |
| SIMPO | LLaMA-2-7b | 72.47 | 53.19 | **66.44** | 82.21 | 67.28 |
| | LLaMA-3-8b | 57.38 | 49.84 | 64.13 | 86.86 | 66.94 |

## F.7 EVALUATION DATA CREATION WITHOUT USING TRUE

The determination of question answerability in our dataset is based on a combination of substring matching and TRUE criteria, as detailed in Section 2. Additionally, we developed an alternative version of the evaluation data that relies solely on substring matching, disregarding the TRUE criterion. This relaxation of answerability constraints results in an increased number of answerable questions. The findings from this analysis are presented in Table 15. It is worth noting that the overall trends observed in this analysis align with those reported in Table 2, which employs the combined approach of substring matching followed by TRUE verification.

Table 15: Results on ASQA, QAMPARI evaluation datasets where the data are created without using TRUE; **AR%** := Answered Ratio in %; **F1$_{AC}$** := Answer Correctness F1 (Calibrated); **F1$_{GR}$** := Grounded refusals F1; **F1$_{GC}$** := Citation Grounded F1; **TRUST** := TRUST-SCORE. **R** := Refusal prompt is used. **D** := Default prompt is used.

| | | ASQA (779 answerable, 169 unanswerable) | | | | | QAMPARI (586 answerable, 414 unanswerable) | | | | |
|---|---|---|---|---|---|---|---|---|---|---|---|
| | | Responsiveness | Trustworthiness | | | | Responsiveness | Trustworthiness | | | |
| | | AR (%) | Truthfullness | | Attr. Grdness | TRUST | AR (%) | Truthfullness | | Attr. Grdness | TRUST |
| | Prompt | | F1$_{AC}$ | F1$_{GR}$ | F1$_{GC}$ | | | F1$_{AC}$ | F1$_{GR}$ | F1$_{GC}$ | |
| | | | | | | LLaMA-2-7b | | | | | |
| ICL | R | 0.00 | 0.00 | 15.13 | 0.00 | 5.04 | 0.00 | 0.00 | 29.28 | 0.00 | 9.76 |
| PostCite | R | 10.44 | 0.13 | 24.91 | 0.00 | 8.35 | 34.40 | 0.00 | 52.57 | 9.50 | 20.69 |
| PostAttr | R | 10.44 | 0.13 | 24.91 | 0.00 | 8.35 | 34.40 | 0.00 | 52.57 | 3.78 | 18.78 |
| Self-RAG | R | 100.00 | 44.40 | 45.11 | 63.49 | 51.00 | 96.00 | 9.64 | 44.15 | 19.95 | 24.58 |
| ICL | D | 94.30 | 51.13 | 54.01 | 44.86 | 50.00 | 93.60 | 13.31 | 43.37 | 3.88 | 20.19 |
| PostCite | D | 88.71 | 2.64 | 54.63 | 0.98 | 19.42 | 56.30 | 0.00 | 52.85 | 7.73 | 20.19 |
| PostAttr | D | 87.24 | 2.71 | 55.63 | 0.43 | 19.59 | 51.10 | 0.00 | 52.45 | 4.70 | 19.05 |
| Self-RAG | D | 98.00 | 47.22 | 46.27 | 56.59 | 50.03 | 96.20 | 12.13 | 40.83 | 15.44 | 22.80 |
| | | | | | | LLaMA-2-13b | | | | | |
| ICL | R | 17.41 | 19.29 | 31.22 | 14.14 | 21.55 | 26.50 | 0.63 | 53.67 | 0.00 | 18.10 |
| PostCite | R | 90.51 | 2.04 | 56.40 | 1.53 | 19.99 | 100.00 | 0.00 | 36.95 | 8.05 | 15.00 |
| PostAttr | R | 90.51 | 2.04 | 56.40 | 0.17 | 19.54 | 100.00 | 0.00 | 36.95 | 2.95 | 13.30 |
| Self-RAG | R | 100.00 | 48.10 | 45.11 | 69.79 | 54.33 | 72.70 | 4.90 | 60.20 | 26.91 | 30.67 |
| ICL | D | 97.57 | 51.18 | 50.16 | 9.40 | 36.91 | 97.80 | 0.05 | 41.05 | 0.00 | 13.70 |
| PostCite | D | 89.77 | 0.07 | 54.96 | 0.00 | 18.34 | 63.00 | 0.00 | 53.22 | 7.14 | 20.12 |
| PostAttr | D | 89.24 | 0.07 | 55.01 | 0.00 | 18.36 | 58.50 | 0.00 | 52.31 | 4.56 | 18.96 |
| Self-RAG | D | 97.68 | 49.10 | 48.47 | 63.39 | 53.65 | 96.30 | 6.04 | 41.17 | 21.06 | 22.76 |
| | | | | | | LLaMA-3-8b | | | | | |
| ICL | R | 1.48 | 2.12 | 17.09 | **89.14** | 36.12 | 3.90 | 4.77 | 35.42 | 20.24 | 20.14 |
| PostCite | R | 77.53 | 34.32 | 54.76 | 28.01 | 39.03 | 87.00 | 9.90 | 47.98 | 8.42 | 22.10 |
| PostAttr | R | 77.53 | 34.32 | 54.76 | 5.95 | 31.68 | 87.00 | 9.90 | 47.98 | 1.64 | 19.84 |
| ICL | D | 89.66 | 58.83 | **64.47** | 62.12 | 61.81 | 70.80 | 7.48 | 61.03 | 4.81 | 24.44 |
| PostCite | D | 97.26 | 37.48 | 49.41 | 17.89 | 34.93 | 92.00 | 3.35 | 45.43 | 11.14 | 19.97 |
| PostAttr | D | 97.47 | 37.44 | 48.95 | 3.18 | 29.86 | 93.00 | 3.32 | 46.03 | 5.65 | 18.33 |
| | | | | | | TRUST-ALIGN Models | | | | | |
| DPO-LLaMA-2-7b | R | 65.30 | 47.85 | 61.60 | 84.95 | 64.80 | 32.30 | **27.80** | **63.60** | 49.42 | 46.94 |
| DPO-LLaMA-3-8b | R | 56.43 | 48.18 | 57.60 | 88.84 | **64.87** | 22.40 | 26.57 | 56.84 | **58.77** | **47.39** |

## F.8 EFFECT OF DATA SIZE ON DPO PERFORMANCE

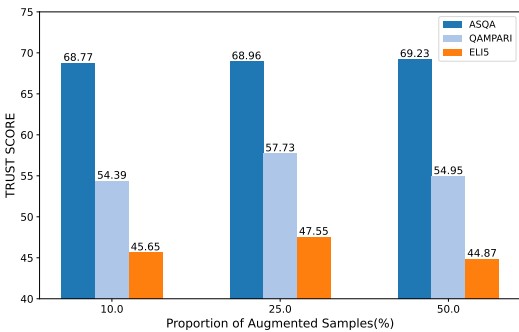

Figure 7: Influence of the proportion of augmented training samples on TRUST-SCORE in LLaMA-3-8b.

During the bulk of our experiments, we chose to utilize the top 50% of augmented samples to form the training dataset for DPO alignment due to cost-effectiveness. Here, we investigate whether varying the quantity and difficulty of the samples would influence the final model performance. Fig. 7 shows how the amount of training samples affect the final performance of TRUST-ALIGN model. Notably, selecting the top 25% of augmented samples achieved the highest performance on QAMPARI (57.73) and ELI5 (47.55). The absence of a clear trend suggesting that "more data is better" can be attributed to the nature of the data itself. Although document recombination can generate a large number of samples, those with lower hallucination severity scores tend to have limited complexity and high information redundancy. As a result, these additional samples do not provide substantial new or challenging information for the model to learn from, limiting their effectiveness in improving model performance. When training with the top 50% of augmented samples, the model may be experiencing overfitting, which could explain the observed decline in performance. Therefore, it is likely that even better performance could be attained by carefully tuning the amount of augmented data used. This finding underscores a limitation of our pipeline, revealing that the diversity of document content plays a crucial role in determining the quality of the augmented samples.

## F.9 FINE-TUNING GPT-4O

We fine-tuned GPT-4o using our SFT dataset. The results reported in Table 16 indicate the consistent improvement in trustworthiness scores of GPT-4o as a result of supervised fine-tuning.

Table 16: Performance of supervised fine-tuned GPT-4o.

| Model | Type | ASQA (610 answerable, 338 unanswerable) | | | | | QAMPARI (295 answerable, 705 unanswerable) | | | | | ELI5 (207 answerable, 793 unanswerable) | | | | |
|---|---|---|---|---|---|---|---|---|---|---|---|---|---|---|---|---|
| | | Resp. | Trustworthiness | | | | Resp. | Trustworthiness | | | | Resp. | Trustworthiness | | | |
| | | AR (%) | Truthfullness | | Att-Grd. | TRUST | AR (%) | Truthfullness | | Att-Grd. | TRUST | AR (%) | Truthfullness | | Att-Grd. | TRUST |
| | | | $F1_{AC}$ | $F1_{GR}$ | $F1_{GC}$ | | | $F1_{AC}$ | $F1_{GR}$ | $F1_{GC}$ | | | $F1_{AC}$ | $F1_{GR}$ | $F1_{GC}$ | |
| GPT-4o | ICL | 84.49 | 62.92 | 61.40 | 73.66 | 65.88 | 60.40 | 14.29 | 75.20 | 20.43 | 33.69 | 66.1 | 35.25 | 68.33 | 37.71 | 41.58 |
| | TRUST-ALIGN (SFT) | 74.26 | 59.22 | 68.62 | 87.54 | **72.09** | 34.6 | 41.56 | 77.15 | 53.64 | **56.99** | 25.5 | 24.1 | 68.34 | 56.09 | **48.99** |

# G GPT-4 BASED DATA PIPELINE

For the GPT-4 data pipeline, we employ GPT-4 to simulate a critic that performs two key tasks in succession. First, it identifies and revises mistakes or supplements missing information in the given response based on correct answers. Second, it validates the attribution of statement-level citations and corrects them accordingly. The detailed instruction is provided in Table 26.

**Coverage critiques.** To ensure that the correct answers are accurately reflected in the given response, we prompt GPT-4 with the corresponding question, correct answers, and reference facts (documents that support the provided correct answers) as context. GPT-4 is then asked to locate specific mistakes or identify any missing correct answers in the given response. After identifying

coverage-related issues, GPT-4 is instructed to minimally revise the original response to correct these issues based on the detected problems. This minimal revision approach is intended to generate more precise data for alignment learning.

**Citation critiques.** Based on the revised content, we further tokenize it into individual statements to enable a more fine-grained citation check in later stages. We format all documents in the instruction as holistic facts and instruct GPT-4 to determine the attribution of each statement relative to these facts. We define three levels of attribution: SUPPORT, OPPOSE, and IRRELEVANT. We then compare GPT-4's attribution results to the original attributions in the response, modifying the original attributions wherever they do not align with GPT-4's critiques. Finally, we concatenate all citation-revised statements to form the final revised response.

# H EXPERIMENTAL SETUP

## H.1 IMPLEMENTATION DETAILS

For all experiments involving our tuned models and baselines, we provided the top 5 retrieved documents as context and used decoding temperatures of 0.1 and 0.5, respectively, with other settings consistent with those in Gao et al. (2023b). We evaluated three representative open-source model families: the LLaMA series [13](Touvron et al., 2023; Dubey et al., 2024), the Qwen series [14] (Yang et al., 2024), and Phi-3.5-mini (Abdin et al., 2024), and three proprietary model families: GPT-4 (OpenAI et al., 2024), GPT-3.5 (Brown et al., 2020) [15], and Claude-3.5-Sonnet [16]. We perform the full parameter fine-tuning for better performance. For supervised fine-tuning (SFT), we trained the models for 2 epochs with a learning rate of 2e-5. For direct preference optimization (DPO) alignment, we trained the models for 2 epochs with a beta value of 0.5. All experiments were conducted on NVIDIA A40 40G GPUs.

## H.2 DATASET DETAILS

Following Liu et al. (2023); Gao et al. (2023b), to form $D$, we divide large text documents into 100-word passages and limit the number of citations $\mathcal{C}_i$ for each claim to a maximum of three. If the response is empty, it is excluded from evaluation. We provide statistics of our evaluation in Table 17.

Table 17: Statistics of the evaluation dataset.

|  | ASQA | QAMPARI | ELI5 | ExpertQA |
|---|---|---|---|---|
| Total # of Samples | 948 | 1000 | 1000 | 2169 |
| # Answerable Samples | 610 | 295 | 207 | 682 |
| # Unanswerable Samples | 338 | 705 | 793 | 1487 |

**ASQA (Stelmakh et al., 2023).** This long-form factoid dataset features ambiguous queries from AmbigQA (Li et al., 2023), requiring multiple short answers to address different aspects. It includes comprehensive long-form answers that combine these short responses.

**QAMPARI (Amouyal et al., 2023).** This factoid QA dataset is derived from Wikipedia, with answers consisting of lists of entities gathered from various passages.

**ELI5 (Fan et al., 2019).** This dataset is a long-form QA collection based on the Reddit forum "Explain Like I'm Five" (ELI5). Most ELI5 questions require the model to utilize knowledge from multiple passages to formulate a complete answer. The ELI5 dataset is frequently used in related research due to its challenging nature (Nakano et al., 2021; Menick et al., 2022; Jiang et al., 2023).

---

[13]LLaMA-2-7b, LLaMA-2-13b, LLaMA-2-70b, LLaMA-3.2-1b, LLaMA-3.2-3b, LLaMA-3-8b

[14]Qwen-2.5-0.5b, Qwen-2.5-1.5b, Qwen-2.5-3b, Qwen-2.5-7b

[15]We utilize the latest version on the AzureOpenAI Service: `https://learn.microsoft.com/en-us/azure/ai-services/openai/concepts/models`

[16]`https://www.anthropic.com/news/claude-3-5-sonnet`

**ExpertQA   (Malaviya et al., 2024).**   This dataset spans various topics and requires domain-specific knowledge to solve long-form questions. To further verify the generalizability of our approach, we test our best model and some of the baselines on this unseen dataset.

## H.3   BASELINES

Table 18: A case study illustrating the necessity of GPT-4o matching for detecting refusals in models prompted with default prompt.

```
Question:  Who killed the man in thelma and louise?

Answer:  "In the movie Thelma and Louise, the man who is killed is not
explicitly identified as being killed by any specific character.  The
movie ends with a shootout and a crash, and it is unclear who
survives or who is responsible for the deaths.  The search results
you provided do not provide any clear answers to this question.

Fuzzy match judgement:  Not refusal
Human judgement:  Refusal
GPT-4o judgement:  Refusal
```

### H.3.1   IN-CONTEXT LEARNING (ICL)

Following Gao et al. (2023b), we prepend with two demonstrations, each consisting of a query, top-5 retrieved passages, and an answer with inline citations.

### H.3.2   POST-HOC SEARCH (POSTCITE)

Following Gao et al. (2023b), we first prompt the model under a closed book setting i.e. without any retrieved passages, to obtain an uncited answer. Then, GTR is used to find the best matching citation among the top-5 retrieved passages for every statement.

### H.3.3   POST-HOC ATTRIBUTE (POSTATTR)

Similar to PostCite, we first obtain model response under a closed book setting. Then we use the TRUE NLI model to find the best matching citation among top-5 retrieved passages.

### H.3.4   SELF-RAG

Self-RAG trains the LLM to retrieve documents on demand using special reflection tokens and enhances generation quality through self-reflection. We compare our results against the 7b and 13b models released, using the default settings as described in (Asai et al., 2024).

### H.3.5   FRONT

FRONT (Huang et al., 2024b) utilizes a fine-grained attribution training framework that first grounds specific supporting quotes, and then generates responses with citations based on those quotes. It tunes the LLM with automatically collected data based on ChatGPT and quality filtering. We reproduce its 7b model for the evaluation.

## H.4   REFUSAL DETECTION

We employed two methods to measure refusals robustly. In a refusal prompt, models were explicitly instructed to respond only with the phrase: *"I apologize, but I couldn't find an answer to your question in the search results."* without providing any further explanation. As the models generally complied with this pattern, we were able to apply fuzzy matching[17] to detect the phrase above indicating refusal. For models responding to a default prompt, refusals did not adhere to a fixed pattern,

---

[17]`Fuzz Partial Ratio` was used to mitigate the impact of string length.

making detection more challenging. Two human annotators verified that fuzzy matching yielded poor performance Table 18. Hence, GPT-4o was employed as an evaluator to classify whether an answer should be considered a refusal. The specific prompt used is provided in Table 27.

# I  DETAILED RESULTS

Table 19, Table 20, Table 21 and Table 22 show the full results of our experiments.

# J  PROMPT TEMPLATES

Table 23, Table 24, Table 25, Table 26 and Table 27 show the prompts used in our experiments.

Table 19: Detailed ASQA results.

| | Prompt | AR% | $AC_{reg}$ | $P_{AC}$ | $R_{AC}$ | $FI_{AC}$ | $R_{ref}$ | $P_{ref}$ | $Fl_{ref}$ | $R_{ans}$ | $P_{ans}$ | $Fl_{ans}$ | $FI_{GR}$ | $R_{cite}$ | $P_{cite}$ | $FI_{GC}$ | TRUST-SCORE |
|---|---|---|---|---|---|---|---|---|---|---|---|---|---|---|---|---|---|
| **Qwen-2.5-0.5b** | | | | | | | | | | | | | | | | | |
| ICL | R | 29.85 | 7.26 | 33.06 | 15.34 | 20.96 | 74.56 | 37.89 | 50.25 | 32.30 | 69.61 | 44.12 | 47.19 | 0.35 | 0.35 | 0.35 | 22.83 |
| PostCite | R | 46.10 | 4.72 | 10.24 | 7.34 | 8.55 | 57.40 | 37.96 | 45.70 | 48.03 | 67.05 | 55.97 | 50.84 | 8.23 | 8.23 | 8.23 | 22.54 |
| PostAttr | R | 46.10 | 4.72 | 10.24 | 7.34 | 8.55 | 57.40 | 37.96 | 45.70 | 48.03 | 67.05 | 55.97 | 50.84 | 2.23 | 2.23 | 2.23 | 20.54 |
| FRONT | R | 100.00 | 27.19 | 35.19 | 54.69 | 42.83 | 0.00 | 0.00 | 0.00 | 100.00 | 64.35 | 78.31 | 39.15 | 47.55 | 44.30 | 45.87 | 42.62 |
| SFT | R | 83.44 | 20.12 | 34.28 | 44.45 | 38.71 | 27.81 | 59.87 | 37.98 | 89.67 | 69.15 | 78.09 | 58.03 | 61.93 | 53.61 | 57.47 | 51.40 |
| TRUST-ALIGN (DPO) | R | 71.84 | 25.98 | 47.95 | 53.53 | 50.59 | 42.31 | 53.56 | 47.27 | 79.67 | 71.37 | 75.29 | 61.28 | 52.98 | 51.84 | 52.40 | 54.76 |
| TRUST-ALIGN (DPO-25%) | R | 77.00 | 22.03 | 39.36 | 47.10 | 42.88 | 37.28 | 57.80 | 45.32 | 84.92 | 70.96 | 77.31 | 61.32 | 59.25 | 57.56 | 58.39 | 54.20 |
| **Qwen-2.5-1.5b** | | | | | | | | | | | | | | | | | |
| ICL | R | 98.52 | 32.70 | 41.78 | 63.98 | 50.55 | 2.66 | 64.29 | 5.11 | 99.18 | 64.78 | 78.37 | 41.74 | 5.47 | 8.62 | 6.69 | 32.99 |
| PostCite | R | 71.73 | 9.80 | 15.51 | 17.30 | 16.36 | 31.66 | 39.93 | 35.31 | 73.61 | 66.03 | 69.61 | 52.46 | 15.40 | 15.40 | 15.40 | 28.07 |
| PostAttr | R | 71.73 | 9.80 | 15.51 | 17.30 | 16.36 | 31.66 | 39.93 | 35.31 | 73.61 | 66.03 | 69.61 | 52.46 | 4.45 | 4.45 | 4.45 | 24.42 |
| FRONT | R | 99.26 | 38.16 | 47.58 | 73.40 | 57.74 | 2.07 | 100.00 | 4.06 | 100.00 | 64.82 | 78.66 | 41.36 | 53.48 | 58.11 | 55.70 | 51.60 |
| SFT | R | 78.27 | 23.09 | 40.30 | 49.02 | 44.24 | 33.14 | 54.37 | 41.18 | 84.59 | 69.54 | 76.33 | 58.75 | 75.67 | 67.01 | 71.08 | 58.02 |
| TRUST-ALIGN (DPO) | R | 72.57 | 27.57 | 49.70 | 56.05 | 52.69 | 42.90 | 55.77 | 48.49 | 81.15 | 71.95 | 76.27 | 62.38 | 68.06 | 65.61 | 66.81 | 60.63 |
| TRUST-ALIGN (DPO-25%) | R | 76.48 | 25.08 | 45.08 | 53.57 | 48.96 | 39.05 | 59.19 | 47.06 | 81.59 | 71.59 | 77.75 | 62.41 | 76.30 | 74.72 | 75.50 | 62.29 |
| **Qwen-2.5-3b** | | | | | | | | | | | | | | | | | |
| ICL | R | 27.43 | 33.55 | 63.11 | 26.90 | 37.72 | 84.02 | 41.28 | 55.36 | 33.77 | 79.23 | 47.36 | 51.36 | 42.49 | 66.08 | 51.72 | 46.93 |
| PostCite | R | 8.76 | 13.58 | 39.96 | 5.44 | 9.58 | 93.79 | 36.65 | 52.70 | 10.16 | 74.70 | 17.89 | 35.30 | 10.94 | 10.94 | 10.94 | 18.61 |
| PostAttr | R | 8.76 | 13.58 | 39.96 | 5.44 | 9.58 | 93.79 | 36.65 | 52.70 | 10.16 | 74.70 | 17.89 | 35.30 | 36.29 | 36.29 | 36.29 | 27.06 |
| FRONT | R | 97.47 | 34.56 | 45.78 | 69.35 | 55.15 | 5.03 | 70.83 | 9.39 | 98.85 | 65.26 | 78.62 | 44.01 | 64.80 | 60.78 | 62.72 | 53.96 |
| SFT | R | 75.21 | 24.28 | 43.85 | 51.25 | 47.26 | 38.17 | 54.89 | 45.03 | 82.62 | 70.69 | 76.19 | 60.61 | 75.99 | 70.40 | 73.09 | 60.32 |
| TRUST-ALIGN (DPO) | R | 49.47 | 23.72 | 63.48 | 48.81 | 55.19 | 71.01 | 50.10 | 58.75 | 60.82 | 79.10 | 68.77 | 63.76 | 78.77 | 78.52 | 78.64 | 65.86 |
| TRUST-ALIGN (DPO-25%) | R | 61.92 | 22.07 | 50.36 | 48.46 | 49.39 | 55.92 | 52.35 | 54.08 | 71.80 | 74.62 | 73.18 | 63.63 | 81.63 | 79.19 | 80.39 | 64.47 |
| **Qwen-2.5-7b** | | | | | | | | | | | | | | | | | |
| ICL | R | 92.09 | 35.79 | 50.06 | 71.64 | 58.94 | 17.46 | 78.67 | 28.57 | 97.38 | 68.04 | 80.11 | 54.34 | 77.62 | 73.42 | 75.46 | 62.91 |
| PostCite | R | 91.46 | 20.91 | 23.44 | 33.31 | 27.52 | 9.76 | 40.74 | 15.75 | 92.13 | 64.82 | 76.10 | 45.93 | 4.19 | 4.19 | 4.19 | 25.88 |
| PostAttr | R | 91.46 | 20.91 | 23.44 | 33.31 | 27.52 | 9.76 | 40.74 | 15.75 | 92.13 | 64.82 | 76.10 | 45.93 | 17.92 | 17.92 | 17.92 | 30.46 |
| FRONT | R | 86.39 | 41.77 | 56.34 | 75.64 | 64.58 | 27.51 | 72.09 | 39.83 | 94.10 | 70.09 | 80.34 | 60.08 | 61.45 | 55.41 | 58.27 | 60.98 |
| SFT | R | 65.30 | 24.25 | 50.36 | 51.10 | 50.73 | 53.25 | 54.71 | 53.97 | 75.57 | 74.47 | 75.02 | 64.50 | 85.79 | 78.67 | 82.07 | 65.77 |
| TRUST-ALIGN (DPO) | R | 59.49 | 25.15 | 57.28 | 52.96 | 55.04 | 62.13 | 54.69 | 58.17 | 71.48 | 77.30 | 74.28 | 66.22 | 84.33 | 82.83 | 83.57 | 68.28 |
| TRUST-ALIGN (DPO-25%) | R | 58.44 | 23.56 | 55.14 | 50.08 | 52.49 | 63.61 | 54.57 | 58.74 | 70.66 | 77.80 | 74.05 | 66.40 | 87.60 | 85.17 | 86.37 | 68.42 |
| **Phi3.5-mini** | | | | | | | | | | | | | | | | | |
| ICL | R | 63.19 | 38.58 | 50.70 | 49.78 | 50.24 | 39.35 | 38.11 | 38.72 | 64.59 | 65.78 | 65.18 | 51.95 | 40.49 | 45.03 | 42.64 | 48.28 |
| PostCite | R | 23.10 | 21.76 | 28.36 | 10.18 | 14.98 | 76.92 | 35.67 | 48.73 | 23.11 | 64.38 | 34.02 | 41.38 | 9.40 | 9.40 | 9.40 | 21.92 |
| PostAttr | R | 23.10 | 21.76 | 28.36 | 10.18 | 14.98 | 76.92 | 35.67 | 48.73 | 23.11 | 64.38 | 34.02 | 41.38 | 1.24 | 1.24 | 1.24 | 19.20 |
| FRONT | R | 99.79 | 41.59 | 52.06 | 80.74 | 63.30 | 0.59 | 100.00 | 1.18 | 100.00 | 64.48 | 78.41 | 39.79 | 74.60 | 68.89 | 71.63 | 58.24 |
| SFT | R | 66.46 | 24.75 | 51.10 | 52.77 | 51.92 | 51.78 | 55.03 | 53.35 | 76.56 | 74.13 | 75.32 | 64.34 | 87.38 | 78.62 | 82.77 | 66.34 |
| TRUST-ALIGN (DPO) | R | 66.56 | 24.81 | 51.36 | 53.13 | 52.23 | 51.48 | 54.89 | 53.13 | 76.56 | 74.01 | 75.26 | 64.20 | 87.88 | 82.99 | 85.36 | 67.26 |
| TRUST-ALIGN (DPO-25%) | R | 66.14 | 25.46 | 53.51 | 55.00 | 54.25 | 54.44 | 57.32 | 55.84 | 77.54 | 75.44 | 76.48 | 66.16 | 87.13 | 81.63 | 84.29 | 68.23 |
| **LLaMA-2-7b** | | | | | | | | | | | | | | | | | |
| ICL | R | 0.00 | 12.78 | 0.00 | 0.00 | 0.00 | 100.00 | 35.65 | 52.57 | 0.00 | 0.00 | 0.00 | 26.28 | 0.00 | 0.00 | 0.00 | 8.76 |
| PostCite | R | 10.44 | 8.49 | 0.25 | 0.04 | 0.07 | 90.53 | 36.04 | 51.56 | 10.98 | 67.68 | 18.90 | 35.23 | 0.00 | 0.00 | 0.00 | 11.77 |
| PostAttr | R | 10.44 | 8.49 | 0.25 | 0.04 | 0.07 | 90.53 | 36.04 | 51.56 | 10.98 | 67.68 | 18.90 | 35.23 | 0.00 | 0.00 | 0.00 | 11.77 |
| Self-RAG | R | 100.00 | 28.87 | 37.13 | 57.71 | 45.19 | 0.00 | 0.00 | 0.00 | 100.00 | 64.35 | 78.31 | 39.15 | 59.27 | 68.35 | 63.49 | 49.28 |
| FRONT | R | 100.00 | 40.72 | 49.69 | 77.22 | 60.47 | 0.00 | 0.00 | 0.00 | 100.00 | 64.35 | 78.31 | 39.15 | 68.45 | 69.27 | 68.86 | 56.16 |
| ICL | D | 94.30 | 32.29 | 42.06 | 62.79 | 50.38 | 11.54 | 72.22 | 19.90 | 97.54 | 66.55 | 79.12 | 49.51 | 44.21 | 43.14 | 43.67 | 47.85 |
| PostCite | D | 88.71 | 1.91 | 1.98 | 2.73 | 2.30 | 16.27 | 51.40 | 24.72 | 91.48 | 66.35 | 76.91 | 50.82 | 0.98 | 0.98 | 0.98 | 18.03 |
| PostAttr | D | 87.24 | 1.91 | 2.01 | 2.73 | 2.32 | 18.05 | 50.41 | 26.58 | 90.16 | 66.51 | 76.55 | 51.56 | 0.43 | 0.43 | 0.43 | 18.10 |
| Self-RAG | D | 98.00 | 30.11 | 38.63 | 59.41 | 46.82 | 2.37 | 42.11 | 4.48 | 98.20 | 64.48 | 77.84 | 41.16 | 50.69 | 64.05 | 56.59 | 48.19 |
| SFT | R | 80.17 | 29.21 | 47.96 | 59.76 | 53.21 | 36.69 | 63.66 | 47.15 | 89.51 | 71.84 | 79.71 | 63.43 | 83.36 | 76.18 | 79.61 | 65.42 |
| TRUST-ALIGN (DPO) | R | 65.30 | 25.04 | 52.10 | 52.87 | 52.48 | 55.33 | 56.84 | 56.07 | 76.72 | 75.61 | 76.16 | 66.12 | 85.35 | 82.57 | 83.94 | 67.51 |
| TRUST-ALIGN (DPO-25%) | R | 71.31 | 27.17 | 51.32 | 56.87 | 53.95 | 48.82 | 60.66 | 54.10 | 82.46 | 74.41 | 78.23 | 66.16 | 84.37 | 82.07 | 83.20 | 67.77 |
| **LLaMA-2-13b** | | | | | | | | | | | | | | | | | |
| ICL | R | 17.41 | 9.17 | 50.54 | 13.67 | 21.52 | 86.39 | 37.29 | 52.10 | 19.51 | 72.12 | 30.71 | 41.40 | 18.81 | 18.81 | 13.83 | 25.58 |
| PostCite | R | 90.51 | 1.88 | 1.89 | 2.66 | 2.21 | 14.20 | 53.33 | 22.43 | 93.11 | 66.20 | 77.38 | 49.91 | 1.53 | 1.53 | 1.53 | 17.88 |
| PostAttr | R | 90.51 | 1.88 | 1.89 | 2.66 | 2.21 | 14.20 | 53.33 | 22.43 | 93.11 | 66.20 | 77.38 | 49.91 | 0.17 | 0.17 | 0.17 | 17.43 |
| Self-RAG | R | 100.00 | 30.82 | 39.87 | 61.96 | 48.52 | 0.00 | 0.00 | 0.00 | 100.00 | 64.35 | 78.31 | 39.15 | 66.42 | 73.52 | 69.79 | 52.49 |
| ICL | D | 97.57 | 33.31 | 40.57 | 62.35 | 49.16 | 5.03 | 73.91 | 9.42 | 99.02 | 65.30 | 78.70 | 44.06 | 7.22 | 13.25 | 9.35 | 34.19 |
| PostCite | D | 89.77 | 0.06 | 0.03 | 0.04 | 0.04 | 15.09 | 52.58 | 23.45 | 92.46 | 66.27 | 77.21 | 50.33 | 0.00 | 0.00 | 0.00 | 16.79 |
| PostAttr | D | 89.24 | 0.06 | 0.03 | 0.04 | 0.04 | 16.57 | 54.90 | 25.45 | 92.46 | 66.67 | 77.47 | 51.46 | 0.00 | 0.00 | 0.00 | 17.17 |
| Self-RAG | D | 97.68 | 31.36 | 40.53 | 61.73 | 48.93 | 3.85 | 42.11 | 7.22 | 98.52 | 64.90 | 78.26 | 42.74 | 58.31 | 69.44 | 63.39 | 51.69 |
| **LLaMA-3.2-1b** | | | | | | | | | | | | | | | | | |
| ICL | R | 60.23 | 33.07 | 37.17 | 34.80 | 35.95 | 41.12 | 36.87 | 38.88 | 60.98 | 65.15 | 63.00 | 50.94 | 7.92 | 13.42 | 9.96 | 32.28 |
| PostCite | R | 43.57 | 1.04 | 0.73 | 0.49 | 0.59 | 59.76 | 37.76 | 46.28 | 45.41 | 67.07 | 54.15 | 50.22 | 0.24 | 0.24 | 0.24 | 17.02 |
| PostAttr | R | 45.78 | 0.76 | 0.58 | 0.41 | 0.48 | 54.44 | 35.80 | 43.19 | 45.90 | 64.52 | 53.64 | 48.42 | 0.00 | 0.00 | 0.00 | 16.30 |
| FRONT | R | 79.11 | 27.23 | 43.72 | 53.76 | 48.22 | 27.51 | 46.97 | 34.70 | 82.79 | 67.33 | 74.26 | 54.48 | 47.15 | 49.48 | 48.29 | 50.33 |
| SFT | R | 63.82 | 21.19 | 45.80 | 45.42 | 45.61 | 54.14 | 53.35 | 53.74 | 74.38 | 74.07 | 74.22 | 59.73 | 69.13 | 73.10 | 71.08 | 60.87 |
| TRUST-ALIGN (DPO) | R | 41.67 | 15.06 | 49.16 | 31.83 | 38.64 | 73.96 | 45.21 | 56.12 | 50.33 | 77.72 | 61.09 | 58.61 | 80.93 | 77.84 | 79.35 | 58.87 |
| TRUST-ALIGN (DPO-25%) | R | 51.69 | 17.93 | 49.80 | 40.01 | 44.37 | 67.46 | 49.78 | 57.55 | 62.30 | 77.55 | 69.09 | 63.19 | 80.41 | 77.68 | 79.02 | 62.19 |
| **LLaMA-3.2-3b** | | | | | | | | | | | | | | | | | |
| ICL | R | 1.27 | 0.63 | 52.78 | 1.04 | 2.04 | 99.41 | 32.75 | 49.27 | 1.64 | 83.33 | 3.22 | 27.98 | 53.47 | 54.44 | 53.95 | 27.99 |
| PostCite | R | 47.26 | 16.56 | 36.64 | 26.91 | 31.03 | 63.91 | 43.20 | 51.55 | 53.44 | 72.77 | 61.63 | 56.59 | 22.99 | 22.99 | 22.99 | 36.87 |
| PostAttr | R | 47.15 | 15.76 | 35.18 | 25.78 | 29.76 | 64.20 | 43.31 | 51.68 | 53.44 | 72.93 | 61.68 | 56.59 | 4.69 | 4.69 | 4.69 | 30.39 |
| FRONT | R | 95.25 | 40.68 | 52.94 | 78.37 | 63.19 | 10.95 | 82.22 | 19.32 | 98.69 | 66.67 | 79.58 | 49.45 | 60.04 | 55.09 | 57.46 | 56.70 |
| SFT | R | 68.04 | 23.75 | 47.89 | 50.64 | 49.23 | 51.48 | 57.43 | 54.29 | 78.85 | 74.57 | 76.65 | 65.47 | 80.23 | 71.52 | 75.63 | 63.44 |
| TRUST-ALIGN (DPO) | R | 77.85 | 31.81 | 54.63 | 62.84 | 58.42 | 42.31 | 68.10 | 52.19 | 89.02 | 73.58 | 80.56 | 66.38 | 85.00 | 83.43 | 84.21 | 70.14 |
| TRUST-ALIGN (DPO-25%) | R | 62.45 | 23.28 | 50.74 | 49.24 | 49.98 | 56.80 | 53.93 | 55.33 | 73.11 | 75.34 | 74.21 | 64.77 | 83.95 | 77.08 | 80.37 | 65.04 |
| **LLaMA-3-8b** | | | | | | | | | | | | | | | | | |
| ICL | R | 1.48 | 0.69 | 67.14 | 1.54 | 3.01 | 99.70 | 36.08 | 52.99 | 2.13 | 92.86 | 4.17 | 28.58 | 92.86 | 80.95 | 86.50 | 39.36 |
| PostCite | R | 77.53 | 22.15 | 30.17 | 36.36 | 32.98 | 27.51 | 43.66 | 33.76 | 80.33 | 66.67 | 72.86 | 53.31 | 28.01 | 28.01 | 28.01 | 38.10 |
| PostAttr | R | 77.53 | 22.15 | 30.17 | 36.36 | 32.98 | 27.51 | 43.66 | 33.76 | 80.33 | 66.67 | 72.86 | 53.31 | 5.95 | 5.95 | 5.95 | 30.75 |
| ICL | D | 89.66 | 36.41 | 49.83 | 70.17 | 58.28 | 20.41 | 70.41 | 31.65 | 95.25 | 68.35 | 79.59 | 55.62 | 61.40 | 61.77 | 61.59 | 58.50 |
| PostCite | D | 97.26 | 27.65 | 28.91 | 43.69 | 34.80 | 4.73 | 61.54 | 8.79 | 98.36 | 65.08 | 78.33 | 43.56 | 17.89 | 17.89 | 17.89 | 32.08 |
| PostAttr | D | 97.47 | 27.65 | 28.84 | 43.69 | 34.75 | 4.14 | 58.33 | 7.73 | 98.36 | 64.94 | 78.23 | 42.98 | 3.18 | 3.18 | 3.18 | 26.97 |
| FRONT | D | 99.05 | 41.78 | 51.34 | 79.03 | 62.25 | 2.37 | 88.89 | 4.60 | 99.84 | 64.86 | 78.63 | 41.62 | 65.89 | 66.40 | 66.14 | 56.67 |
| SFT | R | 68.99 | 25.22 | 50.59 | 54.24 | 52.35 | 51.18 | 58.84 | 54.75 | 80.16 | 74.77 | 77.37 | 66.06 | 86.09 | 76.38 | 80.95 | 66.45 |
| TRUST-ALIGN (DPO) | R | 56.43 | 23.53 | 57.72 | 50.63 | 53.94 | 64.79 | 53.03 | 58.32 | 68.30 | 77.76 | 72.66 | 65.49 | 88.93 | 87.60 | 88.26 | 69.23 |
| TRUST-ALIGN (DPO-25%) | R | 56.75 | 25.06 | 60.15 | 53.05 | 56.38 | 64.20 | 52.93 | 58.02 | 68.36 | 77.51 | 72.65 | 65.33 | 85.64 | 84.71 | 85.17 | 68.96 |
| **Closed-source Models** | | | | | | | | | | | | | | | | | |
| GPT-3.5 | R | 71.20 | 27.30 | 50.36 | 55.72 | 52.91 | 48.82 | 60.44 | 54.01 | 82.30 | 74.37 | 78.13 | 66.07 | 84.66 | 83.24 | 83.94 | 67.64 |
| GPT-4 | R | 86.81 | 37.93 | 54.81 | 73.95 | 62.96 | 28.99 | 78.40 | 42.35 | 95.57 | 70.84 | 81.37 | 61.85 | 85.82 | 82.93 | 84.35 | 69.72 |
| Claude-3.5 | R | 84.60 | 36.29 | 52.79 | 69.41 | 59.97 | 34.02 | 78.77 | 47.52 | 94.92 | 72.19 | 82.01 | 64.77 | 67.29 | 69.43 | 68.35 | 64.36 |
| GPT-3.5 | D | 94.41 | 34.67 | 46.27 | 67.88 | 55.03 | 14.20 | 90.57 | 24.55 | 99.18 | 67.60 | 80.40 | 52.48 | 78.13 | 77.95 | 78.04 | 61.85 |
| GPT-4 | D | 92.72 | 41.13 | 52.58 | 76.65 | 62.37 | 16.86 | 82.61 | 28.01 | 98.03 | 68.03 | 80.32 | 54.17 | 79.48 | 79.92 | 79.70 | 65.41 |
| Claude-3.5 | D | 82.49 | 32.68 | 47.64 | 62.86 | 54.20 | 37.87 | 77.11 | 50.79 | 93.77 | 73.15 | 82.18 | 66.49 | 57.41 | 60.44 | 58.88 | 59.86 |

Table 20: Detailed QAMPARI results.

| | Prompt | AR% | AC$_{reg}$ | P$_{AC}$ | R$_{AC}$ | F1$_{AC}$ | R$_{ref}$ | P$_{ref}$ | F1$_{ref}$ | R$_{ans}$ | P$_{ans}$ | F1$_{ans}$ | F1$_{GR}$ | R$_{cite}$ | P$_{cite}$ | F1$_{GC}$ | TRUST-SCORE |
|---|---|---|---|---|---|---|---|---|---|---|---|---|---|---|---|---|---|
| **Qwen-2.5-0.5b** | | | | | | | | | | | | | | | | | |
| ICL | R | 11.40 | 4.39 | 1.70 | 2.45 | 90.07 | 71.67 | 79.82 | 14.92 | 38.60 | 21.52 | 50.67 | | 0.00 | 0.00 | 0.00 | 17.71 |
| PostCite | R | 17.00 | 0.92 | 0.53 | 0.67 | 84.82 | 72.05 | 77.92 | 21.36 | 37.06 | 27.10 | 52.51 | | 5.72 | 5.72 | 5.72 | 19.63 |
| PostAttr | R | 17.00 | 0.92 | 0.53 | 0.67 | 84.82 | 72.05 | 77.92 | 21.36 | 37.06 | 27.10 | 52.51 | | 0.90 | 0.90 | 0.90 | 18.03 |
| FRONT | R | 99.30 | 7.47 | 25.14 | 11.52 | 0.57 | 57.14 | 1.12 | 98.98 | 29.41 | 45.34 | 23.23 | | 15.08 | 16.82 | 15.90 | 16.88 |
| SFT | R | 18.50 | 20.78 | 13.03 | 16.02 | 87.80 | 75.95 | 81.45 | 33.56 | 53.51 | 41.25 | 61.35 | | 27.35 | 28.31 | 27.82 | 35.06 |
| TRUST-ALIGN (DPO) | R | 17.90 | 20.86 | 12.66 | 15.76 | 88.65 | 76.13 | 81.91 | 33.56 | 55.31 | 41.77 | 61.84 | | 29.70 | 29.75 | 29.73 | 35.78 |
| TRUST-ALIGN (DPO-25%) | R | 17.40 | 23.95 | 14.13 | 17.77 | 88.94 | 75.91 | 81.91 | 32.54 | 55.17 | 40.94 | 61.42 | | 31.98 | 32.36 | 32.17 | 37.12 |
| **Qwen-2.5-1.5b** | | | | | | | | | | | | | | | | | |
| ICL | R | 85.00 | 10.51 | 30.28 | 15.61 | 19.86 | 93.33 | 32.75 | 96.61 | 33.53 | 49.78 | 41.27 | | 8.47 | 8.76 | 8.61 | 21.83 |
| PostCite | R | 11.20 | 6.24 | 2.37 | 3.44 | 90.50 | 71.85 | 80.10 | 15.25 | 40.18 | 22.11 | 51.11 | | 13.95 | 13.95 | 13.95 | 22.83 |
| PostAttr | R | 11.20 | 6.24 | 2.37 | 3.44 | 90.50 | 71.85 | 80.10 | 15.25 | 40.18 | 22.11 | 51.11 | | 1.07 | 1.07 | 1.07 | 18.54 |
| FRONT | R | 98.80 | 10.42 | 34.88 | 16.05 | 1.56 | 91.67 | 3.07 | 99.66 | 29.76 | 45.83 | 24.45 | | 10.34 | 13.20 | 11.60 | 17.37 |
| SFT | R | 25.50 | 25.77 | 22.27 | 23.89 | 85.67 | 81.07 | 83.31 | 52.20 | 60.39 | 56.00 | 69.66 | | 37.41 | 37.96 | 37.68 | 43.74 |
| DPO | R | 20.00 | 29.45 | 19.97 | 23.80 | 90.07 | 79.38 | 84.39 | 44.07 | 65.00 | 52.53 | 68.46 | | 50.79 | 51.17 | 50.98 | 47.75 |
| TRUST-ALIGN (DPO-25%) | R | 18.40 | 36.89 | 23.01 | 28.34 | 91.49 | 79.04 | 84.81 | 42.03 | 67.39 | 51.77 | 68.29 | | 51.65 | 52.25 | 51.95 | 49.53 |
| **Qwen-2.5-3b** | | | | | | | | | | | | | | | | | |
| ICL | R | 22.30 | 26.91 | 20.34 | 23.17 | 85.11 | 77.22 | 80.97 | 40.00 | 52.91 | 45.56 | 63.27 | | 40.21 | 42.25 | 41.20 | 42.55 |
| PostCite | R | 0.10 | 0.00 | 0.00 | 0.00 | 99.86 | 70.47 | 82.63 | 0.00 | 0.00 | 0.00 | 41.31 | | 0.00 | 0.00 | 0.00 | 13.77 |
| PostAttr | R | 0.10 | 0.00 | 0.00 | 0.00 | 99.86 | 70.47 | 82.63 | 0.00 | 0.00 | 0.00 | 41.31 | | 25.00 | 25.00 | 25.00 | 22.10 |
| FRONT | R | 79.10 | 14.20 | 38.07 | 20.69 | 28.65 | 96.65 | 44.20 | 97.63 | 36.41 | 53.04 | 48.62 | | 25.21 | 26.16 | 25.67 | 31.66 |
| SFT | R | 27.20 | 30.02 | 27.68 | 28.80 | 83.26 | 80.63 | 81.93 | 52.20 | 56.62 | 54.32 | 68.12 | | 37.12 | 37.56 | 37.34 | 44.75 |
| TRUST-ALIGN (DPO) | R | 48.10 | 28.79 | 46.94 | 35.69 | 66.81 | 90.75 | 76.96 | 83.73 | 51.35 | 63.66 | 70.31 | | 45.51 | 45.77 | 45.64 | 50.55 |
| TRUST-ALIGN (DPO-25%) | R | 17.50 | 40.53 | 24.04 | 30.18 | 92.34 | 78.91 | 85.10 | 41.02 | 69.14 | 51.49 | 68.29 | | 47.67 | 47.74 | 47.71 | 48.73 |
| **Qwen-2.5-7b** | | | | | | | | | | | | | | | | | |
| ICL | R | 56.30 | 22.04 | 42.05 | 28.92 | 55.74 | 89.93 | 68.83 | 85.08 | 44.58 | 58.51 | 63.67 | | 38.82 | 39.76 | 39.28 | **43.96** |
| PostCite | R | 26.70 | 9.04 | 8.18 | 8.59 | 79.15 | 76.13 | 77.61 | 40.68 | 44.94 | 42.70 | 60.16 | | 1.05 | 1.05 | 1.05 | 23.27 |
| PostAttr | R | 26.70 | 9.04 | 8.18 | 8.59 | 79.15 | 76.13 | 77.61 | 40.68 | 44.94 | 42.70 | 60.16 | | 13.55 | 13.55 | 13.55 | 27.43 |
| FRONT | R | 84.70 | 11.47 | 32.94 | 17.02 | 21.13 | 97.39 | 34.73 | 98.64 | 34.36 | 50.96 | 42.85 | | 24.48 | 24.48 | 24.48 | 28.12 |
| SFT | R | 31.70 | 32.41 | 34.83 | 33.58 | 80.43 | 83.02 | 81.70 | 60.68 | 56.47 | 58.50 | 70.10 | | 48.93 | 49.23 | 49.08 | 50.92 |
| DPO | R | 32.10 | 28.89 | 31.44 | 30.11 | 80.43 | 83.51 | 81.94 | 62.03 | 57.01 | 59.42 | 70.68 | | 53.48 | 53.48 | 53.48 | 51.42 |
| TRUST-ALIGN (DPO-25%) | R | 29.00 | 32.02 | 31.48 | 31.75 | 83.83 | 83.24 | 83.53 | 59.66 | 60.69 | 60.17 | 71.85 | | 55.71 | 56.15 | 55.93 | 53.18 |
| **Phi3.5-mini** | | | | | | | | | | | | | | | | | |
| ICL | R | 70.20 | 8.46 | 20.14 | 11.92 | 31.35 | 74.16 | 44.07 | 73.90 | 31.05 | 43.73 | 43.90 | | 10.66 | 14.43 | 12.26 | 22.69 |
| PostCite | R | 76.90 | 2.47 | 6.45 | 3.57 | 25.67 | 78.35 | 38.68 | 83.05 | 31.86 | 46.05 | 42.36 | | 4.49 | 4.49 | 4.49 | 16.81 |
| PostAttr | R | 76.90 | 2.47 | 6.45 | 3.57 | 25.67 | 78.35 | 38.68 | 83.05 | 31.86 | 46.05 | 42.36 | | 0.46 | 0.46 | 0.46 | 15.46 |
| FRONT | R | 100.00 | 7.75 | 26.27 | 11.97 | 0.00 | 0.00 | 0.00 | 100.00 | 29.50 | 45.56 | 22.78 | | 19.15 | 24.49 | 21.50 | 18.75 |
| SFT | R | 29.10 | 35.28 | 34.80 | 35.04 | 84.96 | 84.49 | 84.72 | 62.71 | 63.57 | 63.14 | 73.93 | | 49.13 | 49.63 | 49.38 | 52.78 |
| DPO | R | 30.10 | 36.06 | 36.79 | 36.42 | 84.11 | 84.84 | 84.47 | 64.07 | 62.79 | 63.42 | 73.95 | | 53.26 | 53.55 | 53.40 | 54.59 |
| TRUST-ALIGN (DPO-25%) | R | 30.10 | 35.32 | 36.04 | 35.68 | 84.82 | 85.55 | 85.19 | 65.76 | 64.45 | 65.10 | 75.14 | | 52.21 | 52.82 | 52.52 | 54.45 |
| **LLaMA-2-7b** | | | | | | | | | | | | | | | | | |
| ICL | R | 0.00 | 0.00 | 0.00 | 0.00 | 100.00 | 70.50 | 82.70 | 0.00 | 0.00 | 0.00 | 41.35 | | 0.00 | 0.00 | 0.00 | 13.78 |
| PostCite | R | 34.40 | 0.00 | 0.00 | 0.00 | 70.21 | 75.46 | 72.74 | 45.42 | 38.95 | 41.94 | 57.34 | | 9.50 | 9.50 | 9.50 | 22.28 |
| PostAttr | R | 34.40 | 0.00 | 0.00 | 0.00 | 70.21 | 75.46 | 72.74 | 45.42 | 38.95 | 41.94 | 57.34 | | 3.78 | 3.78 | 3.78 | 21.44 |
| Self-RAG | R | 96.00 | 4.45 | 14.49 | 6.81 | 5.25 | 92.50 | 9.93 | 98.98 | 30.42 | 46.53 | 28.23 | | 17.92 | 22.50 | 19.95 | 18.33 |
| FRONT | R | 100.00 | 11.18 | 37.89 | 17.27 | 0.00 | 0.00 | 0.00 | 100.00 | 29.50 | 45.56 | 22.78 | | 24.20 | 24.32 | 24.26 | 21.44 |
| ICL | D | 93.60 | 5.49 | 17.51 | 8.36 | 8.23 | 90.63 | 15.08 | 97.97 | 30.88 | 46.95 | 31.02 | | 3.83 | 3.93 | 3.88 | 14.42 |
| PostCite | D | 56.30 | 0.00 | 0.00 | 0.00 | 45.67 | 73.68 | 56.39 | 61.02 | 31.97 | 41.96 | 49.18 | | 7.73 | 7.73 | 7.73 | 18.97 |
| PostAttr | D | 51.10 | 0.00 | 0.00 | 0.00 | 50.21 | 72.39 | 59.30 | 54.24 | 31.31 | 39.70 | 49.50 | | 4.70 | 4.70 | 4.70 | 18.07 |
| Self-RAG | D | 96.20 | 5.04 | 16.45 | 7.72 | 4.40 | 81.58 | 8.34 | 97.63 | 29.94 | 45.82 | 27.08 | | 13.25 | 18.50 | 15.44 | 16.75 |
| SFT | R | 31.60 | 32.63 | 34.96 | 33.76 | 81.13 | 83.63 | 82.36 | 62.03 | 57.91 | 59.90 | 71.13 | | 46.25 | 46.49 | 46.37 | 50.42 |
| TRUST-ALIGN (DPO) | R | 32.30 | 30.64 | 33.55 | 32.03 | 80.85 | 84.19 | 82.49 | 63.73 | 58.20 | 60.84 | 71.67 | | 49.34 | 49.50 | 49.42 | 51.04 |
| TRUST-ALIGN (DPO-25%) | R | 31.50 | 32.50 | 34.70 | 33.56 | 81.13 | 83.50 | 82.30 | 61.69 | 57.78 | 59.67 | 70.99 | | 52.01 | 52.19 | 52.10 | 52.22 |
| **LLaMA-2-13b** | | | | | | | | | | | | | | | | | |
| ICL | R | 26.50 | 0.47 | 0.42 | 0.44 | 79.01 | 75.78 | 77.36 | 39.66 | 44.15 | 41.79 | 59.57 | | 0.00 | 0.00 | 0.00 | 20.00 |
| PostCite | R | 100.00 | 0.00 | 0.00 | 0.00 | 0.00 | 0.00 | 0.00 | 100.00 | 29.50 | 45.56 | 22.78 | | 8.05 | 8.05 | 8.05 | 10.28 |
| PostAttr | R | 100.00 | 0.00 | 0.00 | 0.00 | 0.00 | 0.00 | 0.00 | 100.00 | 29.50 | 45.56 | 22.78 | | 2.95 | 2.95 | 2.95 | 8.58 |
| Self-RAG | R | 72.70 | 1.90 | 4.69 | 2.71 | 32.91 | 84.98 | 47.44 | 86.10 | 34.94 | 49.71 | 48.58 | | 25.73 | 28.20 | 26.91 | **26.07** |
| ICL | D | 97.80 | 0.00 | 0.00 | 0.00 | 3.12 | 100.00 | 6.05 | 100.00 | 30.16 | 46.35 | 26.20 | | 0.00 | 0.00 | 0.00 | 8.73 |
| PostCite | D | 63.00 | 0.00 | 0.00 | 0.00 | 39.01 | 74.32 | 51.16 | 67.80 | 31.75 | 43.24 | 47.20 | | 7.14 | 7.14 | 7.14 | 18.11 |
| PostAttr | D | 58.50 | 0.00 | 0.00 | 0.00 | 43.69 | 74.22 | 55.00 | 63.73 | 32.14 | 42.73 | 48.86 | | 4.56 | 4.56 | 4.56 | 17.81 |
| Self-RAG | D | 96.30 | 2.39 | 7.80 | 3.66 | 4.40 | 83.78 | 8.36 | 97.97 | 30.01 | 45.95 | 27.15 | | 19.46 | 22.95 | 21.06 | 17.29 |
| **LLaMA-3.2-1b** | | | | | | | | | | | | | | | | | |
| ICL | R | 19.20 | 8.01 | 5.22 | 6.32 | 82.55 | 72.03 | 76.93 | 23.39 | 35.94 | 28.34 | 52.64 | | 0.26 | 0.71 | 0.38 | 19.78 |
| PostCite | R | 41.20 | 0.27 | 0.38 | 0.32 | 59.15 | 70.92 | 64.50 | 42.03 | 30.10 | 35.08 | 49.79 | | 1.61 | 1.61 | 1.61 | 17.24 |
| PostAttr | R | 34.00 | 0.59 | 0.67 | 0.63 | 65.11 | 69.55 | 67.25 | 31.86 | 29.61 | | 48.43 | | 0.21 | 0.21 | 0.21 | 16.42 |
| FRONT | R | 98.60 | 4.92 | 16.44 | 7.57 | 1.70 | 85.71 | 3.34 | 99.32 | 29.72 | 45.75 | 24.54 | | 14.77 | 15.91 | 15.32 | 15.81 |
| SFT | R | 26.00 | 29.87 | 26.32 | 27.98 | 84.40 | 80.41 | 82.35 | 57.69 | 54.05 | 68.20 | | | 37.31 | 38.01 | 37.96 | 44.71 |
| DPO | R | 20.00 | 33.69 | 22.84 | 27.22 | 89.79 | 79.12 | 84.12 | 43.39 | 64.00 | 51.72 | 67.92 | | 49.42 | 49.42 | 49.42 | 48.19 |
| TRUST-ALIGN (DPO-25%) | R | 21.90 | 34.83 | 25.85 | 29.68 | 89.22 | 80.54 | 84.66 | 48.47 | 65.30 | 55.64 | 70.15 | | 49.31 | 49.31 | 49.31 | 49.71 |
| **LLaMA-3.2-3b** | | | | | | | | | | | | | | | | | |
| ICL | R | 34.10 | 14.98 | 17.31 | 16.06 | 71.91 | 76.93 | 74.34 | 48.47 | 41.94 | 44.97 | 59.65 | | 12.19 | 13.63 | 12.87 | 29.53 |
| PostCite | R | 39.60 | 5.53 | 7.43 | 6.34 | 64.11 | 74.83 | 69.06 | 48.47 | 36.11 | 41.39 | 55.22 | | 6.83 | 6.83 | 6.83 | 22.80 |
| PostAttr | R | 42.00 | 4.34 | 6.17 | 5.10 | 60.99 | 74.14 | 66.93 | 34.52 | 40.56 | | 53.74 | | 0.27 | 0.27 | 0.27 | 19.70 |
| FRONT | R | 92.70 | 8.56 | 26.90 | 12.99 | 9.93 | 95.89 | 17.99 | 98.98 | 31.50 | 47.79 | 32.89 | | 18.47 | 19.96 | 19.19 | 21.69 |
| SFT | R | 27.60 | 29.06 | 27.19 | 28.09 | 84.11 | 81.91 | 83.00 | 55.59 | 59.42 | 57.44 | 70.22 | | 37.83 | 38.24 | 38.03 | 45.45 |
| DPO | R | 48.20 | 23.48 | 38.36 | 29.13 | 67.09 | 91.31 | 77.35 | 84.75 | 51.87 | 64.35 | 70.85 | | 45.43 | 45.88 | 45.65 | 48.54 |
| TRUST-ALIGN (DPO-25%) | R | 18.30 | 39.01 | 24.20 | 29.87 | 91.49 | 78.95 | 84.76 | 41.69 | 67.21 | 51.46 | 68.11 | | 49.79 | 50.15 | 49.97 | 49.32 |
| **LLaMA-3-8b** | | | | | | | | | | | | | | | | | |
| ICL | R | 3.90 | 25.36 | 3.35 | 5.92 | 97.87 | 71.80 | 82.83 | 8.14 | 61.54 | 14.37 | 48.60 | | 17.22 | 24.53 | 20.24 | 24.92 |
| PostCite | R | 87.00 | 4.08 | 12.05 | 6.10 | 14.04 | 76.15 | 23.71 | 89.49 | 30.34 | 45.32 | 34.52 | | 8.42 | 8.42 | 8.42 | 16.35 |
| PostAttr | R | 87.00 | 4.08 | 12.05 | 6.10 | 14.04 | 76.15 | 23.71 | 89.49 | 30.34 | 45.32 | 34.52 | | 1.64 | 1.64 | 1.64 | 14.09 |
| ICL | D | 70.80 | 4.12 | 9.89 | 5.82 | 35.60 | 85.96 | 50.35 | 86.10 | 35.88 | 50.65 | 50.50 | | 4.45 | 5.23 | 4.81 | 20.38 |
| PostCite | D | 92.00 | 1.60 | 5.24 | 2.45 | 8.37 | 73.75 | 15.03 | 92.88 | 29.78 | 45.10 | 30.07 | | 11.14 | 11.14 | 11.14 | 14.55 |
| PostAttr | D | 93.00 | 1.58 | 5.24 | 2.43 | 7.80 | 78.57 | 14.19 | 94.92 | 30.11 | 45.71 | 29.95 | | 5.65 | 5.65 | 5.65 | 12.68 |
| FRONT | R | 100.00 | 8.76 | 29.68 | 13.53 | 0.00 | 0.00 | 0.00 | 100.00 | 29.50 | 45.56 | 22.78 | | 20.77 | 20.07 | 20.42 | 18.91 |
| SFT | R | 24.20 | 37.56 | 30.81 | 33.85 | 87.66 | 81.53 | 84.48 | 52.54 | 64.05 | 57.73 | 71.11 | | 47.78 | 48.24 | 48.01 | 50.99 |
| TRUST-ALIGN (DPO) | R | 22.40 | 40.95 | 31.09 | 35.35 | 89.08 | 80.93 | 84.81 | 49.83 | 65.63 | 56.65 | 70.73 | | 58.58 | 58.96 | 58.77 | **54.95** |
| TRUST-ALIGN (DPO-25%) | R | 24.40 | 43.56 | 36.03 | 39.44 | 88.37 | 82.41 | 85.28 | 54.92 | 66.39 | 60.11 | 72.70 | | 61.03 | 61.10 | 61.06 | 57.73 |
| **Closed-source Models** | | | | | | | | | | | | | | | | | |
| GPT-3.5 | R | 65.30 | 19.29 | 42.69 | 26.57 | 45.25 | 91.93 | 60.65 | 90.51 | 40.89 | 56.33 | 58.49 | | 30.75 | 32.92 | 31.80 | 38.95 |
| GPT-4 | R | 73.40 | 21.12 | 52.55 | 30.13 | 37.30 | 98.87 | 54.17 | 98.98 | 39.78 | 56.75 | 55.46 | | 34.44 | 36.51 | 35.45 | **40.35** |
| Claude-3.5 | R | 69.80 | 20.20 | 47.79 | 28.40 | 41.70 | 97.35 | 58.39 | 97.29 | 41.12 | 57.80 | 58.10 | | 32.23 | 33.46 | 32.83 | 39.78 |
| GPT-3.5 | D | 94.50 | 13.32 | 42.68 | 20.30 | 6.81 | 87.27 | 12.63 | 97.63 | 30.48 | 46.45 | 29.54 | | 20.97 | 21.47 | 21.22 | 23.69 |
| GPT-4 | D | 87.70 | 17.50 | 52.01 | 26.19 | 17.45 | 100.00 | 29.71 | 100.00 | 33.64 | 50.34 | 40.03 | | 29.66 | 30.39 | 30.02 | 32.08 |
| Claude-3.5 | D | 69.90 | 0.00 | 0.00 | 0.00 | 41.13 | 96.35 | 57.65 | 96.27 | 40.63 | 57.14 | 57.40 | | 0.00 | 0.00 | 0.00 | 19.13 |

Table 21: Detailed ELI5 results.

| | Prompt | AR% | AC$_{reg}$ | P$_{AC}$ | R$_{AC}$ | F1$_{AC}$ | R$_{ref}$ | P$_{ref}$ | F1$_{ref}$ | R$_{ans}$ | P$_{ans}$ | F1$_{ans}$ | F1$_{GR}$ | R$_{cite}$ | P$_{cite}$ | F1$_{GC}$ | TRUST-SCORE |
|---|---|---|---|---|---|---|---|---|---|---|---|---|---|---|---|---|---|
| | | | | | | | **Qwen-2.5-0.5b** | | | | | | | | | | |
| ICL | R | 82.30 | 5.40 | 8.59 | 34.14 | 13.73 | 19.04 | 85.31 | 31.13 | 87.44 | 21.99 | 35.15 | 33.14 | 0.30 | 0.49 | 0.37 | 15.75 |
| PostCite | R | 89.80 | 7.10 | 6.07 | 26.33 | 9.87 | 10.97 | 85.29 | 19.44 | 92.75 | 21.38 | 34.75 | 27.10 | 4.10 | 4.10 | 4.10 | 13.69 |
| PostAttr | R | 89.80 | 7.10 | 6.07 | 26.33 | 9.87 | 10.97 | 85.29 | 19.44 | 92.75 | 21.38 | 34.75 | 27.10 | 0.68 | 0.68 | 0.68 | 12.55 |
| FRONT | R | 99.90 | 6.93 | 8.29 | 40.02 | 13.74 | 0.13 | 100.00 | 0.25 | 100.00 | 20.72 | 34.33 | 17.29 | 29.86 | 26.28 | 27.95 | 19.66 |
| SFT | R | 35.50 | 2.23 | 8.31 | 14.25 | 10.50 | 68.85 | 84.65 | 75.94 | 52.17 | 30.42 | 38.43 | 57.19 | 22.11 | 17.56 | 19.57 | 29.09 |
| TRUST-ALIGN (DPO) | R | 21.70 | 1.83 | 13.36 | 14.01 | 13.68 | 82.85 | 83.91 | 83.38 | 39.13 | 37.33 | 38.21 | 60.79 | 23.89 | 21.66 | 22.72 | 32.40 |
| TRUST-ALIGN (DPO-25%) | R | 26.40 | 1.70 | 9.09 | 11.59 | 10.19 | 78.31 | 84.38 | 81.23 | 44.44 | 34.85 | 39.07 | 60.15 | 25.60 | 24.07 | 24.81 | 31.72 |
| | | | | | | | **Qwen-2.5-1.5b** | | | | | | | | | | |
| ICL | R | 99.40 | 12.00 | 12.42 | 59.66 | 20.56 | 0.63 | 83.33 | 1.25 | 99.52 | 20.72 | 34.30 | 17.78 | 5.07 | 4.90 | 4.99 | 14.44 |
| PostCite | R | 91.50 | 12.23 | 9.58 | 42.35 | 15.63 | 9.84 | 91.76 | 17.77 | 96.62 | 21.86 | 35.65 | 26.71 | 5.17 | 5.17 | 5.17 | 15.84 |
| PostAttr | R | 91.50 | 12.23 | 9.58 | 42.35 | 15.63 | 9.84 | 91.76 | 17.77 | 96.62 | 21.86 | 35.65 | 26.71 | 0.62 | 0.62 | 0.62 | 14.32 |
| FRONT | R | 99.90 | 9.63 | 11.81 | 57.00 | 19.57 | 0.13 | 100.00 | 0.25 | 100.00 | 20.72 | 34.33 | 17.29 | 37.08 | 38.35 | 37.70 | 24.85 |
| SFT | R | 41.30 | 3.50 | 10.61 | 21.18 | 14.14 | 62.93 | 85.01 | 72.32 | 57.49 | 28.81 | 38.39 | 55.35 | 32.10 | 24.35 | 27.69 | 32.39 |
| TRUST-ALIGN (DPO) | R | 33.60 | 3.57 | 15.38 | 24.96 | 19.03 | 70.87 | 84.64 | 77.14 | 50.72 | 31.25 | 38.67 | 57.91 | 32.54 | 30.78 | 31.63 | 36.19 |
| TRUST-ALIGN (DPO-25%) | R | 30.40 | 3.57 | 16.67 | 24.48 | 19.83 | 74.91 | 85.34 | 79.79 | 50.72 | 34.54 | 41.10 | 60.44 | 38.43 | 37.08 | 37.75 | 39.34 |
| | | | | | | | **Qwen-2.5-3b** | | | | | | | | | | |
| ICL | R | 68.80 | 11.77 | 18.94 | 62.96 | 29.12 | 36.31 | 91.99 | 51.95 | 87.92 | 26.45 | 40.67 | 46.31 | 29.64 | 40.81 | 34.34 | 36.59 |
| PostCite | R | 49.70 | 13.73 | 15.39 | 36.96 | 21.73 | 52.08 | 82.11 | 63.73 | 56.52 | 23.54 | 33.24 | 48.49 | 7.56 | 7.56 | 7.56 | 25.93 |
| PostAttr | R | 49.70 | 13.73 | 15.39 | 36.96 | 21.73 | 52.08 | 82.11 | 63.73 | 56.52 | 23.54 | 33.24 | 48.49 | 1.31 | 1.31 | 1.31 | 23.84 |
| FRONT | R | 93.60 | 8.47 | 11.41 | 51.61 | 18.69 | 7.94 | 98.44 | 14.70 | 99.52 | 22.01 | 36.05 | 25.37 | 39.00 | 35.92 | 37.40 | 27.15 |
| SFT | R | 34.50 | 2.83 | 11.88 | 19.81 | 14.85 | 71.88 | 87.02 | 78.73 | 58.94 | 35.36 | 44.20 | 61.47 | 39.81 | 32.64 | 35.87 | 37.40 |
| TRUST-ALIGN (DPO) | R | 13.50 | 2.20 | 28.52 | 18.60 | 22.52 | 91.80 | 84.16 | 87.82 | 33.82 | 51.85 | 40.94 | 64.38 | 42.96 | 41.10 | 42.01 | 42.97 |
| TRUST-ALIGN (DPO-25%) | R | 17.50 | 2.33 | 20.86 | 17.63 | 19.11 | 88.40 | 84.97 | 86.65 | 40.10 | 47.43 | 43.46 | 65.05 | 50.38 | 45.76 | 47.96 | 44.04 |
| | | | | | | | **Qwen-2.5-7b** | | | | | | | | | | |
| ICL | R | 82.70 | 12.20 | 17.67 | 70.61 | 28.27 | 21.31 | 97.69 | 34.99 | 98.07 | 24.55 | 39.26 | 37.13 | 45.61 | 42.76 | 44.13 | 36.51 |
| PostCite | R | 95.60 | 19.87 | 13.27 | 61.27 | 21.82 | 5.04 | 90.91 | 9.56 | 98.07 | 21.23 | 34.91 | 22.23 | 7.03 | 7.03 | 7.03 | 17.03 |
| PostAttr | R | 95.60 | 19.87 | 13.27 | 61.27 | 21.82 | 5.04 | 90.91 | 9.56 | 98.07 | 21.23 | 34.91 | 22.23 | 0.96 | 0.96 | 0.96 | 15.00 |
| FRONT | R | 57.60 | 7.97 | 19.21 | 53.46 | 28.27 | 49.18 | 91.98 | 64.09 | 83.57 | 30.03 | 44.19 | 54.14 | 59.31 | 54.15 | 56.61 | 46.34 |
| SFT | R | 25.50 | 3.00 | 18.82 | 23.19 | 20.78 | 80.96 | 86.17 | 83.49 | 50.24 | 40.78 | 45.02 | 64.25 | 50.56 | 43.71 | 46.89 | 43.97 |
| TRUST-ALIGN (DPO) | R | 21.00 | 3.30 | 24.13 | 24.48 | 24.30 | 84.74 | 85.06 | 84.90 | 43.00 | 42.38 | 42.69 | 63.79 | 47.30 | 46.75 | 47.02 | 45.04 |
| TRUST-ALIGN (DPO-25%) | R | 19.80 | 3.10 | 22.05 | 21.10 | 21.57 | 86.13 | 85.16 | 85.64 | 42.51 | 44.44 | 43.46 | 64.55 | 50.21 | 49.79 | 50.00 | 45.37 |
| | | | | | | | **phi3.5-mini** | | | | | | | | | | |
| ICL | R | 81.50 | 13.03 | 17.30 | 68.12 | 27.59 | 22.07 | 94.59 | 35.79 | 95.17 | 24.17 | 38.55 | 37.17 | 29.35 | 30.98 | 30.14 | 31.63 |
| PostCite | R | 84.50 | 18.73 | 12.76 | 52.09 | 20.50 | 16.27 | 83.23 | 27.22 | 87.44 | 21.42 | 34.41 | 30.81 | 4.67 | 4.67 | 4.67 | 18.66 |
| PostAttr | R | 84.50 | 18.73 | 13.23 | 54.03 | 21.26 | 16.27 | 83.23 | 27.22 | 87.44 | 21.42 | 34.41 | 30.81 | 0.68 | 0.68 | 0.68 | 17.58 |
| FRONT | R | 96.60 | 9.03 | 13.03 | 60.79 | 21.46 | 4.04 | 94.12 | 7.74 | 99.03 | 21.22 | 34.95 | 21.35 | 64.31 | 58.77 | 61.41 | 34.74 |
| SFT | R | 24.50 | 3.30 | 20.75 | 24.56 | 22.50 | 82.47 | 86.62 | 84.50 | 51.21 | 43.27 | 46.90 | 65.70 | 51.78 | 42.68 | 46.79 | 45.00 |
| TRUST-ALIGN (DPO) | R | 24.90 | 3.47 | 21.42 | 25.76 | 23.39 | 82.98 | 87.62 | 85.23 | 55.07 | 45.78 | 50.00 | 67.62 | 50.40 | 44.78 | 47.42 | 46.14 |
| TRUST-ALIGN (DPO-25%) | R | 23.50 | 3.33 | 22.34 | 25.36 | 23.76 | 86.14 | 86.14 | 83.10 | 46.53 | 42.98 | 45.70 | 65.15 | 49.08 | 41.80 | 45.15 | 44.69 |
| | | | | | | | **LLaMA-2-7b** | | | | | | | | | | |
| ICL | R | 0.50 | 2.63 | 0.00 | 0.00 | 0.00 | 100.00 | 88.70 | 2.42 | 100.00 | 4.72 | 46.71 | 0.00 | 0.00 | 0.00 | | 15.57 |
| PostCite | R | 0.90 | 6.33 | 22.22 | 0.97 | 1.86 | 99.12 | 79.31 | 88.12 | 0.97 | 22.22 | 1.85 | 44.98 | 5.04 | 5.04 | 5.04 | 17.29 |
| PostAttr | R | 0.90 | 6.33 | 22.22 | 0.97 | 1.86 | 99.12 | 79.31 | 88.12 | 0.97 | 22.22 | 1.85 | 44.98 | 0.00 | 0.00 | 0.00 | 15.61 |
| Self-RAG | R | 73.50 | 6.80 | 9.57 | 33.98 | 14.94 | 29.13 | 87.17 | 43.67 | 83.57 | 23.54 | 36.73 | 40.20 | 12.34 | 15.65 | 13.80 | 22.98 |
| FRONT | R | 100.00 | 9.57 | 13.07 | 63.12 | 21.66 | 0.00 | 0.00 | 0.00 | 100.00 | 20.70 | 34.30 | 17.15 | 52.44 | 53.01 | 52.72 | 30.51 |
| ICL | D | 95.30 | 12.03 | 12.07 | 55.56 | 19.83 | 5.55 | 93.62 | 10.48 | 98.55 | 21.41 | 35.17 | 22.82 | 15.73 | 16.92 | 16.30 | 19.65 |
| PostCite | D | 83.90 | 8.13 | 7.45 | 30.19 | 11.95 | 16.14 | 79.50 | 26.83 | 84.06 | 20.74 | 33.27 | 30.05 | 4.90 | 4.90 | 4.90 | 15.63 |
| PostAtr | D | 84.00 | 8.13 | 7.44 | 30.19 | 11.94 | 15.89 | 78.75 | 26.44 | 83.57 | 20.60 | 33.05 | 29.74 | 0.93 | 0.93 | 0.93 | 14.20 |
| Self-RAG | D | 97.90 | 8.13 | 7.97 | 37.68 | 13.16 | 2.40 | 90.48 | 4.67 | 99.03 | 20.94 | 34.57 | 19.62 | 9.01 | 12.05 | 10.31 | 14.36 |
| SFT | R | 29.50 | 3.80 | 18.36 | 26.17 | 21.58 | 77.05 | 86.67 | 81.58 | 54.59 | 38.31 | 45.02 | 63.30 | 45.25 | 35.19 | 39.59 | 41.49 |
| TRUST-ALIGN (DPO) | R | 21.60 | 3.30 | 22.07 | 23.03 | 22.54 | 83.98 | 84.95 | 84.46 | 43.00 | 41.20 | 42.08 | 63.27 | 48.46 | 46.29 | 47.35 | 44.39 |
| TRUST-ALIGN (DPO-25%) | R | 24.20 | 3.63 | 21.35 | 24.96 | 23.02 | 81.84 | 85.62 | 83.69 | 47.34 | 40.50 | 43.65 | 63.67 | 47.59 | 44.60 | 46.05 | 44.25 |
| | | | | | | | **LLaMA-2-13b** | | | | | | | | | | |
| ICL | R | 46.40 | 6.90 | 14.44 | 32.37 | 19.97 | 58.39 | 86.38 | 69.68 | 64.73 | 28.88 | 39.94 | 54.81 | 3.79 | 6.28 | 4.73 | 26.50 |
| PostCite | R | 76.60 | 2.27 | 1.44 | 5.31 | 2.27 | 25.73 | 87.18 | 39.73 | 85.51 | 23.11 | 36.38 | 38.05 | 0.72 | 0.72 | 0.72 | 13.68 |
| PostAttr | R | 76.60 | 2.27 | 1.44 | 5.31 | 2.27 | 25.73 | 87.18 | 39.73 | 85.51 | 23.11 | 36.38 | 38.05 | 0.09 | 0.09 | 0.09 | 13.47 |
| Self-RAG | R | 22.10 | 2.40 | 12.37 | 13.20 | 12.77 | 81.59 | 83.06 | 82.32 | 36.23 | 33.94 | 35.05 | 58.68 | 22.09 | 27.60 | 24.54 | 32.00 |
| ICL | D | 96.50 | 13.07 | 12.71 | 59.26 | 20.93 | 3.91 | 88.57 | 7.49 | 98.07 | 21.04 | 34.64 | 21.06 | 2.45 | 3.25 | 2.80 | 14.93 |
| PostCite | D | 7.00 | 0.57 | 7.14 | 2.43 | 3.62 | 92.18 | 78.60 | 84.85 | 3.86 | 11.43 | 5.78 | 45.31 | 4.73 | 4.73 | 4.73 | 17.89 |
| PostAttr | D | 6.70 | 0.57 | 7.46 | 2.42 | 3.66 | 93.44 | 79.42 | 85.86 | 7.25 | 22.39 | 10.95 | 48.41 | 0.71 | 0.71 | 0.71 | 17.59 |
| Self-RAG | D | 98.00 | 9.73 | 7.38 | 34.94 | 12.19 | 2.02 | 80.00 | 3.94 | 98.07 | 20.80 | 34.20 | 19.07 | 5.71 | 8.06 | 6.68 | 12.65 |
| | | | | | | | **LLaMA-3.2-1b** | | | | | | | | | | |
| ICL | R | 88.40 | 5.87 | 7.94 | 33.90 | 12.87 | 11.73 | 80.17 | 20.46 | 88.89 | 20.81 | 33.73 | 27.10 | 4.05 | 7.38 | 5.23 | 15.07 |
| PostCite | R | 18.40 | 3.43 | 2.17 | 1.93 | 2.04 | 81.97 | 79.66 | 80.80 | 19.81 | 22.28 | 20.97 | 50.88 | 1.02 | 1.02 | 1.02 | 17.98 |
| PostAttr | R | 18.40 | 3.50 | 2.17 | 1.93 | 2.04 | 81.97 | 79.66 | 80.80 | 19.81 | 22.28 | 20.97 | 50.88 | 0.07 | 0.07 | 0.07 | 17.66 |
| FRONT | R | 97.20 | 6.63 | 9.77 | 45.89 | 16.11 | 3.40 | 96.43 | 6.58 | 99.52 | 21.19 | 34.94 | 20.76 | 30.82 | 29.58 | 30.19 | 22.35 |
| SFT | R | 20.50 | 2.00 | 14.63 | 14.49 | 14.56 | 85.25 | 85.03 | 85.14 | 42.93 | 42.72 | 63.93 | 41.83 | 33.62 | 37.28 | 38.59 |
| TRUST-ALIGN (DPO) | R | 9.60 | 1.27 | 20.83 | 9.66 | 13.20 | 93.82 | 82.30 | 87.68 | 22.71 | 48.96 | 31.02 | 59.35 | 48.96 | 47.48 | 48.21 | 40.25 |
| TRUST-ALIGN (DPO-25%) | R | 13.80 | 1.63 | 19.93 | 13.29 | 15.95 | 90.67 | 83.41 | 86.89 | 30.92 | 46.38 | 37.10 | 61.99 | 42.45 | 40.12 | 41.25 | 39.73 |
| | | | | | | | **LLaMA-3.2-3b** | | | | | | | | | | |
| ICL | R | 21.90 | 3.83 | 18.04 | 19.08 | 18.55 | 80.45 | 81.69 | 81.07 | 30.92 | 29.22 | 30.05 | 55.56 | 32.21 | 29.34 | 30.70 | 34.94 |
| PostCite | R | 92.80 | 16.33 | 11.08 | 49.68 | 18.12 | 8.20 | 90.28 | 15.03 | 96.62 | 21.55 | 35.24 | 25.14 | 4.44 | 4.44 | 4.44 | 15.90 |
| PostAttr | R | 92.80 | 16.33 | 11.30 | 50.64 | 18.48 | 8.20 | 90.28 | 15.03 | 96.62 | 21.55 | 35.24 | 25.14 | 0.53 | 0.53 | 0.53 | 14.72 |
| FRONT | R | 86.90 | 7.73 | 12.35 | 51.85 | 19.95 | 15.76 | 95.42 | 27.06 | 97.10 | 23.13 | 37.36 | 32.21 | 41.90 | 42.04 | 41.97 | 31.38 |
| SFT | R | 14.70 | 1.67 | 19.16 | 13.61 | 15.92 | 90.04 | 83.70 | 86.76 | 32.85 | 46.26 | 38.42 | 62.59 | 59.18 | 48.53 | 53.33 | 43.95 |
| TRUST-ALIGN (DPO) | R | 17.50 | 2.23 | 20.00 | 16.91 | 18.33 | 87.52 | 84.12 | 85.78 | 36.71 | 43.43 | 39.79 | 62.79 | 57.14 | 54.66 | 55.87 | 45.66 |
| TRUST-ALIGN (DPO-25%) | R | 4.10 | 0.50 | 22.36 | 4.43 | 7.40 | 98.11 | 81.13 | 88.81 | 12.56 | 63.41 | 20.97 | 54.89 | 60.98 | 57.72 | 59.31 | 40.53 |
| | | | | | | | **LLaMA-3-8b** | | | | | | | | | | |
| ICL | R | 0.00 | 0.00 | 0.00 | 0.00 | 0.00 | 100.00 | 79.30 | 88.46 | 0.00 | 0.00 | 0.00 | 44.23 | 0.00 | 0.00 | 0.00 | 14.74 |
| PostCite | R | 62.00 | 10.80 | 13.87 | 41.55 | 20.80 | 40.86 | 85.26 | 55.24 | 72.95 | 24.35 | 36.52 | 45.88 | 8.06 | 8.06 | 8.06 | 24.91 |
| PostAttr | R | 62.00 | 10.80 | 13.87 | 41.55 | 20.80 | 40.86 | 85.26 | 55.24 | 72.95 | 24.35 | 36.52 | 45.88 | 1.25 | 1.25 | 1.25 | 22.64 |
| FRONT | R | 99.50 | 9.17 | 11.47 | 55.15 | 18.99 | 0.63 | 100.00 | 1.25 | 100.00 | 20.80 | 34.44 | 17.85 | 44.90 | 44.48 | 44.69 | 27.18 |
| ICL | D | 84.60 | 11.90 | 14.74 | 60.23 | 23.69 | 17.65 | 90.91 | 29.57 | 93.24 | 22.81 | 36.66 | 33.11 | 31.32 | 30.74 | 31.03 | 29.28 |
| PostCite | D | 98.90 | 17.40 | 11.49 | 54.91 | 19.00 | 1.26 | 90.91 | 2.49 | 99.52 | 20.83 | 34.45 | 18.47 | 6.33 | 6.33 | 6.33 | 14.60 |
| PostAttr | D | 98.90 | 17.40 | 11.49 | 54.91 | 19.00 | 1.13 | 81.82 | 2.24 | 99.03 | 20.73 | 34.28 | 18.26 | 1.02 | 1.02 | 1.02 | 12.76 |
| SFT | R | 23.60 | 3.27 | 21.19 | 24.15 | 22.57 | 82.98 | 86.13 | 84.52 | 48.79 | 42.80 | 45.60 | 65.06 | 51.77 | 42.79 | 46.85 | 44.83 |
| TRUST-ALIGN (DPO) | R | 15.50 | 2.77 | 24.30 | 18.20 | 20.81 | 89.66 | 84.14 | 86.81 | 35.27 | 47.10 | 40.33 | 63.57 | 50.75 | 49.74 | 50.24 | 44.87 |
| TRUST-ALIGN (DPO-25%) | R | 16.00 | 3.10 | 28.54 | 22.06 | 24.89 | 89.28 | 84.29 | 86.71 | 36.23 | 46.88 | 40.87 | 63.79 | 54.17 | 53.80 | 53.98 | 47.55 |
| | | | | | | | **Closed-source Models** | | | | | | | | | | |
| GPT-3.5 | R | 49.00 | 8.47 | 23.03 | 54.51 | 32.38 | 58.26 | 90.59 | 70.91 | 76.81 | 32.45 | 45.62 | 58.27 | 56.57 | 58.03 | 57.29 | 49.31 |
| GPT-4 | R | 61.50 | 10.50 | 22.09 | 65.62 | 33.05 | 45.65 | 94.03 | 61.46 | 88.89 | 29.92 | 44.77 | 53.11 | 61.33 | 62.35 | 61.84 | 49.33 |
| Claude-3.5 | R | 59.00 | 2.87 | 7.66 | 21.82 | 11.34 | 48.05 | 92.93 | 63.34 | 85.99 | 30.17 | 44.67 | 54.00 | 11.64 | 13.34 | 12.43 | 25.92 |
| GPT-3.5 | D | 93.50 | 14.33 | 14.58 | 65.86 | 23.88 | 7.57 | 92.31 | 13.99 | 97.58 | 21.60 | 35.38 | 24.68 | 46.46 | 46.10 | 46.28 | 31.61 |
| GPT-4 | D | 82.80 | 15.00 | 18.18 | 72.71 | 29.09 | 21.19 | 97.67 | 34.82 | 98.07 | 24.52 | 39.23 | 37.02 | 48.20 | 48.47 | 48.33 | 38.15 |
| Claude-3.5 | D | 56.60 | 3.40 | 7.89 | 21.58 | 11.56 | 51.07 | 93.32 | 66.01 | 85.99 | 31.45 | 46.05 | 56.03 | 10.22 | 12.43 | 11.22 | 26.27 |

Table 22: Detailed ExpertQA results.

| | Prompt | AR% | $AC_{reg}$ | $P_{AC}$ | $R_{AC}$ | $F1_{AC}$ | $R_{ref}$ | $P_{ref}$ | $F1_{ref}$ | $R_{ans}$ | $P_{ans}$ | $F1_{ans}$ | $F1_{GR}$ | $R_{cite}$ | $P_{cite}$ | $F1_{GC}$ | TRUST-SCORE |
|---|---|---|---|---|---|---|---|---|---|---|---|---|---|---|---|---|---|
| **Qwen-2.5-0.5b** | | | | | | | | | | | | | | | | | |
| ICL | R | 78.24 | 10.14 | 15.02 | 37.37 | 21.42 | 21.59 | 68.01 | 32.77 | 77.86 | 31.29 | 44.64 | 38.71 | 0.33 | 0.69 | 0.44 | 20.19 |
| PostCite | R | 51.41 | 7.93 | 10.73 | 17.55 | 13.32 | 48.69 | 68.69 | 56.99 | 51.61 | 31.57 | 39.18 | 48.08 | 5.53 | 5.67 | 5.6 | 22.33 |
| PostAttr | R | 51.41 | 7.93 | 10.73 | 17.55 | 13.32 | 48.69 | 68.69 | 56.99 | 51.61 | 31.57 | 39.18 | 48.08 | 1.4 | 1.58 | 1.49 | 20.96 |
| FRONT | R | 99.86 | 9.56 | 12.01 | 38.15 | 18.27 | 0.13 | 66.67 | 0.27 | 99.85 | 31.44 | 47.82 | 24.05 | 36.72 | 32.75 | 34.62 | 25.65 |
| | | | | | | | | | | | | | | | | | |
| TRUST-ALIGN (DPO) | R | 32.96 | 4.1 | 17.74 | 18.6 | 18.16 | 75.52 | 77.24 | 76.37 | 51.47 | 49.09 | 50.25 | 63.31 | 35.55 | 34.6 | 35.07 | 38.85 |
| **Qwen-2.5-1.5b** | | | | | | | | | | | | | | | | | |
| ICL | R | 98.34 | 20.3 | 20.24 | 63.29 | 30.67 | 2.08 | 86.11 | 4.07 | 99.27 | 31.74 | 48.1 | 26.09 | 6.59 | 7.22 | 6.89 | 21.22 |
| PostCite | R | 62.19 | 14.64 | 16.73 | 33.09 | 22.22 | 40.28 | 73.05 | 51.93 | 67.6 | 34.17 | 45.4 | 48.66 | 15.32 | 18.89 | 16.92 | 29.27 |
| PostAttr | R | 62.19 | 14.64 | 16.73 | 33.09 | 22.22 | 40.28 | 73.05 | 51.93 | 67.6 | 34.17 | 45.4 | 48.66 | 11.41 | 15.53 | 13.15 | 28.01 |
| FRONT | R | 99.59 | 16.23 | 19.17 | 60.73 | 29.15 | 0.61 | 100 | 1.2 | 100 | 31.57 | 47.99 | 24.6 | 49.36 | 51.12 | 50.22 | 34.66 |
| | | | | | | | | | | | | | | | | | |
| TRUST-ALIGN (DPO) | R | 30.2 | 5.71 | 25.57 | 24.56 | 25.06 | 81.24 | 79.79 | 80.51 | 55.13 | 57.4 | 56.25 | 68.38 | 52.25 | 50.65 | 51.44 | 48.29 |
| **Qwen-2.5-3b** | | | | | | | | | | | | | | | | | |
| ICL | R | 68.88 | 20.43 | 25.59 | 56.06 | 35.14 | 35.98 | 79.26 | 49.49 | 79.47 | 36.28 | 49.82 | 49.65 | 35.53 | 53.4 | 42.67 | 42.49 |
| PostCite | R | 0.05 | 6.26 | 0.00 | 0.00 | 0.00 | 99.93 | 68.54 | 81.31 | 0.00 | 0.00 | 0.00 | 40.66 | 0.00 | 0.00 | 0.00 | 13.55 |
| PostAttr | R | 0.05 | 6.26 | 0.00 | 0.00 | 0.00 | 99.93 | 68.54 | 81.31 | 0.00 | 0.00 | 0.00 | 40.66 | 0.00 | 0.00 | 0.00 | 13.55 |
| FRONT | R | 95.48 | 13.6 | 17.06 | 51.81 | 25.67 | 5.85 | 88.78 | 10.98 | 98.39 | 32.4 | 48.75 | 29.86 | 46.28 | 42.81 | 44.48 | 33.34 |
| | | | | | | | | | | | | | | | | | |
| TRUST-ALIGN (DPO) | R | 17.15 | 3.53 | 29.7 | 16.2 | 20.97 | 92.06 | 76.18 | 83.37 | 37.24 | 68.28 | 48.2 | 65.79 | 61.29 | 59.25 | 60.25 | 49 |
| **Qwen-2.5-7b** | | | | | | | | | | | | | | | | | |
| ICL | R | 84.56 | 20.18 | 24.92 | 67.01 | 36.33 | 20.24 | 89.85 | 33.04 | 95.01 | 35.33 | 51.51 | 42.28 | 55.82 | 56.36 | 56.09 | 44.9 |
| PostCite | R | 42.14 | 13.76 | 22.34 | 29.94 | 25.58 | 61.6 | 72.99 | 66.81 | 50.29 | 37.53 | 42.98 | 54.9 | 13.31 | 14.27 | 13.77 | 31.42 |
| PostAttr | R | 42.14 | 13.76 | 22.34 | 29.94 | 25.58 | 61.6 | 72.99 | 66.81 | 50.29 | 37.53 | 42.98 | 54.9 | 11.92 | 13.06 | 12.46 | 30.98 |
| FRONT | R | 65.51 | 12.36 | 23.99 | 49.98 | 32.41 | 42.77 | 85.03 | 56.91 | 83.58 | 40.11 | 54.21 | 55.56 | 70.72 | 64.28 | 67.35 | 51.77 |
| | | | | | | | | | | | | | | | | | |
| TRUST-ALIGN (DPO) | R | 24.99 | 5.51 | 28.87 | 22.95 | 25.57 | 86.48 | 79.04 | 82.59 | 50 | 62.92 | 55.72 | 69.16 | 63.38 | 62.04 | 62.7 | 52.48 |
| **phi3.5-mini** | | | | | | | | | | | | | | | | | |
| ICL | R | 85.15 | 23.12 | 25.66 | 69.5 | 37.49 | 18.43 | 85.09 | 30.29 | 92.96 | 34.33 | 50.14 | 40.22 | 35.06 | 37.28 | 36.14 | 37.95 |
| PostCite | R | 52.01 | 21.89 | 22.43 | 37.1 | 27.96 | 52.12 | 74.45 | 61.31 | 61 | 36.88 | 45.97 | 53.64 | 7.52 | 7.26 | 7.39 | 29.66 |
| PostAttr | R | 52.01 | 21.89 | 22.43 | 37.1 | 27.96 | 52.12 | 74.45 | 61.31 | 61 | 36.88 | 45.97 | 53.64 | 5.75 | 5.64 | 5.7 | 29.1 |
| FRONT | R | 97.37 | 14.25 | 18.65 | 57.75 | 28.19 | 3.43 | 89.47 | 6.61 | 99.12 | 32.01 | 48.39 | 27.5 | 68.61 | 63.24 | 65.82 | 40.5 |
| | | | | | | | | | | | | | | | | | |
| TRUST-ALIGN (DPO) | R | 26.05 | 5.52 | 30.56 | 25.32 | 27.69 | 85.74 | 79.49 | 82.5 | 51.76 | 62.48 | 56.62 | 69.56 | 65.13 | 58.44 | 61.6 | 52.95 |
| **LLaMA-2-7b** | | | | | | | | | | | | | | | | | |
| ICL | R | 0.51 | 2.75 | 0.00 | 0.00 | 0.00 | 99.46 | 68.54 | 81.15 | 0.44 | 27.27 | 0.87 | 41.01 | 9.09 | 10 | 9.52 | 16.84 |
| PostCite | R | 5.62 | 14.62 | 15.98 | 2.86 | 4.85 | 94.28 | 68.49 | 79.34 | 5.43 | 30.33 | 9.2 | 44.27 | 5.26 | 5.19 | 5.23 | 18.12 |
| PostAttr | R | 5.62 | 14.62 | 15.98 | 2.86 | 4.85 | 94.28 | 68.49 | 79.34 | 5.43 | 30.33 | 9.2 | 44.27 | 2.27 | 2.25 | 2.26 | 17.13 |
| FRONT | R | 100 | 4.17 | 6.13 | 19.5 | 9.33 | 0.00 | 31.44 | 0.00 | 100 | 31.44 | 47.84 | 23.92 | 78.99 | 70.93 | 74.75 | 36 |
| | | | | | | | | | | | | | | | | | |
| TRUST-ALIGN (DPO) | R | 20.01 | 4.34 | 32.18 | 20.48 | 25.03 | 90.45 | 77.52 | 83.49 | 42.82 | 67.28 | 52.33 | 67.91 | 63.98 | 61.02 | 62.46 | 51.8 |
| **LLaMA-3.2-1b** | | | | | | | | | | | | | | | | | |
| ICL | R | 90 | 13.97 | 14.54 | 41.62 | 21.55 | 10.56 | 72.35 | 18.43 | 91.2 | 31.86 | 47.23 | 32.83 | 7.16 | 12.25 | 9.04 | 21.14 |
| PostCite | R | 30.84 | 12.09 | 5.53 | 5.43 | 5.48 | 68.59 | 68 | 68.3 | 29.62 | 30.19 | 29.9 | 49.1 | 2.6 | 2.75 | 2.67 | 19.08 |
| PostAttr | R | 48.41 | 10.97 | 6.79 | 10.46 | 8.24 | 51.04 | 67.83 | 58.25 | 47.21 | 30.67 | 37.18 | 47.72 | 1.45 | 1.5 | 1.5 | 19.15 |
| FRONT | R | 95.62 | 10.27 | 13.84 | 42.08 | 20.83 | 5.38 | 84.21 | 10.11 | 97.8 | 32.16 | 48.4 | 29.26 | 38.04 | 36.87 | 37.45 | 29.18 |
| | | | | | | | | | | | | | | | | | |
| TRUST-ALIGN (DPO) | R | 15.44 | 3.14 | 30.85 | 15.15 | 20.32 | 93.28 | 75.63 | 83.53 | 34.46 | 70.15 | 46.21 | 64.87 | 62.74 | 61.48 | 62.1 | 49.1 |
| **LLaMA-3.2-3b** | | | | | | | | | | | | | | | | | |
| ICL | R | 58.74 | 17.46 | 25.72 | 48.04 | 33.5 | 44.86 | 74.53 | 56 | 66.57 | 35.64 | 46.42 | 51.21 | 35.09 | 42.34 | 38.37 | 41.03 |
| PostCite | R | 82.85 | 24.16 | 17.71 | 46.68 | 25.68 | 18.29 | 73.12 | 29.26 | 85.34 | 32.39 | 46.95 | 38.11 | 5.34 | 5.24 | 5.29 | 23.03 |
| PostAttr | R | 82.85 | 24.75 | 17.56 | 46.26 | 25.45 | 18.63 | 74.46 | 29.8 | 86.07 | 32.67 | 47.36 | 38.58 | 3.39 | 3.4 | 3.4 | 22.48 |
| FRONT | R | 83.36 | 12.03 | 18.76 | 49.73 | 27.24 | 21.72 | 89.47 | 34.96 | 94.43 | 35.62 | 51.73 | 43.34 | 51.21 | 50.62 | 50.91 | 40.5 |
| | | | | | | | | | | | | | | | | | |
| TRUST-ALIGN (DPO) | R | 7.24 | 1.34 | 31.32 | 7.21 | 11.72 | 98.05 | 72.47 | 83.34 | 18.77 | 81.53 | 30.51 | 56.93 | 80.25 | 76.54 | 78.35 | 49 |
| **LLaMA-3-8b** | | | | | | | | | | | | | | | | | |
| ICL | R | 0.65 | 0.28 | 70.24 | 1.44 | 2.82 | 99.87 | 68.91 | 81.55 | 1.76 | 85.71 | 3.45 | 42.5 | 68.93 | 70 | 69.46 | 38.26 |
| PostCite | R | 15.68 | 4.87 | 21.13 | 10.53 | 14.06 | 85.27 | 69.33 | 76.48 | 17.74 | 35.59 | 23.68 | 50.08 | 7.15 | 7.03 | 7.09 | 23.74 |
| PostAttr | R | 15.68 | 4.87 | 21.13 | 10.53 | 14.06 | 85.27 | 69.33 | 76.48 | 17.74 | 35.59 | 23.68 | 50.08 | 6.29 | 6.29 | 6.29 | 23.47 |
| FRONT | R | 99.26 | 16.67 | 19.97 | 63.05 | 30.34 | 0.94 | 87.5 | 1.86 | 99.71 | 31.58 | 47.97 | 24.92 | 57.2 | 56.21 | 56.7 | 37.32 |
| | | | | | | | | | | | | | | | | | |
| TRUST-ALIGN (DPO) | R | 17.29 | 4.38 | 39.89 | 20.82 | 27.36 | 93.48 | 76.67 | 84.24 | 37.98 | 72.75 | 49.9 | 67.07 | 70.55 | 69.67 | 70.11 | 54.85 |
| **Closed-source Models** | | | | | | | | | | | | | | | | | |
| GPT-3.5 | R | 59.47 | 24.53 | 28.01 | 52.98 | 36.65 | 48.02 | 81.23 | 60.36 | 75.81 | 40.08 | 52.43 | 56.39 | 63.19 | 64.68 | 63.93 | 52.32 |
| GPT-4 | R | 72.20 | 25.92 | 29.66 | 68.11 | 41.32 | 35.98 | 88.72 | 51.2 | 90.03 | 39.21 | 54.63 | 52.91 | 70 | 69.66 | 69.83 | 54.69 |
| Claude-3.5 | R | 73.95 | 6.19 | 8.32 | 19.57 | 11.68 | 34.03 | 89.56 | 49.32 | 91.35 | 38.84 | 54.51 | 51.91 | 9.76 | 11.84 | 10.7 | 24.76 |

Table 23: GPT-4 prompting templates used for generating natural response based on gold claims for ASQA and ELI5.

| Type | Template |
|------|----------|
| ASQA | Please provide a high-quality answer to the given question using the provided document.  The answer must include all the answer labels, and each answer label used should be marked with its index immediately after it in the format [Answer Label X], where X is the index of the answer label in the provided list starting from 1.  For example, [Answer Label 1].  Ensure the answer is coherent and natural and does not exceed four statements.  You cannot make up any factual information based on your imagination:  The additional information added from the given document should be relevant to the question and grounded by the document, but must not contain any factual information that cannot be inferred from the given answer labels.  (e.g., if the answer label does not mention a specific year, you cannot introduce a specific year in the final answer).
Question:  {question}
Document:  {passage}
{answers}
Output: |
| ELI5 | Given a problem and some claims as answer tags, please generate a high-quality response.  The response needs to follow the following requirements:
1.  Use only all of the claims:  Ensure that the response contains and only contains information from the given claims, without introducing any new information.  Guarantee covering all claims in the response.
2.  Each statement must contain valuable information:  Every statement must either directly originate from the claims or infer from the claims, avoiding any irrelevant and unuseful information included in the response.  You can use each claim only for one time.
3.  Condense and combine:  If there are similarities between claims, merge them into a comprehensive statement to make the response more concise.  For example, if two claims both mention similar aspect of health benefits, they can be merged into one statement.
4.  Fluent and natural:  Ensure that the statements in the response are coherent and natural, using connecting words and maintaining logical order between statements.
5.  Answer tags in response:  Indicate each claim immediately after the corresponding content in the response with the format [Claim X], where X is the index of the claim in the provided list starting from 1.  For example, [Claim 1].
Question:  {question}
{claims}
Generated Response: |

Table 24: Prompt used for acquiring domain labels and knowledge demanding score.

**Prompt**

The examples below are questions from the same cluster. Identify a single short topic they share in common, for example: Philosophy, Lifestyle, Linear Algebra, Biochemistry, Economics, etc. Additionally, evaluate if the topics in the examples are broadly suitable as knowledge-demanding questions that require additional research or grounding. Exclude any sensitive, inappropriate, or irrelevant content, such as sex, explicit violence, ads & scams, and other NSFW subjects. Consider a wide range of content, including scientific, educational, historical, cultural, and practical applications. Provide a rating from 1 to 7 based on the topic's dependence on additional knowledge or search materials: a score of 1 indicates the question can be answered with common sense alone, without needing any additional information lookup; a score of 5 means the topic requires a combination of common sense and additional lookup, roughly an equal split between the two; a score of 7 indicates that answering the question directly would be difficult, and without additional information, the answer would likely be incorrect. The output format should be like this: Topic: the_topic, Demanding value rating: score.

Table 25: Two types of instruction templates used as model input; answers are included only for tuning purposes.

| Type | Prompt |
|---|---|
| Default | Instruction: Write an accurate, engaging, and concise answer for the given question using only the provided search results (some of which might be irrelevant) and cite them properly. Use an unbiased and journalistic tone. Always cite for any factual claim. When citing several search results, use [1][2][3]. Cite at least one document and at most three documents in each statement. If multiple documents support the statement, only cite a minimum sufficient subset of the documents.
Document [1]: {passage1}
Document [2]: {passage2}
...
Question: {question}
Answer: {answer} |
| Refusal | {Default Instruction} + If none of the provided documents contains the answer, only respond with ``I apologize, but I couldn't find an answer to your question in the search results.'' Do not add further explanation as to why an answer cannot be provided; just state the response above as-is.
Document [1]: {passage1}
Document [2]: {passage2}
...
Question: {question}
Answer: {answer} |

Table 26: The prompts used for GPT-4 based critic.

**Coverage Critic Prompt**

```
[INSTRUCTION]
You will be given Question and the corresponding correct answers, along with a candidate answer and
reference facts.  Please follow these steps to process the candidate answer:
1.  Carefully read and understand the given Question, the list of correct answers, and the candidate
answer.
2.  For each given correct answer, first determine if there is a conflict with the candidate answer:
- If there is no conflict, and it is included in the candidate answer, extract the matched term from the
candidate answer and classify them as "upvote".
- If there is a conflict, identify the specific conflicting span within the candidate answer (accurately
pinpoint the details), classify it as "downvote", then only minimally modify the conflicting part of
the candidate answer to correct it according to the corresponding correct answer (using context from the
reference fact).  Classify the modified span as "revise".
- If there is a conflict, but it is not included in the candidate answer, extend the candidate answer
to include the correct answer (using material from the corresponding part of the reference facts), and
classify the extended portion as "revise".
3.  At the end of your response, provide the following:  - The final revised candidate answer that
includes all correct answers and has no conflicts (if no modification is needed, output the original
one).

[TASK]
Question:  {QUESTION}
Correct Answers:  {SHORT_ANS}
Candidate Answer:  {CANDIDATE}
Reference Facts:  {FACT}
```

**Citation Critic Prompt**

```
[INSTRUCTION]
Given a question and a list of CLAIMs, use the provided FACTs to determine which numbered FACTs togeter
SUPPORT, OPPOSE, or are IRRELEVANT to each CLAIM. Follow these to give your judgement:
1.  "SUPPORT" means the FACT directly participates in supporting the factuality of the CLAIM. The CLAIM
should be strongly implied by the FACT.
2.  "OPPOSE" means the FACT contributes to prove the CLAIM contains at least one factual error.
3.  "IRRELEVANT" means the FACT does not contribute directly to either SUPPORT or OPPOSE the given
CLAIM.
4.  Carefully read the given question and FACTs to ensure you have a clear understanding of them.
5.  For each CLAIM, analyze its content to show all factual arguments and assertions.
6.  Look into the details of each FACT, and find factual-related points of each FACT.
7.  Before determining your final judgement for all CLAIMs at the end, state your reasoning and evidence
first.
8.  In your final judgement, give a numbered list with each line corresponding to a CLAIM. Then, for
each CLAIM, separately list the index of each FACT for "SUPPORT", "OPPOSE", and "IRRELEVANT", with the
format [FACT X], where X is the index of the FACT starting from 1.  For example, suppose we have two
CLAIMs and three FACTs in total:  "/n/n1.  SUPPORT: [FACT 1][FACT 3], OPPOSE: NONE, IRRELEVANT: [FACT
2]/n2.  SUPPORT: NONE, OPPOSE: [FACT 2], IRRELEVANT: [FACT 1][FACT 3]".  If no FACT, then just give
"NONE".

[TASK]
Question:  {QUESTION}
CLAIM: {CLAIM_PLACEHOLDER}
FACTs:  {FACT_PLACEHOLDER}
```

Table 27: The prompts used for GPT-4o based evaluator for refusal detection.

**GPT Evaluator Prompt**

Instruction: You are an intelligent evaluator. Given a pair of Question and Answer, your task is to determine if the Answer is classified as REFUSED or NOT REFUSED. Please follow these steps to produce your judgement:
1. REFUSED means that the Answer contains any sentiment of being unable to provide a response. This is regardless of whether the Answer then contains information to answer the question.
2. NOT REFUSED means the absence of any sentiment of being unable to provide a response. The Answer attempts to address the Question.
3. Carefully read and understand the Question and Answer.
4. Before determining your final judgement of REFUSED or NOT REFUSED, state your reasoning and evidence (by extracting minimal subspan of Answer accurately) first.
5. For each Answer, determine if it is REFUSED or NOT REFUSED.
6. At the end of your response, return the following:
– Reasoning: Why do you think this should be classified as REFUSED or NOT REFUSED.
– Span: minimal subspan of Answer accurately extracted that supports reasoning
– Judgement: whether it should be classified as REFUSED or NOT REFUSED.

Question: Who set the record for longest field goal?
Answer: I apologize, but I couldn't find an answer to your question in the search results.
Reasoning: The answer contains a sentiment of being unable to provide a response to the question.
Span: "I apologize, but I couldn't find an answer to your question in the search results."
Judgement: REFUSED

Question: Who killed the man in thelma and louise?
Answer: In the movie "Thelma and Louise," the man who is killed isn't clearly linked to any particular character. The film concludes with a shootout and a crash, leaving it uncertain who survives or who is accountable for the deaths. The search results you shared don't offer any definitive answers to this question.
Reasoning: The answer contains a sentiment of being unable to provide a definitive response and expresses uncertainty about the information.
Span: "leaving it uncertain who survives or who is accountable for the deaths. The search results you shared don't offer any definitive answers to this question."
Judgement: REFUSED

Question: When did the us break away from England?
Answer: When the US left England is not explicitly stated. Hence I cannot provide an answer to your question.
Question: Who played galen in planet of the apes?
Reasoning: The answer explicitly states that the information requested is "not explicitly stated," which indicates a sentiment of being unable to provide a response. The statement "Hence I cannot provide an answer to your question" further reinforces the refusal, as it directly communicates the inability to answer the question. Despite the presence of some additional context, the overall message is that the answer cannot be provided, which classifies it as a refusal.
Span: "When the US left England is not explicitly stated. Hence I cannot provide an answer to your question."
Judgement: REFUSED

Question: Which is the most rainy place on earth?
Answer: Several places on Earth claim to be the most rainy, such as Lloró, Colombia, which reported an average annual rainfall of 12,717 mm between 1952 and 1989, and López de Micay, Colombia, which reported an annual 12,892 mm between 1960 and 2012 [3]. However, the official record is held by Mawsynram, India with an average annual rainfall of 11,872 mm [3], although nearby town Sohra, India, also known as Cherrapunji, holds the record for most rain in a calendar month for July 1861 and most rain in a year from August 1860 to July 1861 [1].
Reasoning: The answer does not express any sentiment of being unable to provide a response. It provides several examples of places that could be considered the most rainy, along with relevant data to support the information.
Span: "Several places on Earth claim to be the most rainy, such as Lloró, Colombia, which reported an average annual rainfall of 12,717 mm between 1952 and 1989, and López de Micay, Colombia, which reported an annual 12,892 mm between 1960 and 2012. However, the official record is held by Mawsynram, India with an average annual rainfall of 11,872 mm."
Judgement: NOT REFUSED

Question: {Q}
Answer: {A}

Return the following:
Reasoning:
Span:
Judgement: