# OpenReview forum: "Measuring and Enhancing Trustworthiness of LLMs in RAG through Grounded Attributions and Learning to Refuse"
_ICLR.cc/2025/Conference — ICLR 2025 Oral_

### Official Review · Reviewer_ewWr · 2024-11-03

**Soundness:** 3
**Presentation:** 3
**Contribution:** 3
**Rating:** 8
**Confidence:** 3

**Summary:**

This paper examines the trustworthiness of LLMs within the RAG framework. The authors introduce a unified evaluation metric, TRUST-SCORE, designed to assess both response truthfulness and attribution groundedness, ensuring that model outputs are based on retrieved documents rather than on the model's internal parametric knowledge. Additionally, the authors propose a new alignment method, TRUST-ALIGN, to enhance LLM trustworthiness according to the TRUST-SCORE metric, demonstrating improvements over baseline approaches such as in-context learning when also evaluated with the proposed metric.

**Strengths:**

The paper is well-written, with a relevant set of experiments to evaluate the meaningfulness of the proposed metric and alignment method across various datasets, and model configurations. The inclusion of ablation studies and additional results allows for a deeper understanding of the method's impact.

The rationale behind each term in the proposed metric is clear and grounded in prior research, making it easy to interpret the components individually and understand their contributions to the final score. This breakdown helps readers see how each part of the metric contributes to the final score, making it easier to understand which factors are driving the results. Such clarity is particularly valuable for RAG models where responses must be both accurate and grounded in retrieved documents rather than relying on the model's internal knowledge.

**Weaknesses:**

The results in Table 2 are hard to interpret. The best DPO values come from different model sizes depending on the dataset, which makes it hard to draw general conclusions about which configuration works best overall. Additionally, it’s unclear how reliable the TRUST-SCORE is as a standalone measure, since its value can end up high simply due to averaging across metrics., rather than showing a clear advantage in each metric. This approach can lead to "metric gaming," where high performance in one sub-metric can boost the final score even if other terms don’t improve much, which might overstate the model’s actual abilities. For example, PostCite TRUST score of 22.28 in row 2 of Table 2 under QAMPARI is higher than other baselines, despite having an EM score of zero.

Another problem is that while FRONT seems to be the strongest baseline, we don’t have results for it across different model sizes and families, particularly those where DPO performs best in Tables 2 and 3. This makes it challenging to judge how beneficial TRUST-ALIGN really is.

Lastly, the paper doesn’t clearly explain how the proposed alignment method is different from previous SFT and DPO methods mentioned in the related work section, leaving some ambiguity around what’s truly novel here.

**Questions:**

> L246: “The positive response corresponds to an answer that encompasses expected gold claims for q and corresponding citations referring to the documents.”

**Q1:** If the response contains partial gold claims or partial citations, is it included in the dataset? If so, as + or -?

> L277:  “we fine-tune LLaMA-2-7b on the source datasets, creating $M_{sft}$ (Appendix E.1).”

**Q2:** It's not clear how fine tune is performed and the appendix does not clarify that. Is the fine tuning task next token prediction? If so, how are questions, documents and answers stitched together?

**Q3:** Why does TRUST-ALIGN without refusal HT increase the $F1_{rg}$ score in the ELI5 dataset in Table 4? Is this a typo (see 2nd item in the list of typos below)?

**Q4:** I understand the importance of disentangling parametric knowledge from grounded knowledge for the purpose of this study, but in practice, one would leverage both. If I understand correctly, the alignment method proposed by the authors could be used to reduce hallucination. How do the metric and alignment method introduced in this work generalize to real-world scenarios?

## Typos:

L282: 19K. Figure 2 says 20K and 50% of 40K is 20K.

L449: the difference is 0.48% for QAMPARI and -0.78% for ELI5.

---

> ### Author Response · Authors · 2024-11-21
> **Rebuttal by Authors**
>
> Thank you for your constructive feedback. We have carefully reviewed and addressed the points you raised, and we are happy to provide further clarification if needed. If you find that your concerns have been resolved, we would be grateful if you could consider improving your score.
>
> **[Clarification 1]: Table 2 clarification**
>
> Thank you for your feedback. Trust-Align **does not aim to introduce a new state-of-the-art LLM for RAG**, as such, we **do not identify a single best configuration**. Instead, the method demonstrates how to enhance an LLM's capabilities in **three key dimensions of trustworthiness**: answer correctness ($EM^{F1}\_{AC}$ score), citation groundedness ($F\_{CG}$), and refusal groundedness ($F\_{RG}$) **relative to its own baseline**.
>
> Table 2 aims to highlight the effectiveness of Trust-Align in enhancing the performance of **each model family**, rather than comparing between models. To better convey this within-model improvement, we will adjust how the results are presented, specifically, we will highlight the best values within each family.
>
> ------
>
> **[Clarification 2]: Reliability of Trust-Score as a standalone measure**
>
> Thank you for bringing up this crucial point. We agree that the **standard arithmetic average (AM)** may not represent the skewed score of the sub-metrics. Thus, depending on what is important, defined by the **application** and **use case**, one may take a weighted average. As far as we understand, there’s **no one-size-fits-all solution** in this case, as aggregation would inevitably lead to information loss. Thus, the individual sub-metrics are independently crucial.

---

> ### Author Response · Authors · 2024-11-21
> **Rebuttal by Authors**
>
> **[Clarification 3]: Additional FRONT results demonstrating the effectiveness of Trust-Align**
>
> We thank the reviewer for raising this point. In our revised version, we have included the complete results of FRONT. Additionally, we have attached the results of FRONT across different model sizes and families (corresponding to tables 2 and 3) for your convenience below.
>
> Trust-Align performance relative to the additional FRONT results is **on par with the existing FRONT results**.  Moreover, Trust-Align delivers **substantial improvements in $F1\_{RG}$ across all 27 configurations** and **enhances $F1\_{CG}$ in 24 out of 27 configurations**, further demonstrating its effectiveness in improving response groundedness and citation quality.
>
> _Table 1: Additional FRONT results_
> |   **Model**   |  **Dataset** | **Method**| AR (%) | $EM^{F1}_{AC}$ | $F1\_{RG}$ | $F1\_{CG}$ | TRUST  |
> |------------------|----------|-------|--------|------------------------------|------------------|------------------|--------|
> | **LLaMA-2-7b**   | ASQA     | FRONT | 100.00 | 60.47                       | 39.15            | 68.86            | 56.16  |
> |                  |          | DPO   | 65.30  | 52.48                       | 66.12            | 83.94            | 67.51  |
> |                  | QAMPARI  | FRONT | 100.00 | 17.27                       | 22.78            | 24.26            | 21.44  |
> |                  |          | DPO   | 32.30  | 32.03                       | 71.67            | 49.42            | 51.04  |
> |                  | ELI5     | FRONT | 100.00 | 21.66                       | 17.15            | 52.72            | 30.51  |
> |                  |          | DPO   | 21.60  | 22.54                       | 63.27            | 47.35            | 44.39  |
> | **LLaMA-3.2-1b** | ASQA     | FRONT | 79.11  | 48.22                       | 54.48            | 48.29            | 50.33  |
> |                  |          | DPO   | 41.67  | 38.64                       | 58.61            | 79.35            | 58.87  |
> |                  | QAMPARI  | FRONT | 98.60  | 7.57                        | 24.54            | 15.32            | 15.81  |
> |                  |          | DPO   | 20.00  | 27.22                       | 67.92            | 49.42            | 48.19  |
> |                  | ELI5     | FRONT | 97.20  | 16.11                       | 20.76            | 30.19            | 22.35  |
> |                  |          | DPO   | 9.60   | 13.20                       | 59.35            | 48.21            | 40.25  |
> | **LLaMA-3.2-3b** | ASQA     | FRONT | 95.25  | 63.19                       | 49.45            | 57.46            | 56.70  |
> |                  |          | DPO   | 77.85  | 59.82                       | 66.38            | 84.21            | 70.14  |
> |                  | QAMPARI  | FRONT | 92.70  | 12.99                       | 32.89            | 19.19            | 21.69  |
> |                  |          | DPO   | 48.20  | 29.13                       | 70.85            | 45.65            | 48.54  |
> |                  | ELI5     | FRONT | 86.90  | 19.95                       | 32.21            | 41.97            | 31.38  |
> |                  |          | DPO   | 17.50  | 18.33                       | 62.79            | 55.87            | 45.66  |
> | **LLaMA-3-8b**   | ASQA     | FRONT | 99.05  | 62.25                       | 41.62            | 66.14            | 56.67  |
> |                  |          | DPO   | 56.43  | 53.94                       | 65.49            | 88.26            | 69.23  |
> |                  | QAMPARI  | FRONT | 100.00 | 13.53                       | 22.78            | 20.42            | 18.91  |
> |                  |          | DPO   | 22.40  | 35.35                       | 70.73            | 58.77            | 54.95  |
> |                  | ELI5     | FRONT | 99.50  | 18.99                       | 17.85            | 44.69            | 27.18  |
> |                  |          | DPO   | 15.50  | 20.81                       | 63.57            | 50.24            | 44.87  |
> | **Phi3.5-mini**  | ASQA     | FRONT | 99.79  | 63.30                       | 39.79            | 71.63            | 58.24  |
> |                  |          | DPO   | 66.56  | 52.23                       | 64.20            | 85.36            | 67.26  |
> |                  | QAMPARI  | FRONT | 100.00 | 11.97                       | 22.78            | 21.50            | 18.75  |
> |                  |          | DPO   | 30.10  | 36.42                       | 73.95            | 53.40            | 54.59  |
> |                  | ELI5     | FRONT | 96.60  | 21.46               | 21.35            | 61.41            | 34.74  |
> |                  |          | DPO   | 24.90  | 23.39                     | 67.62            | 47.42            | 46.14  |
>
> [Continued below]

---

> > ### Author Response · Authors · 2024-11-21
> > **Rebuttal by Authors**
> >
> > [Continued from above]
> >
> > **Model**   |  **Dataset** | **Method**| AR (%) | $EM^{F1}\_{AC}$ | $F1\_{RG}$ | $F1\_{CG}$ | TRUST  |
> > |------------------|----------|-------|--------|------------------------------|------------------|------------------|--------|
> > | **Qwen-2.5-0.5b**| ASQA     | FRONT | 100.00 | 42.83                       | 39.15            | 45.87            | 42.62  |
> > |                  |          | DPO   | 71.84  | 50.59                       | 61.28            | 52.40            | 54.76  |
> > |                  | QAMPARI  | FRONT | 99.30  | 11.52                       | 23.23            | 15.90            | 16.88  |
> > |                  |          | DPO   | 17.90  | 15.76                       | 61.84            | 29.73            | 35.78  |
> > |                  | ELI5     | FRONT | 99.90  | 13.74                       | 17.29            | 27.95            | 19.66  |
> > |                  |          | DPO   | 21.70  | 13.68                       | 60.79            | 22.72            | 32.40  |
> > | **Qwen-2.5-1.5b**| ASQA     | FRONT | 99.26  | 57.74                       | 41.36            | 55.70            | 51.60  |
> > |                  |          | DPO   | 72.57  | 52.68                       | 62.38            | 66.81            | 60.62  |
> > |                  | QAMPARI  | FRONT | 98.80  | 16.05                       | 24.45            | 11.60            | 17.37  |
> > |                  |          | DPO   | 20.00  | 23.80                       | 68.46            | 50.98            | 47.75  |
> > |                  | ELI5     | FRONT | 99.90  | 19.57                       | 17.29            | 37.70            | 24.85  |
> > |                  |          | DPO   | 33.60  | 19.03                       | 57.91            | 31.63            | 36.19  |
> > | **Qwen-2.5-3b**  | ASQA     | FRONT | 97.47  | 55.15                       | 44.01            | 62.72            | 53.96  |
> > |                  |          | DPO   | 49.47  | 55.19                       | 63.76            | 78.64            | 65.86  |
> > |                  | QAMPARI  | FRONT | 79.10  | 20.69                       | 48.62            | 25.67            | 31.66  |
> > |                  |          | DPO   | 48.10  | 35.69                       | 70.31            | 45.64            | 50.55  |
> > |                  | ELI5     | FRONT | 93.60  | 18.69                       | 25.37            | 37.40            | 27.15  |
> > |                  |          | DPO   | 13.50  | 22.52                       | 64.38            | 42.01            | 42.97  |
> > | **Qwen-2.5-7b**  | ASQA     | FRONT | 86.39  | 64.58                       | 60.08            | 58.27            | 60.98  |
> > |                  |          | DPO   | 59.49  | 55.04                       | 66.22            | 83.57            | 68.28  |
> > |                  | QAMPARI  | FRONT | 84.70  | 17.02                       | 42.85            | 24.48            | 28.12  |
> > |                  |          | DPO   | 32.10  | 30.11                       | 70.68            | 53.48            | 51.42  |
> > |                  | ELI5     | FRONT | 57.60  | 28.27                       | 54.14            | 56.61            | 46.34  |
> > |                  |          | DPO   | 21.00  | 24.30                       | 63.79            | 47.02            | 45.04  |

---

> > > ### Author Response · Authors · 2024-11-21
> > > **Rebuttal by Authors**
> > >
> > > **[Clarification 4]: Distinction from previous SFT and DPO works**
> > >
> > > We acknowledge that the distinctions in our approach may not have been clearly highlighted, and we appreciate the opportunity to clarify.
> > >
> > > In this work, we aim to address the **groundedness problem in RAG more holistically** by introducing the concept of **grounded answers, refusals, and attributions**—where the model must refuse to answer if the provided documents lack sufficient information pertaining to the query, answer with information grounded in the documents, and provide appropriate attributions for the context (set of documents). To improve an LLM's RAG fitness across these three aspects, we contributed in **two broad directions**: 1\) Proposing a **metric** that can **holistically measure groundedness**, and 2\) Constructing an **alignment dataset** to **enhance the groundedness score** of a given model.
> > >
> > > When it comes to the second aspect of our contribution which we believe the reviewer is concerned about, while the overarching elements of our framework may share similarities with prior works \[1-5\]—e.g., collecting seed data from open-source datasets, automatic data construction, fine-tuning with SFT/DPO—there are **fundamental differences** in our data construction approach that contributes towards novelty. One of the core distinctions of a data construction method lies in the specific problem it is designed to address. In our case, we aim to **enhance the model's performance in three key aspects**: refusal groundedness, answer groundedness, and citation groundedness. A non-ideal model is prone to exhibit **five types of hallucinations related to one of the groundedness aspects**: Unwarranted Refusal, Over-Responsiveness, Overcitation, Improper Citation, and Inaccurate Claims (as shown in Table 1). Focusing on these **broad categories of hallucinations** (including the important ones i.e. refusal-based) is one of the primary distinguishing factors compared to related works. This led us to design **dedicated data augmentation techniques** that encompass these non-idealities in the training and test sets; thus, the construction method differs substantially (Section 4).
> > >
> > > Notably, similar works \[1-5\] improve the overall framework with focus on data construction but **do not propose advancements in fine-tuning or preference optimization techniques** for RAG. Similarly, it is **not the focus of our work to propose a novel fine-tuning technique for RAG**. We believe that specialized training techniques tailored for this task is interesting but out of scope of this work.
> > >
> > > We appreciate the opportunity to discuss the contribution of our work and are happy to address any further questions or clarifications the reviewer may have.
> > >
> > > References:
> > >
> > > \[1\] Asai, A., Wu, Z., Wang, Y., Sil, A., & Hajishirzi, H. (2023). Self-RAG: Learning to Retrieve, Generate, and Critique through Self-Reflection. Retrieved November 21, 2024, from arXiv.org website: [https://arxiv.org/abs/2310.11511](https://arxiv.org/abs/2310.11511)
> > >
> > > \[2\] Ye, X., Sun, R., Arik, Sercan Ö, & Pfister, T. (2023). Effective Large Language Model Adaptation for Improved Grounding and Citation Generation. Retrieved November 21, 2024, from arXiv.org website: [https://arxiv.org/abs/2311.09533](https://arxiv.org/abs/2311.09533)
> > >
> > > \[3\] Huang, L., Feng, X., Ma, W., Gu, Y., Zhong, W., Feng, X., … Qin, B. (2024). Learning Fine-Grained Grounded Citations for Attributed Large Language Models. Retrieved November 21, 2024, from arXiv.org website: [https://arxiv.org/abs/2408.04568](https://arxiv.org/abs/2408.04568)
> > >
> > > \[4\] Li, D., Sun, Z., Hu, B., Liu, Z., Hu, X., Liu, X., & Zhang, M. (2024). Improving Attributed Text Generation of Large Language Models via Preference Learning. Retrieved November 21, 2024, from arXiv.org website: [https://arxiv.org/abs/2403.18381](https://arxiv.org/abs/2403.18381)
> > >
> > > \[5\] Zhang, T., Patil, S. G., Jain, N., Shen, S., Zaharia, M., Stoica, I., & Gonzalez, J. E. (2024). RAFT: Adapting Language Model to Domain Specific RAG. Retrieved November 21, 2024, from arXiv.org website: https://arxiv.org/abs/2403.10131v1

---

> > > > ### Author Response · Authors · 2024-11-21
> > > > **Rebuttal by Authors**
> > > >
> > > > **[Clarification 5]: If the response contains partial gold claims or partial citations, is it included in the dataset? If so, as + or -?**
> > > >
> > > > For responses containing partial gold claims or partial citations, we include it in the dataset as the **negative (unpreferred) response**. Specifically, it is categorized as having **“Inaccurate Claims”** or **“Improper Citation” hallucination type** respectively as defined in our paper.
> > > >
> > > > ------
> > > >
> > > > **[Clarification 6]: Is the fine tuning task next token prediction? If so, how are questions, documents and answers stitched together?**
> > > >
> > > > Thank you for bringing this up. This section could indeed benefit from further clarification. The fine-tuning task is indeed a **next token prediction task**. The model learns to predict the next token based on the given input, which includes instructions, documents, and questions formatted as a structured prompt. To fine-tune the model, we designed the input format to guide the model's behavior and align it with the objectives of the task. We attach the **complete prompt format** used for fine-tuning to illustrate how the questions, documents and answers stitched together:
> > > >
> > > > ```
> > > > Instruction: Write an accurate, engaging, and concise answer for the given question using only the provided search results (some of which might be irrelevant) and cite them properly. Use an unbiased and journalistic tone. Always cite for any factual claim. When citing several search results, use \[1\]\[2\]\[3\]. Cite at least one document and at most three documents in each statement. If multiple documents support the statement, only cite a minimum sufficient subset of the documents. If none of the provided documents contains the answer, only respond with ‘‘I apologize, but I couldn’t find an answer to your question in the search results.’’
> > > >
> > > > Question: Who was looking for a heart in the wizard of oz?
> > > >
> > > > Document \[1\]: {passage1}
> > > > Document \[2\]: {passage2}
> > > > Document \[3\]: {passage3}
> > > > Document \[4\]: {passage4}
> > > > Document \[5\]: {passage5}
> > > >
> > > > Answer:
> > > > ```
> > > > Model output: `The Tin Woodman was looking for a heart in "The Wizard of Oz" [1][2][4].`
> > > >
> > > > We hope this clarifies the reviewer’s doubts and we are happy to provide more clarification if needed.

---

> ### Author Response · Authors · 2024-11-21
> **Rebuttal by Authors**
>
> **[Clarification 7]: Why does TRUST-ALIGN without refusal HT increase the score in the ELI5 (Table 4)?**
>
> Thank you for helping to bring this typo and others mentioned in "Typos" to our attention. The value in the text for ELI5’s $F1\_{RG}$ was wrongly calculated with Trust-Score’s value; it will be corrected in the next version. As shown in the table, there is indeed an increase in $F1\_{RG}$ for ELI5 by 0.78% for the case without refusal HT. We provide a discussion below as to why an increase is observed.
>
> We observe that the increase in overall $F1\_{RG}$ is due to the **recall for answerable questions** ($R\_{ans}$, proportion of answered and answerable questions over total answerable questions) **increasing more than the corresponding drop in precision** ($P\_{ans}$, proportion of answered answerable questions over total answered questions), resulting in an **increase in $F1\_{ans}$**. This increase **outweighs the drop in its $F1\_{ref}$,** leading to the **overall rise in $F1\_{RG}$**.
>
> _Table 2: Component wise breakdown of $F1\_{RG}$_
> | Model | $R\_{ref}$ | $P\_{ref}$ | $F1\_{ref}$ | $R\_{ans}$ | $P\_{ans}$ | $F1\_{ans}$ | $F1\_{RG}$ |
> |----------------------|-----------|-----------|------------|-------------|-------------|------------|-------------|
> | DPO-LLaMA-2-7b | 83.98 | 84.95 | 84.46 | 43.00 | 41.20 | 42.08 | 63.27 |
> | Without refusal HT | 78.43 | 86.75 | 82.38 | 54.10 | 39.57 | 45.71 | 64.05 |
>
> ------
>
> **[Clarification 8]: How does the metric and alignment method introduced in this work generalize to real-world scenarios?**
>
> It is an interesting question, thank you for asking. In **critical domains** like healthcare and law, one should **only rely on vetted knowledge retrieved from a vetted knowledge** base to ground the answer. The answers are not admissible otherwise. We may allow the LLMs to employ the commonsense reasoning skills learned during pre-training, but those **must not override what has been stated in the vetted documents**. This is especially important for facts, but also the rules of reasoning should they change from context to context. For instance, the legal age of driving varies from country to country, but it is typically 18 which could be encoded in the LLM parameters. If the reasoning involves such information, the LLM must use the vetted legal age of, say 21, over its parametric knowledge that states that it is 18\. This is where **grounded citation** ($F1\_{CG}$) comes into the picture. Again, should an LLM be unable to determine the legal age in some country to answer some question (where knowing this information is necessary, but not necessarily sufficient), it should simply refuse to do so, as it cannot ground the final answer in the retrieved documents. This is an aspect that **grounded refusals** ($F1\_{RG}$) seeks to measure. Using the age of 18, as encoded in the parameters, cannot be truthfully grounded and thus should not be trusted. Such scenarios are meant to be captured by our **proposed metrics** and **mitigated by alignment**.
>
> If the reviewer feels that this point should be emphasized in the draft, we would love to do it.

---

> > ### Comment · Reviewer_ewWr · 2024-11-22
> > **Thank you**
> >
> > Thank you for addressing my questions and conducting extra experiments. I am satisfied with the answers. I updated my ratings to reflect my new opinion that this paper is a clear accept.

---

> > > ### Author Response · Authors · 2024-11-26
> > > **Thank you!**
> > >
> > > Dear Reviewer ewWr,
> > >
> > > Thank you so much!
> > >
> > > --
> > >
> > > Authors

---

### Official Review · Reviewer_4fcY · 2024-11-03

**Soundness:** 3
**Presentation:** 3
**Contribution:** 3
**Rating:** 8
**Confidence:** 3

**Summary:**

The paper introduces a metric called "trust-score" for assessing groundedness, alongside an alignment approach aimed at improving this metric. It provides a useful metric for advancing the groundedness of large language models (LLMs) for a variety of applications. I am actually surprised there has been no metrics like this for groundedness.

**Strengths:**

It provides a useful metric for advancing the groundedness of large language models (LLMs) for a variety of applications.

**Weaknesses:**

Minor Issue: The definition of the Trust Score, particularly the subscript notation, is somewhat unclear (and also in Figure 1). It might help to explore ways to improve readability.

Distinction Between Parametric Knowledge and Groundedness: Could you further discuss the practical implications of distinguishing between parametric knowledge and groundedness? While it adds rigor to make this distinction, to what extent would a user be concerned about whether information comes from parametric knowledge versus grounded sources? In which scenarios might this distinction be more critical?

**Questions:**

Choice of Models in Figure 2 (Steps 4–6): Why are different models used in Steps 4 (GPT-4) and Steps 5–6 (LLaMA-2-7B)? Is it because Step 5 requires fine-tuning? However, fine-tuning is also possible with GPT-4 via API, correct? If the decision was driven by the use of DPO for alignment in Step 6, could you clarify the rationale for these model choices?

Method Clarification (Line 271): Could you elaborate on the methodology for selecting documents similar to those containing gold claims but still irrelevant to the query?

Comparative Baseline (Line 359): What results would you observe if compared to a simple instruction-based baseline? For instance, Baseline 1 could involve instructing the model explicitly to avoid over-responding.

Clarification on SFT Model (Line 387): Regarding the SFT model, do you mean fine-tuning with the trust-align dataset, specifically with only positive responses (r+)? Please clarify if so.

---

> ### Author Response · Authors · 2024-11-21
> **Rebuttal by Authors**
>
> Thank you for your thoughtful comments and we appreciate the feedback you have given us. We have addressed your points below and are happy to assist in any further clarifications.
>
> **[Clarification 1]: Clarifying Trust-Score notation**
>
> Thank you for the suggestion. We will update the figure and corresponding part of our article with a **clearer notation system** for improved readability. Since this will require changes at various places in the draft, we will incorporate them in the **next version after the discussion phase ends** to minimize notation conflicts during the rebuttal.
>
> ------
>
> **[Clarification 2]: Distinction Between Parametric Knowledge and Groundedness**
>
> This is an interesting question, thank you for asking! There are numerous real-world scenarios where it is critical to distinguish between parametric knowledge and external knowledge (groundedness). Parametric knowledge refers to knowledge derived from the model’s training data and stored in the model’s weights. In contrast, external knowledge pertains to information sourced from external documents or real-time data. This distinction becomes particularly important when LLMs are used as **information consolidators** (primarily using augmented knowledge) rather than as **information generators** (relying solely on parametric knowledge). The need for groundedness is especially pressing in **high-stakes applications** such as legal tasks (maintaining accurate records), healthcare (summarizing patient-doctor conversations), and finance (extracting compliance details or assessing fund performance), where **outputs must faithfully align with the provided documents**.
>
> Another case where such distinction is important is where the **answer depends on changing real-world facts**. Parametric knowledge is static after training and cannot easily reflect updates in real-world information. This limitation is significant for **AI-based search engines** like Perplexity and You.com, where the primary role of the LLM is to act as an information consolidator, synthesizing **grounded, up-to-date web search results**, rather than generating responses based on potentially outdated parametric knowledge. Groundedness enables LLMs to adaptively provide **contextually relevant and current information**, making them more reliable for **applications requiring real-time awareness**.
>
> We appreciate the opportunity to discuss the importance of distinguishing parametric knowledge from external knowledge (groundedness) and are happy to address any further questions or clarifications the reviewer may have.
>
> ------
>
> **[Clarification 3]: Choice of Models in Figure 2 (Steps 4–6)**
>
> Trust-Align constructs a preference dataset consisting of **positive (preferred) responses in Step 4** and **negative (unpreferred) responses in Steps 5-6**. The choice of GPT-4 for Step 4 and LLaMA-2-7B for the later steps is primarily driven by the **quality and quantity of responses** we aimed to generate.
>
> For positive samples, our goal was to produce responses in the format: 'statement1 \[1\]\[2\] statement2 \[3\].' Given the **state-of-the-art instruction-following capabilities of GPT-4**, it was the natural choice over open-source models (e.g., LLaMA-2-7B) for generating responses that **effectively stitch together gold claims** with their corresponding attributions. Notably, a range of frontier models can perform this task; however, GPT-4's **greater accessibility** to us was one reason we chose it over other APIs, such as Claude-3.5.
>
> The procedure for generating negative responses is different. We focused on two aspects when choosing a model: 1\) the **naturalness or quality of the response**, and 2\) the **quantity or a sufficient number of responses** to perform preference optimization. We found that LLaMA-2-7B tends to generate **more diverse hallucinations**, which allowed us to **retrieve more negative samples** and consequently have a greater number of (positive, negative) tuples for preference alignment compared to GPT-4.
>
> While simple ICL was initially explored for generating negative responses, it was found to produce outputs of poor quality. To address this limitation, fine-tuning was carried out to align LLaMA-2-7B to generate responses in the format: 'statement1 \[1\]\[2\] statement2 \[3\],' without requiring explicit ICL. This fine-tuning step significantly **improved the quality of negative responses**, improving coherence and clarity. Thank you for asking this question and we hope that our answer clarifies the points mentioned.

---

> > ### Comment · Reviewer_4fcY · 2024-11-26
> > **Thanks for the reply**
> >
> > Clarification 2 is excellent and is prob worth adding some of the response to the paper to strengthen the contribution.

---

> ### Author Response · Authors · 2024-11-21
> **Rebuttal by Authors**
>
> **[Clarification 4]: Method Clarification (Line 271)**
>
> Thanks for pointing this out. This section could indeed benefit from further clarification. The irrelevant documents used in our method are selected for their **similarity to the question** (cosine similarity \> 0.7) but are carefully filtered to ensure they **do not contain any gold claims**, as determined by the **TRUE NLI model**. This approach ensures that we have **high-quality negative examples**.
>
> From an initial pool of 100 documents pre-retrieved using **GTR/BM25** (as detailed in the "Collecting D’s" section of the methods), we filter a subset of 50 documents. These documents are chosen based on their **cosine similarity** to the question while ensuring, via the **TRUE NLI model**, that they do not support any claims related to the question. From this filtered subset, we sample **groups of five documents** to create document sets, with questions associated with these sets labeled as **unanswerable**.
>
> ------
>
> **[Clarification 5]: Comparative Baseline (Line 359)**
>
> Thank you for bringing this up, we have provided an elaborate discussion on this in **Appendix G.3**.
>
> Instruction-based refusal (ICL), which combines in-context examples with a refusal instruction, is a **key baseline** in our study. Results for this baseline are presented in Tables 2 and 3 of the main paper under rows labeled "ICL."
>
> While Tables 2 and 3 include refusal prompt results only (ICL), we conducted a small study  on frontier models (GPT-4, GPT-3.5, Claude-3.5) and Trust-Aligned LLaMA to determine how the refusal prompt compares to the default prompt. As shown in Table 12, the **refusal prompt (R) outperforms the default prompt (D)**.
>
> Templates:
> - Default prompt: `Write an accurate, engaging, and concise answer for the given question using only the provided search results…`
> - Refusal prompt: [Appends to the Default Prompt] `If none of the provided documents contains the answer, only respond with 'I apologize, but I couldn't find an answer...’`
>
> _Table 1: Refusal and default prompting results on LLaMA-2-7B (more models can be found in the paper appendix)_
>
> | Baseline | Prompt | AR% | $EM^{F1}\_{AC}$ | $F1\_{RG}$ | $F1\_{CG}$ | $\\bf{TRUST}$ |
> | --- | --- | --- |  --- | --- | --- | --- |
> | ICL | Refusal | 0.00 | 0.00 |  26.28 | 0.00 | 8.76 |
> | ICL | Default | 94.30 | 50.38 | 49.51 | 43.67 | 47.85 |
>
> Taking LLaMA-2-7b as an example, **models rarely refuse under the default prompt** (AR% close to 100), while adding a **refusal prompt in ICL drastically reduces AR%**, often to near zero, indicating **indiscriminate refusal**. At both extremes, Trust-Score scores suffer due to errors in correctly refusing questions and lower citation groundedness scores. In contrast, Trust-Align enables models to identify and correctly answer appropriate questions, resulting in **nuanced refusal ability** and improvements in $F1\_{RG}$. This highlights that **prompting alone is insufficient to improve the model's trustworthiness effectively**. Interestingly, refusal prompting appears to yield **greater benefits in more capable models**, such as LLaMA-2-13b and LLaMA-3-8b.
>
> Although these findings are detailed in the appendix, we recognize their relevance to the reviewer’s question and will incorporate this discussion into the main paper for added clarity.
>
> ------
>
> **[Clarification 6]: Clarification on SFT Model (Line 387)**
>
> Yes, we first perform Supervised Fine-Tuning (SFT) on the based model with the **positive responses (r+)** from the Trust-Align dataset. Performing SFT prior to Direct Preference Optimization (DPO) is part of the **standard preference optimization pipeline** as outlined in Rafailov, R. (2023) and Ziegler, D. M. (2019). Initialization of the reference model with a supervised fine-tuned model **reduces the distribution shift** between the reference policy and the true reference distribution leading to **training stability** as noted by several works for instance Tunstall, L. (2019). The absence of the distillation SFT step often results in models **failing to learn effectively from feedback**, leading to poor performance. Thus, performing SFT prior to DPO is critical to training stability and ultimate model performance.
>
> ------
>
> References:
>
> \[1\] Rafailov, R., Sharma, A., Mitchell, E., Ermon, S., Manning, C.D., & Finn, C. (2023). Direct Preference Optimization: Your Language Model is Secretly a Reward Model. ArXiv, abs/2305.18290
>
> \[2\] Ziegler, D. M., Stiennon, N., Wu, J., Brown, T. B., Radford, A., Amodei, D., … Irving, G. (2019). Fine-Tuning Language Models from Human Preferences. Retrieved November 19, 2024, from arXiv.org website: [https://arxiv.org/abs/1909.08593](https://arxiv.org/abs/1909.08593)
>
> \[3\] Tunstall, L., Beeching, E., Lambert, N., Rajani, N., Rasul, K., Belkada, Y., Huang, S., Werra, L.V., Fourrier, C., Habib, N., Sarrazin, N., Sanseviero, O., Rush, A.M., & Wolf, T. (2023). Zephyr: Direct Distillation of LM Alignment. ArXiv, abs/2310.16944

---

> > ### Comment · Reviewer_4fcY · 2024-11-26
> > **Thanks for the reply**
> >
> > Clarification 4: thanks! worth adding a little bit details in the manuscript.
> >
> > Clarification 5: fascinating "adding a refusal prompt in ICL drastically reduces AR%, often to near zero". thanks for making it more prominent.

---

> > > ### Author Response · Authors · 2024-11-26
> > >
> > > Dear Reviewer 4fcY,
> > >
> > > Thanks so much for the constructive feedback!
> > >
> > > --
> > >
> > > Authors

---

### Official Review · Reviewer_xtne · 2024-11-04

**Soundness:** 3
**Presentation:** 3
**Contribution:** 4
**Rating:** 8
**Confidence:** 2

**Summary:**

The authors develop several new metrics for assessing LLM response groundedness in a RAG setting. These metrics comprise a holistic  groundedness/trustworthiness metric called Trust-Score. In addition, the authors develop a fine-tuning method called Trust-Aligne for increasing LLM response groundedness. The authors fine-tune several Llama, Qwen, and Phi models using this and other baseline methods, and they also compare the groundedness of the resulting models' responses to larger proprietary models prompted with ICL.

**Strengths:**

Originality: high. I don't think I've seen a thorough assessment of grounded refusals before.
Quality: moderate.
Clarity: moderate. The definition of Trust-Score is clear, nuanced, and well-motivated. The description of the dataset generation and fine-tuning process is also detailed and clear.
Significance: moderate.

**Weaknesses:**

Given that the best fine-tuned models only slightly out-performed GPT-4, it seems there are limited practical takeaways to be found for e.g. an application developer choosing and LLM. i.e. the practitioner is still well-justified in simply choosing a non-fine-tuned frontier model for their RAG application.

I found tables 2 and 3 very difficult to interpret, though mostly because it's not clear to me why the specific comparisons presented are valid or meaningful.

No confidence intervals are given.

**Questions:**

Why are the comparisons presented in Tables 2 and 3 meaningful? e.g. Trust-Score on LLaMA-3.2-3b fine-tuned with DPO compared to LLaMA-2-7b using FRONT? Why are comparisons not restricted to within-model?

Why is groundedness important in its own right if parametric knowledge can get the job done? If a model possesses the parametric knowledge to answer an otherwise un-answerable question, is "over-responsiveness" less an issue of hallucination than, say, poor instruction-following?

---

> ### Author Response · Authors · 2024-11-21
> **Rebuttal by Authors**
>
> Thank you for your constructive feedback. We have carefully addressed each of your points below. If you have any additional questions or need further clarification, we are more than willing to assist.
>
> **[Clarification 1]: ...best fine-tuned models only slightly out-performed GPT-4 ... limited practical takeaways to be found**
>
> This is an excellent question. Trust-Align does not aim to introduce a new state-of-the-art LLM for RAG. Instead, it demonstrates how to enhance a given LLM's capabilities in three key dimensions of trustworthiness: answer correctness ( $EM^{F1}\_{AC}$ score), citation groundedness ( $F\_{CG}$ ), and refusal groundedness ($F\_{RG}$). The effectiveness of Trust-Align depends on the **inherent capabilities of a model** i.e. to what extent they can be good for RAG. For example, on ExpertQA, Trust-Align significantly increased LLaMA-3-8b Trust-Score from 38.26% to 54.85% making it better than GPT-4 on refusal (67.07% vs 52.91%) and citation groundedness (70.11% vs 69.83%). While it is **not our goal to achieve SOTA**, given the trend observed: LLaMA 3b (49.0%) < LLaMA 7b (51.8%) < LLaMA 8b (54.85%), one can potentially achieve a much **higher score with larger open-source models (>10b)**.
>
> Observing that a (potentially) much smaller model of 8B size could outperform GPT-4 and Claude-3.5 at Trust-Score, we believe there is **significant room for improvement in frontier models**. These models can greatly **benefit from Trust-Align to achieve a higher Trust-Score**. We are eager to explore Trust-Align's potential on these models as soon as DPO support becomes available from their providers.
>
> Thus, while choosing a non-fine-tuned frontier model can deliver performance on par with a fine-tuned sub-10B parameter model, as demonstrated across different model families such as Qwen, LLaMA, and Phi, it is advisable to fine-tune a model to achieve better alignment for RAG applications.
>
> We appreciate the opportunity to discuss the contribution of our work and are happy to address any further questions or clarifications the reviewer may have.
>
> ------
>
> **[Clarification 2]: Clarifying Tables 2 and 3 and the corresponding comparisons presented**
>
> Thank you for your feedback. While we highlighted various findings from our experiments that may include inter-model comparisons, our main aim was to propose a method (Trust-Align) that **improves a given model's RAG appropriateness as measured by Trust-Score**.
>
> Table 2 and 3 aims to demonstrate the effectiveness of Trust-Align in improving the performance of models across different families, such as LLaMA, Qwen, and Phi. The emphasis is on showcasing the **improvements within each model family**, rather than identifying a single model that performs best across families. To better communicate these within-family improvements, we have revised how the results are presented by specifically highlighting the best values within each model family and tuned the corresponding discussion to better convey the contributions.
>
> ------
>
> **[Clarification 3]: Confidence intervals**
>
> Thank you for bringing this up. We intentionally omitted the significance testing results from the main draft to improve table readability. However, if the reviewers feel these results should be included, we would be happy to incorporate them in the next version. For your reference, we provide below the results of an independent sample t-test comparing baseline models with Trust-Aligned models.
>
> Table 2: Results from significance testing on TRUST SCORE
> | Dataset   | t-statistic       | p-value                |
> |-----------|-------------------|------------------------|
> | ASQA      | 7.70 | 9.85e-10  |
> | ELI5      | 6.40 | 1.18e-06 |
> | QAMPARI   | 7.83 | 3.31e-08  |
> | EXPERTQA | 8.63 | 8.00e-10   |
>
> Across all datasets, we observed a **t-statistic > 0 and p < 0.001** (Table 2), indicating **significant improvements for TRUST SCORE** at a significance level of 0.01.

---

> > ### Comment · Reviewer_xtne · 2024-11-23
> >
> > Thanks for improving the tables, they are much easier to interpret now.
> >
> > As you point out, the fact that this fine-tuning can improve the grounding behavior of an 8B model to exceed that of frontier models suggests that there might be significant room for improvement in frontier model behavior as well. Is that the main message of your paper? I for one did not come away with that as the call to action.

---

> ### Author Response · Authors · 2024-11-21
> **Rebuttal by Authors**
>
> **[Clarification 4]: Importance of groundedness when parametric knowledge potentially suffices.**
>
> > If a model possesses the parametric knowledge to answer an otherwise unanswerable question, is "over-responsiveness" less an issue of hallucination than, say, poor instruction-following?
>
> This is an interesting question. There are numerous scenarios where parametric knowledge may not be preferred for generating factual responses. Prominent examples include **AI-based search engines** like Perplexity and You.com, which prioritize **grounding their outputs in web data**. These platforms utilize various LLMs as **information consolidators** (leveraging augmented knowledge) rather than as **information generators** (relying on parametric knowledge). Other RAG applications where outputs must reference provided documents include legal tasks (maintaining accurate records), healthcare (summarizing patient-doctor conversations), and finance (extracting compliance details and assessing fund performance).
>
> While parametric knowledge can indeed enhance the performance of RAG systems, its utility is most pronounced in applications where the **ground-truth answers remain consistent**, regardless of whether augmented information is used. In such cases, the model’s output is not conditioned on attached documents; instead, the documents serve to improve the likelihood of producing a correct answer. However, in our work, which focuses on **attributed text generation**, we assume that the provided documents contain the necessary answers. If the documents are insufficient, the model is expected to generate a refusal, even if it has the correct answer encoded internally in its weights. One could, however, design a system that attributes parametric knowledge when a claim is generated without referencing the documents. This would require redefining the concept of citation groundedness, as citations would no longer correspond to specific documents.
>
> We acknowledge the reviewer’s point that poor instruction-following is a broader issue that underpins various problems, including hallucination and over-responsiveness. Hallucination can be viewed as a **specific and perhaps extreme manifestation of poor instruction-following**. Another manifestation is the **inappropriate use of valid parametric knowledge**, or "over-responsiveness," where models generate correct answers without grounding them in the provided documents. While this is **not hallucination** per se, it could be mechanistically correlated and warrants further research.
>
> In this work, we specifically address the **subset problem of RAG performance**, focusing on reducing **RAG-specific hallucinations** by improving citation groundedness and ensuring appropriate refusals. Poor instruction-following is a larger challenge, and while being a strong instruction-following model is likely a sufficient condition to reduce RAG hallucinations, the reverse is not necessarily true. If the reviewer feels this discussion should be included in the main draft, we would be happy to incorporate it.

---

> > ### Comment · Reviewer_xtne · 2024-11-23
> >
> > I'm (very weakly) suggesting that perhaps grounding behavior might be most effectively improved by improving instruction-following in general rather than focusing on fine-tuning for groundedness specifically. At least for models with high parametric knowledge, and especially in context of overall model performance.

---

> > > ### Author Response · Authors · 2024-11-26
> > > **Reply to Comment by Reviewer xtne**
> > >
> > > > Thanks for improving the tables, they are much easier to interpret now.
> > >
> > > We are grateful to the reviewer for constructive comments that helped us make the draft clearer and tables easy to understand.
> > >
> > > > As you point out, the fact that this fine-tuning can improve the grounding behavior of an 8B model to exceed that of frontier models suggests that there might be significant room for improvement in frontier model behavior as well. Is that the main message of your paper? I for one did not come away with that as the call to action.
> > >
> > > The main message of the paper is two folds:
> > >
> > > (Contribution-1) Proposing a **new measure**, Trust-Score, which evaluates an **LLM’s fitness for RAG applications more holistically**. It involves measuring of claim groundedness and attribution groundedness corresponding to the claims. A distinctive feature of the Trust-Score is that it **measures the model’s capacity to produce appropriate refusal responses** i.e. how effectively the model declines to provide an answer when the source documents lack sufficient information. Our analysis, summarized in Tables 2–4, reveals that both open-weight and frontier models exhibit heavily underperform on Trust-Score metric.
> > >
> > > (Contribution-2) To make model responses (claims and attributions) grounded in the documents, we propose Trust-Align. Trust-Align contribution primarily comes at the dataset level where we construct a **preference dataset** covering a range of samples specifically designed to reduce errors affecting the Trust-Score—Inaccurate Answer, Over-Responsiveness, Excessive Refusal, Over-Citation, and Improper Citation. We regard them as **LLM hallucinations within a RAG framework**.
> > >
> > > Given that frontier model providers do not facilitate DPO fine-tuning, claiming that these models will *undoubtedly* benefit from Trust-Align will be inappropriate. However, our findings—showing substantial improvements across the studied models and datasets, with average t-statistics of 7.64 points across benchmarks—indicate that **similar gains could be expected for frontier models**.
> > >
> > > While we are keen to apply Trust-Align (DPO) to frontier models as soon as it is available, we **explored its potential impact by performing SFT of GPT-4o** using positive samples from the Trust-Align dataset. The results are shown below:
> > >
> > > _Table: Trust-Align of GPT-4o using positive samples._
> > > | **Model**    | **Dataset** | **Method**      | AR (%)  | $EM^{F1}_{AC}$ | $F1_{RG}$ | $F1_{CG}$ | TRUST  |
> > > |--------------|-------------|-----------------|---------|----------------|-----------|-----------|--------|
> > > | **GPT-4o**   | ASQA        | ICL             | 84.49   | 62.92          | 61.40     | 73.66     | 65.88  |
> > > |              |             | SFT             | 74.26   | 59.22          | 68.62     | 87.54     | 72.09  |
> > > |              | QAMPARI     | ICL             | 60.40   | 14.29          | 75.20     | 20.43     | 33.69  |
> > > |              |             | SFT             | 34.60   | 41.56          | 77.15     | 53.64     | 56.99  |
> > > |              | ELI5        | ICL             | 66.10   | 35.25          | 68.33     | 37.71     | 41.58  |
> > > |              |             | SFT             | 25.50   | 24.10          | 68.34     | 56.09     | 48.99  |
> > >
> > >
> > > GPT-4o sees an improvement in Trust-Score by 6.21 (ASQA), 23.3 (QAMPARI), 7.41 (ELI5) points when aligned using a subset of Trust-Align data. We will include results of DPO on frontier models as soon as it is facilitated by the model providers.
> > >
> > > We are happy to clarify any further doubts the reviewer may have regarding our contributions and are grateful to the reviewer for such constructive discussions.
> > >
> > >
> > > > I'm (very weakly) suggesting that perhaps grounding behavior might be most effectively improved by improving instruction-following in general rather than focusing on fine-tuning for groundedness specifically. At least for models with high parametric knowledge, and especially in context of overall model performance.
> > >
> > > Thank you for the interesting suggestion. We agree that instruction following and grounding are interconnected tasks, as highlighted in the previous response. This is an intriguing direction that we will explore in future work.

---

### Official Review · Reviewer_U7F1 · 2024-11-04

**Soundness:** 4
**Presentation:** 4
**Contribution:** 4
**Rating:** 8
**Confidence:** 4

**Summary:**

Authors present a study of ‘grounded’ RAG in LLMs, i.e., a method for evaluating and aligning LLMs for RAG to increase the correctness of RAG, that LLMs cite the relevant literature and correct information, and identify when they do not have the necessary information to respond accurately (i.e., the “groundedness problem”). Using the direct preference optimization, the fine-tuned TRUST-ALIGN model approaches SOTA performance.

**Strengths:**

The paper is an original and significant contribution to the field of RAG in LLMs as it helps to mitigate the problems of citation hallucination or groundedness of claims.
The paper clearly articulates an extended experimental protocol, providing the rationale for its methodology. The findings are supported by the presentation of the results, and extensive appendices documenting the various steps of the study.

**Weaknesses:**

I cannot find any statement with regard to data validation by human reviewers. Despite the multiple steps involved in generating the training sets, it's unclear how the training data were validated. It would appear that the soundness of the generated dataset is reliant on the natural language inference engine and GTP-4 validation alone.
It is claimed that many of the results are significant but I cannot find any statistical tests to support those claims.

**Questions:**

1017 Collecting Quality Questions. The dataset construction begins by collecting a set of high- quality (challenging) and diverse questions from source datasets i.e. ASQA, QAMPARI, and ELI5—referred to as seed samples
-> These are used in evaluation, doesn’t this violate the separation principle? I understand these were used as ‘seed samples’ but if these have semantically similar enough, what guarantees are there that the models are not just fitting to the testing data?

---

> ### Author Response · Authors · 2024-11-21
> **Rebuttal by Authors**
>
> Thank you for your detailed and constructive feedback, we answer the questions/provide clarifications below:
>
> **[Clarification 1]: Human evaluation of Trust-Align dataset**
>
> That's a great point, here we provide observations from the human evaluation. From the Trust-Align dataset, we created a data mix containing 97 samples, with approximately 20 samples representing each error type. This was followed by human evaluation where five expert annotators rated each sample. Each response was rated on **three dimensions**, taking inspiration from the criteria used by Gao et al. (2023) and Liu et al. (2023): (1) **Correctness**, (2) **Citation Recall**, and (3) **Citation Precision**. For Correctness, given a response, a set of documents, and the question, a human evaluator assesses whether the answer is correct. The answer is labeled “correct” if it fully satisfies the information requested in the question and if the claims can be inferred from the documents; otherwise, it is marked “wrong.” For Citation Recall, given a sentence and all its cited documents, human evaluators determine whether the complete set of citations “fully supports” or “does not support” the sentence. For Citation Precision, given a sentence and one of its citations, human evaluators decide whether the citation “fully supports,” “partially supports,” or “does not support” the sentence.
>
> _Table 1: Data quality review of Trust-Align data_
> | Metric                        | Agreement (%) |
> |-------------------------------|----------------|
> | All samples             | 79.90          |
> | Positive samples      | 80.41          |
> | Negative samples      | 79.38          |
> | All citations     | 76.96          |
>
> Table 1 demonstrates a **high degree of agreement (79.90%)** between human annotations and our automated response labels, with **80.41% agreement** on positive responses and **79.38% agreement** on negative responses.
>
> Our findings also show a **high degree of agreement (76.96%)** between human annotations and TRUE NLI on the necessity (precision) and sufficiency (recall) of the citations. Additionally, Cohen’s kappa coefficient between humans and TRUE suggests a moderate level of agreement (0.55), underscoring the validity of using TRUE in our data construction pipeline.
>
> This level of agreement is **on par with prior studies** conducting human evaluations which typically have agreement rates in the range of 58-80% [1-4]. We appreciate the reviewer pointing out the important discussion on evaluating the alignment data quality which we will include in the appendix of the updated draft.
>
> ---
> **References:**
>
> [1] Wang, Y., Kordi, Y., Mishra, S., Liu, A., Smith, N. A., Khashabi, D., & Hajishirzi, H. (2023). Self-Instruct: Aligning Language Models with Self-Generated Instructions. Proceedings of the 61st Annual Meeting of the Association for Computational Linguistics (Volume 1: Long Papers). https://doi.org/10.18653/v1/2023.acl-long.754
>
> [2] Gao, T., Yen, H., Yu, J., & Chen, D. (2023). Enabling Large Language Models to Generate Text with Citations. Retrieved November 18, 2024, from arXiv.org website: https://arxiv.org/abs/2305.14627
>
> [3] Kamalloo, E., Jafari, A., Zhang, X., Thakur, N., & Lin, J. (2023). HAGRID: A Human-LLM Collaborative Dataset for Generative Information-Seeking with Attribution. Retrieved November 18, 2024, from arXiv.org website: https://arxiv.org/abs/2307.16883
>
> [4] Chia, Y. K., Cheng, L., Chan, H. P., Liu, C., Song, M., Aljunied, Sharifah Mahani, … Bing, L. (2024). M-Longdoc: A Benchmark For Multimodal Super-Long Document Understanding And A Retrieval-Aware Tuning Framework. Retrieved November 18, 2024, from arXiv.org website: https://arxiv.org/abs/2411.06176

---

> ### Author Response · Authors · 2024-11-21
> **Rebuttal by Authors**
>
> **[Clarification 2]: Statistical testing**
>
> Thank you for bringing this up. We intentionally omitted the significance testing results from the main draft to improve table readability and we are happy to discuss it here. However, if the reviewers feel these results should be included, we will make sure to incorporate them in the next version. For your reference, we provide below the results of an independent sample t-test comparing baseline models with Trust-Aligned models.
>
> _Table 2: Results from significance testing on Trust-Score_
> | Dataset   | t-statistic       | p-value                |
> |-----------|-------------------|------------------------|
> | ASQA      | 7.70 | 9.85e-10  |
> | ELI5      | 6.40 | 1.18e-06 |
> | QAMPARI   | 7.83 | 3.31e-08  |
> | EXPERTQA | 8.63 | 8.00e-10   |
>
> Across all datasets, we observed that **t-statistic > 0 and p < 0.001** (Table 2), indicating **significant improvements for Trust-Score at a significance level of 0.01**.
>
> ------
>
> **[Clarification 3]: Separation of train and test set/test-set integrity**
>
> An important aspect of any research is ensuring that the validation set and test set are well separated from the training set to maintain _test-set integrity_, which was a paramount consideration in our experiments. Thus, we ensure that the **test set remains as disjoint as possible from the training samples**. Precisely, the seed samples are sourced from the training split of three candidate data sources: ASQA, QAMPARI, and ELI5, totaling approximately 10,000 samples. Evaluation is conducted on their **corresponding disjoint test split (2,948 samples)** to preserve the separation between training and test data.
>
> To assess **generalizability**, we also tested on the **out-of-domain dataset ExpertQA**, comprising expert-curated samples from 32 fields such as medicine, law, history, and engineering. This evaluates the aligned models on **questions distinct from training domains** (e.g., Google, Reddit, and Wikipedia). We appreciate the reviewer bringing up this question and will ensure these details are included in the updated draft to avoid any further confusion.

---

### Meta-Review · Area_Chair_ZJmb · 2024-12-19

**Metareview:**

This paper introduces a method (Trust-Align) for enhancing a model’s response across three dimensions: trustworthiness, citation groundedness, and refusal groundedness,  measured by their groundedness metric, Trust-Score. The authors evaluate the proposed metric and alignment method across numerous models, datasets and configurations. The experiments are well documented and clearly articulated providing extensive support for the findings.

**Additional Comments On Reviewer Discussion:**

Lack of clarity: particularly Table 2 & 3 were difficult to read and interpret. The authors have subsequently revised and improved these with satisfactory clarity.

Distinction between parametric knowledge and groundedness:  this was addressed satisfactorily.

Trust-Score notation, confidence intervals, and statistical testing: in response to these, the authors have included further clarification and necessary additional information.

Overall a good contribution.

---

### Decision · Program_Chairs · 2025-01-22

Accept (Oral)